# Testing the efficacy of atmospheric boundary layer height detection algorithms using uncrewed aircraft system data from MOSAiC

Gina Jozef[1,2,3], John Cassano[1,2,3], Sandro Dahlke[4], Gijs de Boer[2,5,6]

[1]Dept. of Atmospheric and Oceanic Sciences, University of Colorado Boulder, Boulder, CO, USA
[2]Cooperative Institute for Research in Environmental Sciences, University of Colorado Boulder, Boulder, CO, USA
[3]National Snow and Ice Data Center, University of Colorado Boulder, Boulder, CO, USA
[4]Alfred Wegener Institute Helmholtz Centre for Polar and Marine Research, Potsdam, Germany
[5]NOAA Physical Sciences Laboratory, Boulder, CO, USA
[6]Integrated Remote and In Situ Sensing, University of Colorado Boulder, Boulder, CO, USA

*Correspondence to:* Gina Jozef (gina.jozef@colorado.edu)

**Abstract.** During the Multidisciplinary drifting Observatory for the Study of Arctic Climate (MOSAiC) expedition,
meteorological conditions over the lowest 1 km of the atmosphere were sampled with the DataHawk2 (DH2) fixed-
wing uncrewed aircraft system (UAS). These in situ observations of the central Arctic atmosphere are some of the
most extensive to date and provide unique insight into the atmospheric boundary layer (ABL) structure. The ABL is
an important component of the Arctic climate, as it can be closely coupled to cloud properties, surface fluxes, and the
atmospheric radiation budget. The high temporal resolution of the UAS observations allows us to manually identify
the ABL height ($Z_{ABL}$) for 65 out of the total 89 flights conducted over the central Arctic Ocean between 23 March
and 26 July 2020 by visually analyzing profiles of virtual potential temperature, humidity, and bulk Richardson
number. Comparing this subjective $Z_{ABL}$ with $Z_{ABL}$ identified by various previously published automated objective
methods allows us to determine which objective methods are most successful at accurately identifying $Z_{ABL}$ in the
central Arctic environment, and how the success of the methods differs based on stability regime. The objective
methods we use are the Liu-Liang, Heffter, virtual potential temperature gradient maximum, and bulk Richardson
number methods. In the process of testing these objective methods on the DH2 data, numerical thresholds were adapted
to work best for the UAS-based sampling. To determine if conclusions are robust across different measurement
platforms, the subjective and objective $Z_{ABL}$ determination processes were repeated using the radiosonde profile
closest in time to each DH2 flight. For both the DH2 and radiosonde data, it is determined that the bulk Richardson
number method is the most successful at identifying $Z_{ABL}$, while the Liu-Liang method is least successful. The results
of this study are expected to be beneficial for upcoming observational and modeling efforts regarding the central
Arctic ABL.

## 1 Introduction

The transfer of energy between the Earth's surface and the overlying atmosphere, particularly at high latitudes, remains
an area of substantial uncertainty in our understanding of the global climate system (de Boer et al., 2012; Tjernström
et al., 2012; Karlsson and Svensson, 2013). The consequences of this uncertainty are significant, with global climate
model projections of present-day sea ice demonstrated to fall short of simulating the observed rate of change (Stroeve

et al., 2007; Stroeve et al., 2012). The thermodynamic structure of the lower atmosphere plays a central role in regulating cloud lifecycle and radiative transfer, and their influence on atmospheric energy transport (Tjernström et al., 2004; Karlsson and Svensson, 2013; Brooks et al., 2017). Significant insight can be gained by measurements collected over the central Arctic Ocean pack ice, focused on the structure of the lower atmosphere, its spatial and temporal variability, the intensity of turbulent energy fluxes, and its connection to surface features. To provide such measurements, uncrewed aircraft were deployed in the lower atmosphere during legs 3 (March through May 2020) and 4 (June through August 2020) of MOSAiC (Multidisciplinary drifting Observatory for the Study of Arctic Climate; Shupe et al. 2020), a year-long expedition that took place from October 2019 to September 2020 in which the icebreaker RV *Polarstern* (Alfred-Wegener-Institut Helmholtz-Zentrum für Polar- und Meeresforschung, 2017) was frozen into the central Arctic ocean sea ice pack and allowed to passively drift across the central Arctic for an entire year (Fig. 1). Additional information on measurements taken of the atmosphere and sea ice during MOSAiC can be found at Shupe et al. (2022) and Nicolaus et al. (2022) respectively.

One important indicator of the extent to which energy may be transferred between the Earth's surface and overlying atmosphere is the atmospheric boundary layer (ABL) height. The ABL is the turbulent lowest part of the atmosphere that is directly influenced by the Earth's surface (Stull, 1988; Marsik et al., 1995). In the central Arctic, the ABL is impacted by interactions between the atmosphere and underlying surface, including both sea ice and open water portions, which can cause either buoyantly or mechanically produced turbulence. The generation of buoyant turbulence can occur through surface energy fluxes emitted from open water regions such as leads (Lüpkes et al., 2008), cold air advection, especially over thin ice (Vihma et al., 2005), or turbulent mixing below cloud base due to cloud top radiative cooling (Tjernström et al., 2004). Mechanical generation, which is the dominant driver of turbulence in the central Arctic (Brooks et al., 2017), can occur due to the interaction between the atmosphere and surface roughness features such as ridges and ice edges (Andreas et al., 2010) or oceanic waves (Jenkins et al., 2012), or due to the presence of a low-level jet (Brooks et al., 2017; Banta, 2003). Solar heating of the Earth's surface and the subsequent formation of buoyant thermals, which is a dominant forcing of the ABL in most parts of the planet (Marsik et al., 1995), plays only a minor role in the central Arctic due to the relatively reflective surfaces found there.

The Arctic ABL is usually either stable or near-neutral, while a convective ABL is rarely observed (Brooks et al., 2017; Esau and Sorokina, 2009). A stable boundary layer forms when there is a deficit of radiation at the surface or when warmer air is advected over a cooler surface, and can range from being nearly well-mixed with moderate turbulence to nearly laminar (Stull, 1988). A neutral boundary layer occurs when air at the surface is neutrally buoyant (Sivaraman et al., 2013) due primarily to mechanically generated turbulence which mixes air between the surface and above atmosphere (Brooks et al., 2017). A convective boundary layer forms when convective thermals create positive buoyancy (Liu and Liang, 2010) and an air parcel at the surface rises adiabatically until becoming neutrally buoyant; when this phenomenon occurs in the central Arctic, it is likely due to the presence of open water such as leads or polynyas (Lüpkes et al., 2008). While the various forms that the Arctic ABL may take are complex, most of the time, the Arctic ABL is capped by a temperature inversion (which may extend to the surface for a stable ABL) and local maximum in potential temperature gradient, marking the entrainment zone, which is a stable layer that makes the

transition from the ABL to the free atmosphere (Stull, 1988). One important difference between the Arctic ABL and that in the mid-latitudes is that there is usually no residual layer above a stable Arctic ABL, due to the lack of a diurnal cycle. Additionally, the Arctic ABL is typically much shallower than that at mid-latitudes (Esau and Sorokina, 2009). These discrepancies cause certain ABL height detection methods to fail when applied to Arctic data.

Knowing the height of the Arctic ABL is important for many applications. First, it is a metric which represents the altitude up to which the atmosphere is directly impacted by surface processes. This can then inform the extent to which the surface interacts with atmospheric features such as clouds (and their influence on radiative transfer in the lower atmosphere), low level jets (LLJs), and temperature inversion layers, which all have important implications for Arctic warming (Serreze and Barry, 2011). For example, a shallow, stable ABL is more likely to be observed with clear skies above (Brooks et al., 2017), which promotes longwave cooling of the surface and decoupling from the above atmosphere. In this instance, a surface-based temperature inversion is likely to constrain warming to the surface, which contributes to Arctic amplification (Lesins et al., 2012). ABL height (hereafter $Z_{ABL}$) plays an important role in many other applications including transfer of air pollutants and weather forecasting (Garratt, 1994), and the proper parameterization of the ABL in numerical weather prediction models. Since any determination of $Z_{ABL}$ is simply an approximation, the most value can be gained if this approximation is as accurate as possible. The goal of the current work is to determine which methods, based on thermodynamic and kinematic UAS profile data, can best accomplish this.

The depth of the ABL has been previously defined using a variety of approaches that involve visualizing the profiles of different thermodynamic and kinematic variables, which are listed in Table 1, along with some examples of associated literature that references use of that variable. Each of these profiles typically exhibits a distinct change in vertical structure at the top of the ABL. Additional methods may exist, such as analyzing the vertical gradient of aerosol content, but are not listed since the current study focuses on $Z_{ABL}$ determination using thermodynamic and kinematic processes.

**Table 1:** List of quantities previously used to identify $Z_{ABL}$, as well as some associated literature in which each variable is referenced.

| Quantity Used | Application of Quantity | Previous Literature |
|---|---|---|
| Virtual potential temperature ($\theta_v$) | $\theta_v$ difference across $\theta_v$ inversion exceeds a threshold at the top of the ABL | Heffter, 1980; Pesenson, 2003; Sivaraman et al., 2013 |
| | $\theta_v$ at the top of an unstable ABL equals $\theta_v$ at the surface | Stull, 1988; Liu and Liang, 2010; Collaud Coen et al., 2014; Seibert et al., 2000 |
| Vertical gradient of virtual potential temperature ($d\theta_v/dz$) | Comparing $d\theta_v/dz$ to a threshold differentiates between ABL, entrainment zone, or free atmosphere | Heffter, 1980; Stull, 1988; Steeneveld et al., 2007; Liu and Liang, 2010; Dai et al., 2011; Sivaraman et al., 2013; Dai et al., 2014; Zhang et al., 2014 |
| | Local maximum in $d\theta_v/dz$ at the top of the ABL | Dai et al., 2014 |

| Vertical gradient of temperature (dT/dz) | dT/dz = zero at the top of a stable ABL | Stull, 1988; Seibert et al., 2000; Dai et al., 2014; Collaud Coen et al., 2014 |
|---|---|---|
| | dT/dz ≤ the dry adiabatic lapse rate at the top of an unstable ABL | Collaud Coen et al., 2014 |
| Bulk Richardson number ($Ri_b$) | $Ri_b$ exceeds critical value above the ABL | Stull, 1988; Seibert et al., 2000; Zilitinkevich and Baklanov, 2002; Steeneveld et al., 2007; Georgoulias et al., 2009; Dai et al., 2011; Sivaraman et al., 2013; Dai et al., 2014; Zhang et al., 2014; Collaud Coen et al., 2014 |
| Total wind speed | Low-level jet occurs at the top of a stable ABL | Stull, 1988; Seibert et al., 2000; Steeneveld et al., 2007; Liu and Liang, 2010; Sivaraman et al., 2013; Zhang et al., 2014 |
| Wind shear | Component-wise wind shear is greatly reduced above the ABL | Dai et al., 2011; Dai et al., 2014; Zhang et al., 2014 |
| Liquid water content and absolute humidity | Air moisture decreases drastically at the top of the ABL | Stull, 1988; Seibert et al., 2000; Pesenson, 2003; Dai et al., 2014 |
| Turbulent kinetic energy (TKE) | TKE ceases at the top of the ABL | Stull, 1988; Seibert et al., 2000; Dai et al., 2014; Zhang et al., 2014 |

Due to the different atmospheric dynamics involved in each of the above approaches, the definition of $Z_{ABL}$ is often debatable amongst experts. Depending on one's purpose for knowing $Z_{ABL}$, different approaches may be most applicable. Of these methods, some of the most widely used ones, and the ones applied in the current analysis of a central Arctic dataset to determine $Z_{ABL}$, are the ones that involve analysis of virtual potential temperature ($\theta_v$), vertical gradient of virtual potential temperature ($d\theta_v/dz$), humidity (relative and absolute), bulk Richardson number ($Ri_b$), and

wind speed profiles. The current focus is on these variables because the physical basis for each one as an indication of $Z_{ABL}$ is relevant for the Arctic atmosphere. Specifically, $\theta_v$ helps identify the entrainment zone above the ABL, the vertical gradient of humidity either decreases or increases noticeably above the ABL (Dai et al., 2014), $Ri_b$ helps identify where turbulence (usually caused by strong wind shear or surface roughness in the Arctic ABL (Grachev et al., 2005)) ceases above the ABL, and wind speed helps identify the top of the ABL when it is capped by an LLJ as

the ABL top is often at or just below the LLJ core (Stull, 1988). Other methods, such as that using temperature inversion top to identify $Z_{ABL}$ (Collaud Coen et al., 2014), do not perform well in the Arctic where a weak temperature inversion can extend well above the ABL. Though turbulent kinetic energy is recognized as perhaps the most valuable profile for $Z_{ABL}$ identification (Stull, 1988; Seibert et al., 2000; Dai et al., 2014; Zhang et al., 2014), these data are not available to aid in the current study.

High resolution data collected by the DataHawk2 uncrewed aircraft system (UAS) allows for determination of $Z_{ABL}$ with high accuracy through manual visual analysis. However, visually determining $Z_{ABL}$ case-by-case is time consuming for processing a large dataset. Therefore, the UAS-derived dataset is leveraged to compare manually (or 'subjectively') determined $Z_{ABL}$ with that identified through previously published automated (or 'objective') methods. While this subjective $Z_{ABL}$ may not necessarily be the 'true' ABL top, as the definition of this quantity can be debatable

among experts and $Z_{ABL}$ is not constant over time, it is the best estimate of $Z_{ABL}$ given the available data. This

evaluation is completed to identify objective methods that can accurately diagnose $Z_{ABL}$ across a larger dataset of central Arctic atmospheric conditions.

To subjectively identify $Z_{ABL}$ in each atmospheric profile from DH2 data, the stability regime of the ABL (stable, neutral, or convective) is categorized and $Z_{ABL}$ is visually identified through combined evaluation of $\theta_v$, humidity
(both relative humidity (RH) and mixing ratio), and $Ri_b$ profiles. Objective identification of $Z_{ABL}$ is derived through the application of four previously published methods: the Liu-Liang method (Liu and Liang, 2010), the Heffter method (Heffter, 1980), the virtual potential temperature gradient maximum (TGRDM) method (Dai et al., 2014), and the $Ri_b$ method (Sivaraman et al., 2013), all adapted to best suit the DH2 profiles examined (aside from the Heffter method, which was kept as standard). Then, statistical comparisons between the objective and subjective $Z_{ABL}$ are conducted.
Next, the objective methods are applied in their adapted form to radiosonde profiles nearest in time to each DH2 flight to determine if these methods are robust across different measurement platforms for central Arctic conditions. Finally, discussion is included on the features that do or do not lend themselves to accurate identification of $Z_{ABL}$ by the objective methods, and findings are summarized to support future studies seeking to identify $Z_{ABL}$ quickly, objectively, and accurately across large atmospheric datasets collected in the central Arctic.

**2 Data and methods**

**2.1 The DataHawk2**

Data presented in this study were obtained between 23 March and 26 July 2020 using the University of Colorado DataHawk2 (DH2) UAS (de Boer et al., submitted). Flights were conducted from the sea ice alongside the *Polarstern*, known as the MOSAiC floe, ranging in location from 86.2° N, 15.8° E on 23 March, to 79.8° N, 1.9° W on 26 July
2020 (Fig. 1). Throughout this period, the MOSAiC floe evolved from snow-covered rigid ice situated in the high Arctic to being covered with melt ponds and leads close to the sea ice edge. The surface atmospheric temperatures also transitioned from nearly -35 °C at the beginning of leg 3 to hovering near 0 °C throughout the entirety of leg 4.

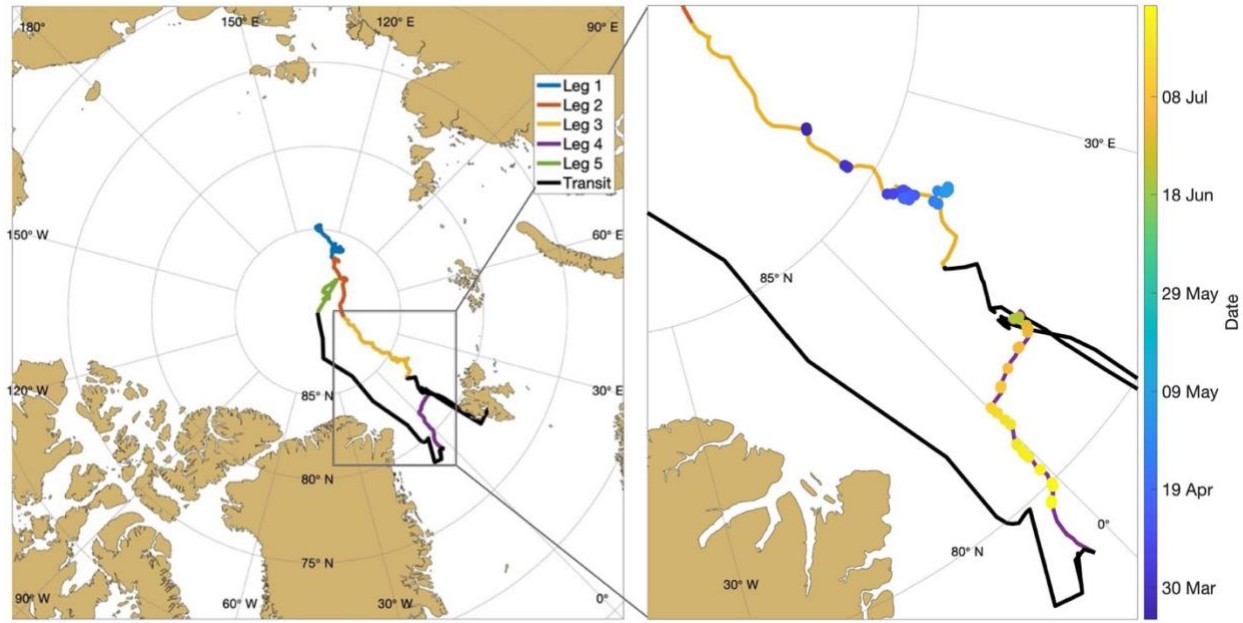

**Figure 1: (Left)** The drift track of the *Polarstern*, separated by color into the 5 different legs. The black "transit" line indicates when the ship was travelling under its own power between legs 3 and 4 and between legs 4 and 5. **(Right)** The zoomed in portion of the *Polarstern* drift during which DH2 flights were conducted (legs 3 and 4). The locations of all of the DH2 flights are overlaid on the drift track and color coded by date, with blue-tinted dots indicating flights conducted during leg 3 and yellow-tinted dots indicating flights conducted during leg 4.

The DH2 (Hamilton et al., 2022) is a fixed-wing, battery powered UAS (1.1 m wingspan, 1.8 kg weight, 40 min endurance) carrying various meteorological sensors, which measure the state of the atmosphere in Earth-relative coordinates. Instrumentation includes a fine wire array providing high frequency (800 Hz) information on temperature and air speed, multiple sensors for temperature and relative humidity (Vaisala RSS421 measuring at 5 Hz and SHT-85 measuring at 100 Hz), and up- and downward looking thermopile sensors to provide infrared brightness temperatures of the sky and surface. Air pressure is measured at 5 Hz by the Vaisala RSS421 sensor. Altitude estimates are obtained using a global navigation satellite system (GNSS) receiver and barometer onboard. The altitude used in the current analysis is a high-resolution (800 Hz) barometric pressure altitude, which is corrected for drift using the GNSS altitude.

Measurements of attitude from the inertial measurement unit, airspeed from a Pitot static probe and ground speed from the GPS receiver support the derivation of high-frequency (10 Hz) horizontal wind estimates. First, the "standard" approach, as laid out in van den Kroonenburg et al. (2008) and Hamilton et al. (2022), is applied, which derives wind estimates by combining GPS velocity measurements in the wind triangle using attitude estimates to rotate airframe-relative winds to Earth-relative coordinates. Additionally, a "hybrid" approach, as laid out in Lawrence and Balsley (2013) and Hamilton et al. (2022), is applied, which derives wind estimates by primarily using airspeed magnitude and GPS velocity, with secondary use of attitude estimates. For the purposes of this study, we use the DH2 winds derived from the "hybrid" approach. Please see Hamilton et al. (2022) and de Boer et al. (submitted) for additional details on the wind derivation as specifically applied to the DH2. Also, because take-offs

and landings were flown manually by a remote pilot, the winds calculated during these times were found to be less reliable and accurate. As a result, we do not use DH2 winds calculated below 30 m altitude for this study. A brief description of the processing methods for the above variables are provided in the metadata for the DH2 dataset used for the current study (Jozef et al., 2021).

Combined, these sensors provide a comprehensive picture of atmospheric thermodynamic and kinematic state along with some context on the surface and sky condition under which these measurements were obtained. Table 2 lists the resolution, repeatability (standard deviation of difference between two successive repeated calibrations), and response time for the Vaisala RSS421 sensor. Uncertainty in the wind speed estimation is not provided, as determining this is still in progress.

**Table 2:** Accuracy and reliability of the variables recorded by the Vaisala RSS421 sensors used in this study.

| Variable | Resolution | Repeatability | Response Time |
|---|---|---|---|
| Pressure | 0.01 hPa | 0.4 hPa | - |
| Temperature | 0.01 °C | 0.1 °C | 0.5 s |
| Humidity | 0.1 %RH | 2 %RH | <0.3 s (at 20 °C) to <10 s (at -40 °C) |

Measurements collected by the DH2 are logged at different frequencies, requiring the implementation of a time alignment process to assure that the time index for each datapoint of each variable is consistent with all other measurements. Data collected by the DH2 during MOSAiC are available for public download through the National Science Foundation Arctic Data Center at https://doi.org/10.18739/A2KH0F08V (Jozef et al., 2021).

During MOSAiC, DH2 flights were conducted whenever flight weather criteria were met and when the team was able to access the ice alongside the *Polarstern*. The weather criteria include near-surface wind speeds with a sustained average below 10 m s$^{-1}$, and gusts below 14 m s$^{-1}$, as well as sufficient visibility to maintain visual contact with the aircraft at all times during flight. In addition, DH2 flights required coordination with other MOSAiC activities, especially those impacting air space over the MOSAiC floe, including manned helicopter flights and other UAS and tethersonde operations.

The most common flight pattern conducted with the DH2, and the flight pattern from which data for this analysis were acquired, was a profiling flight in which the plane flew a spiral ascent and descent pattern, with a radius of 75-100 m between the surface and 1 km altitude (or cloud base, if lower than 1 km), with the aircraft ascending and descending at a rate of 2 m s$^{-1}$ and flying at an airspeed of 14-18 m s$^{-1}$. Each profiling flight lasted an average of 30 min, with some shorter flights when the air temperature was at its coldest (~-35 °C) near the beginning of leg 3, and some longer flights when the air temperature was much warmer (~0 °C) during leg 4. Throughout the measurement period, 89 flights were conducted with the DH2. In the present study, 65 of these flights are found to have a clearly identifiable $Z_{ABL}$ within the altitude range sampled. The remaining flights sampled only the lowest portion of the atmosphere due to cloud cover or other unfavorable environmental conditions and therefore did not observe the full depth of the ABL.

### 2.1.1 Preparing the DataHawk2 data for analysis

The primary profiles of interest for subjective and objective $Z_{ABL}$ identification are $\theta_v$, humidity (RH and mixing ratio), wind speed, $Ri_b$, and $d\theta_v/dz$. $\theta_v$ was calculated using RSS421 temperature, pressure, and RH. Differences in response times of the RSS421 temperature and RH sensors has a negligible impact on the calculation of $\theta_v$ because the moisture content in the Arctic atmosphere is so low that $\theta$ and $\theta_v$ values typically differ on the order of less than 1 K. Regardless, the addition of humidity does not change the structure and location of features in the $\theta_v$ profile, which is what is important for $Z_{ABL}$ identification. To further eliminate the effects of differences in sensor response times during ascent and descent, and for ease of visualization, we average the $\theta_v$, humidity, and wind speed variables over 1 m altitude bins throughout the entire flight (e.g., values at 10.5 m are averaged from 10 to 11 m). This also mitigates the effect of changes in atmospheric conditions near the surface throughout the span of a flight, though the near-surface observations largely remained constant during a given flight. 1 m is chosen as an averaging bin because using a greater bin value would eliminate much of the fine scale detail in the $\theta_v$ and humidity profiles which the DH2 provides, and which makes its data a valuable resource in honing $Z_{ABL}$ detection methods. However, since fine scale fluctuations in wind speeds evident at the 1 m scale are usually artifacts of the wind estimation routines applied to a circular flight pattern, we additionally apply a 60 m running mean, which eliminates small-scale wiggles while retaining the important large-scale features. Next, we exclude periods of manual flight during takeoff and landing (this is usually at altitudes below 5 m) since measurements during manual flight are prone to inaccuracies due to the irregular flight pattern. Lastly, we exclude the first 5 seconds of flight, as the initial measurements after takeoff may be faulty due to hysteresis associated with the sensor sitting still at the surface before launch.

Using the 1 m averaged $\theta_v$ and wind speed component profiles, we calculate the $Ri_b$ profile. $Ri_b$ is calculated at altitude, z, using the following equation from Stull (1988):

$$Ri_b(z) = \frac{\left(\frac{g}{\overline{\theta_v}}\right)\Delta\theta_v\,\Delta z}{\Delta u^2 + \Delta v^2} \tag{1}$$

where g is acceleration due to gravity, $\overline{\theta_v}$ is mean virtual potential temperature over the altitude range being considered, z is altitude, u is zonal wind, v is meridional wind, and $\Delta$ represents the difference over the altitude range used to calculate $Ri_b$ throughout the profile. The only way that $Ri_b$ can be negative is if the value for $\Delta\theta_v$ is negative, indicating a convective atmosphere with buoyancy-driven generation of the turbulence. $Ri_b$ profiles are created by calculating $Ri_b$ over a 30 m altitude range ($\Delta z$), at 5 m resolution (i.e., between 30 and 60 m, then between 35 and 65 m, and so on), rather than using the ground as the reference level, in order to isolate local likelihood of turbulence rather than that over the full depth from the surface (Stull, 1988; Georgoulias et al., 2009; Dai et al., 2014);

Since we do not use DH2 winds below 30 m, and intermediate $Ri_b$ value between the surface and 30 m is calculated using an assumed zero wind at the surface. This results in $Ri_b$ values at 15 m, 45 m, 50 m, 55 m, and so on. It is not crucial to consider the drift speed of the ice for the calculation of this initial $Ri_b$ value since the ice drift speed during MOSAiC was on average less than 0.1 m s$^{-1}$ (Krumpen et al., 2021), and the maximum drift speed during the DH2

flights was about 0.3 m s$^{-1}$, which is negligible compared to the speed of the observed winds. Nevertheless, any error in Ri$_b$ that ensues, due to the drift speed of the ice, is limited to the first level where Ri$_b$ is determined. Lastly, the d$\theta_v$/dz profile is similarly created by calculating it over an altitude range of 30 m, at 5 m resolution.

The above profiles are used to determine stability regime, visually identify Z$_{ABL}$ using criteria founded in this manuscript, and objectively identify Z$_{ABL}$ using the four published methods. For the remainder of this manuscript,
Z$_{ABL}$ determined from manual visual identification is referred to as the 'subjective' Z$_{ABL}$ and that determined by the published methods (which are automated algorithms performed by computers) are referred to as 'objective' Z$_{ABL}$. These terms are used as a simplification to differentiate between manual and automated methods, though they both consider much of the same underlying physical processes that dictate ABL structure and height.

**2.2 Determining the stability regime**

Some of the methods for both subjectively and objectively identifying Z$_{ABL}$ differ depending on the stability regime, so the sampled regime is first identified for each DH2 flight. The three possible stability regimes considered include a convective boundary layer (CBL), stable boundary layer (SBL), and neutral boundary layer (NBL; Liu and Liang, 2010). In a CBL, $\theta_v$ near the surface is greater than that of the overlying ABL (Stull, 1988). In an SBL, the vertical gradient of $\theta_v$ is positive (Stull, 1988). In an NBL, $\theta_v$ at the surface is approximately the same value as that of the
overlying remainder of the ABL (Stull, 1988).

Therefore, stability regimes are identified by comparing $\theta_v$ between the lowest altitude sampled by the DH2 ('i' in the below equations; typically ~5m since altitudes below this are usually sampled with manual flight) and 40 m above, using Eq. (2)-(4) below adapted from Liu and Liang (2010).

$$\theta_{v_{i+40m}} - \theta_{v_i} < -\delta_s = CBL \tag{2}$$

$$\theta_{v_{i+40m}} - \theta_{v_i} > +\delta_s = SBL \tag{3}$$

$$-\delta_s \leq \theta_{v_{i+40m}} - \theta_{v_i} \leq +\delta_s = NBL \tag{4}$$

In these equations, $\delta_s$ is a stability threshold that represents the minimum positive or negative vertical difference of $\theta_v$ near the surface necessary for the ABL to qualify as an SBL or CBL respectively. If this minimum is not either negatively (in the case of a CBL) or positively (in the case of an SBL) reached, the ABL is identified as an NBL (Liu
and Liang, 2010). In an idealized case, $\delta_s$ would be zero. However, in practice it must be specified as a small positive number, and this number depends on the surface characteristics as well as inherent uncertainties or noise in the measurements. For profiles over ocean/ice, this threshold has been defined to be 0.2 K (Liu and Liang, 2010).

While Liu and Liang (2010) compare $\theta_v$ between pressure levels that equate to approximately 40 and 160 m in the conditions we sampled, this range was found to be inadequate for differentiating between an SBL, NBL or CBL in the
Arctic, where the top of the ABL is often below 160 m, and sometimes even below 40 m. Therefore, considering the

$\theta_v$ change below ~45 m more accurately reflects the stability regime of the Arctic ABL. Once the stability regime is identified, criteria based on the $\theta_v$, humidity, and $Ri_b$ profiles are applied to subjectively determine $Z_{ABL}$. For the current dataset, 31 SBL cases, 32 NBL cases, and 2 CBL cases were identified.

### 2.3 Subjective identification of atmospheric boundary layer height

There is no one best method for subjectively identifying $Z_{ABL}$ that is agreed upon throughout the scientific community, evident by the many methods outlined in Table 1, and therefore a subjectively determined $Z_{ABL}$ is prone to error. The best we can do to increase the confidence in a subjectively determined $Z_{ABL}$ is to take into account several of the most commonly used methods and establish criteria which are applied consistently across all profiles. We describe these criteria below.

To subjectively identify $Z_{ABL}$, the $\theta_v$ profile is first analyzed, as the $\theta_v$ profile changes structure above the ABL (Stull, 1988). For a CBL and NBL, above the ABL, $\theta_v$ changes from decreasing or constant with height, to increasing with height, marking the entrainment zone (Stull, 1988). The structure of an SBL, however, can vary a lot more (Mayer et al., 2012; Steeneveld et al., 2007; Zilitinkevich and Baklanov, 2002). In an ideal SBL case, the $\theta_v$ inversion is at its strongest (greatest vertical gradient of $\theta_v$) near the surface and transitions to the free atmosphere (nearly constant or
gradually increasing $\theta_v$ with altitude) above the SBL, with no entrainment zone (Stull, 1988). $Z_{ABL}$ is then identified as the altitude of the shift from the surface-based $\theta_v$ inversion to the free atmosphere (Stull, 1988). In reality, the structure of an SBL is often not that simple, and the height of an SBL can be difficult to identify based on $\theta_v$ alone (Stull, 1988; Zhang et al., 2014). SBLs in the DH2 dataset often include a weaker surface-based $\theta_v$ inversion capped by a layer of enhanced stability (stronger $\theta_v$ inversion), reminiscent of an entrainment zone, likely because of surface-
drag induced turbulence close to the surface. ABLs with this structure form as the near-surface atmosphere fluctuates between weakly stable and near-neutral (Brooks et al., 2017). In more difficult cases such as these, the top of the SBL can be better determined by supplementing the $\theta_v$ profile with the RH and mixing ratio profiles, which usually have an obvious transition at the top of the ABL (Dai et al., 2014). This transition can manifest as either a shift from zero or positive to negative vertical gradient of humidity, or as a humidity inversion. Use of the humidity profiles can also
increase the confidence in identification of CBL and NBL height.

In addition, the $Ri_b$ profile can aid in $Z_{ABL}$ identification (Zhang et al., 2014). $Ri_b$ is an approximation of the ratio between buoyantly produced (from thermals) or suppressed (from static stability) turbulence, and mechanically produced turbulence (from wind shear; Sivaraman et al., 2013). Therefore, $Ri_b$ can help to identify the top of the ABL under the assumption that turbulence ceases above the ABL (Stull, 1988). In the limit of layer thickness becoming
small, $Ri_b$ can be compared to a critical value of ~0.25 (Stull, 1988), with $Ri_b$ below the critical value indicating an atmosphere that is likely to become or remain turbulent, and $Ri_b$ above the critical value indicating that an already laminar layer will not become turbulent, as static stability is strong enough to suppress mechanically generated turbulence. However, $Ri_b$ does not always assume a small layer thickness, so a critical value is not well defined for $Ri_b$. Thus, for $Ri_b$ near the critical value, there is uncertainty in the likelihood of turbulence (AMS Glossary of
Meteorology). However, since we calculate the profile of $Ri_b$ over layers with a consistent thickness of 30 m, we can

assume that the threshold for the likelihood of turbulence should at least be consistent throughout the profile. Additionally, since 30 m is a somewhat shallow thickness, there is less uncertainty in the likelihood of turbulence for $Ri_b$ near the critical value of 0.25 than if we calculated $Ri_b$ over an ever-increasing distance as we progress upward from the surface, when always using the ground as a reference level.

Different studies have found the appropriate critical Richardson number to range from as low as 0.15 to as high as 7.2 in coarse resolution models (Dai et al., 2014), but across the board, lower $Ri_b$ is expected in the ABL, and higher $Ri_b$ is expected above the ABL (Seibert et al., 2000). This increase in $Ri_b$ above the ABL is in large part due to the decrease in wind shear. By examining $Ri_b$ profiles for the DH2 flights, this transition from low values (near zero) to high values (with an increase of a few digits above the lower altitude values) can aid in identifying the top of the ABL.

Table 3 below outlines the subjective criteria applied to determine $Z_{ABL}$ depending on stability regime, which are separated depending on how many kinks there are in the $\theta_v$ profile that might indicate the entrainment zone. The term 'kink' refers to a dramatic shift in slope (i.e., drastic change in vertical gradient). The primary methods applied to determine $Z_{ABL}$ are those in which there are either one or two $\theta_v$ kinks, where we rely most heavily on the $\theta_v$ profile, and secondarily on the humidity and $Ri_b$ profiles. For SBL cases, the humidity profiles often provide more insight
than the $Ri_b$ profile in identifying $Z_{ABL}$. In only a few especially difficult cases, we relied primarily on the $Ri_b$ profiles.

**Table 3:** Subjective criteria for identifying $Z_{ABL}$, depending on stability regime.

| | **One $\theta_v$ kink** | **Multiple $\theta_v$ kinks** | **No clear $\theta_v$ kinks** |
|---|---|---|---|
| **Convective boundary layer (CBL)** | $Z_{ABL}$ is the altitude at which the vertical gradient of $\theta_v$ is positive and may be the bottom of a layer of enhanced stability (greater vertical gradient of $\theta_v$ above), corresponding to a kink in the relative and/or absolute humidity profiles and an increase in $Ri_b$.<br><br>Example: Fig. 2a | | |
| **Neutral boundary layer (NBL)** | $Z_{ABL}$ is the altitude of the singular $\theta_v$ kink marking the bottom of the lowest $\theta_v$ inversion.<br><br>Example: Fig. 2b | $Z_{ABL}$ is the altitude of the $\theta_v$ kink near the bottom of the lowest $\theta_v$ inversion which corresponds to a kink in the humidity profiles and an increase in $Ri_b$.<br><br>Example: Fig. 2c | $Z_{ABL}$ is the altitude of a faint $\theta_v$ slope shift which is identified via a corresponding kink in the humidity profiles and increase in $Ri_b$.<br><br>Example: Fig. 2d |
| **Stable boundary layer (SBL)** | $Z_{ABL}$ is the altitude of the $\theta_v$ kink marking the bottom of a layer of enhanced stability (greater vertical gradient of $\theta_v$), corresponding to a kink in the humidity profiles and sometimes an increase in $Ri_b$.<br><br>Example: Fig. 2e | | $Z_{ABL}$ is the altitude of a faint $\theta_v$ slope shift which is identified via a corresponding kink in the humidity profiles and sometimes an increase in $Ri_b$.<br><br>Example: Fig. 2f |

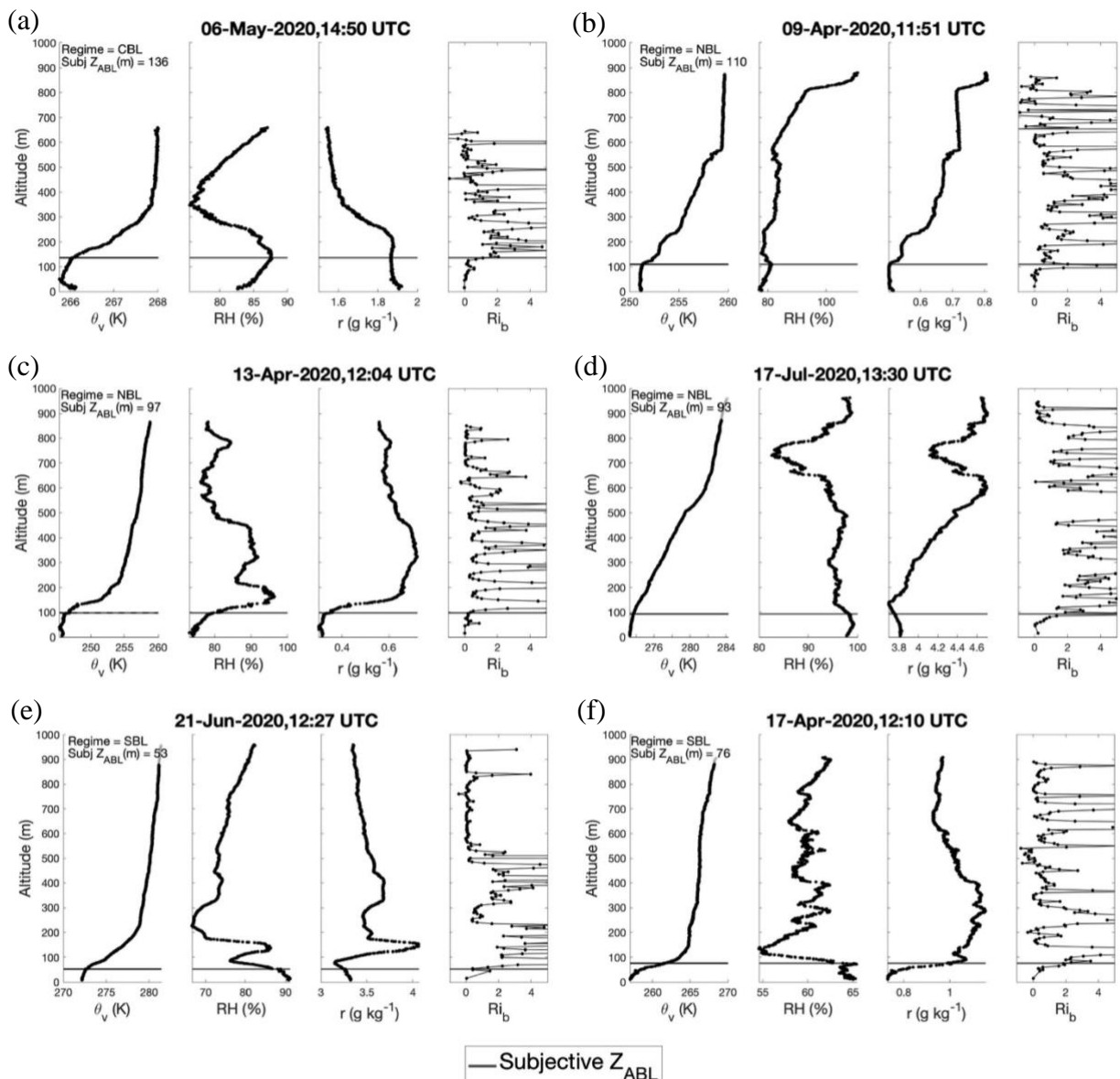

**Figure 2:** For each flight shown in the figure, the $\theta_v$ profile is plotted in the left panel, the RH and mixing ratio profiles are plotted in the middle two panels, and the $Ri_b$ profile is plotted on the right panel. Subjective $Z_{ABL}$ is marked with a horizontal black line on each panel, and is written, along with stability regime, on the left panel. **(a)** Example of a CBL case. **(b-d)** Examples of NBL cases. **(e-f)** Examples of SBL cases.

When applying the above criteria, $Z_{ABL}$ for the majority of cases (about 85%) was clearly identifiable (i.e., relevant $\theta_v$ and humidity kinks were at the same altitude). For the other cases, $Z_{ABL}$ was more ambiguous (e.g., Supplementary Figures S9, S12, S28, S34, S40, S42, S44, S48, S54, and S62), meaning there were multiple features that one could argue marked the ABL top (i.e., the $\theta_v$ and humidity kinks which could both be interpreted as $Z_{ABL}$ were at different altitudes). In these instances, depending on which feature is chosen, $Z_{ABL}$ could differ by on average about 10-30 m, but preferential treatment is given to the kink that also corresponds to an increase in $Ri_b$. Additionally, if kinks in the RH and mixing ratio profiles occur at different altitudes, preferential treatment is given to the kink which occurs at

the same altitude as that in the $\theta_v$ and/or $Ri_b$ profiles. Then, we determine the uncertainty in the subjective $Z_{ABL}$ to be less than 30 m. Uncertainty in the height of a kink in an individual profile is only subject to the vertical averaging procedure and sensor response time, and thus is on the order of only ~1 m.

**2.4 Objective identification of atmospheric boundary layer height**

The strength of the subjective method described above is the knowledge of the expert, which cannot be automated
(outside of possibly a machine learning algorithm, which would be costly and may still not be fully reliable). However, such expert knowledge and the time necessary to individually assess profiles is not always available. Thus, an automated method may often be preferred. Four such methods for objectively determining $Z_{ABL}$ are applied and evaluated. Each of these methods relies on profiles of either $d\theta_v/dz$ or $Ri_b$, some in combination with the $\theta_v$ and/or wind speed profiles. Because the $d\theta_v/dz$ and $Ri_b$ profiles are calculated over an altitude range of 30 m with 5 m
resolution, objective $Z_{ABL}$ detection methods which ultimately rely on these profiles can be determined with a resolution of 5 m. If they ultimately rely on the $\theta_v$ or wind speed profiles, $Z_{ABL}$ can be determined with 1 m resolution. Figure 3 at the end of Sect. 2.4 shows the application of all objective methods for an SBL and NBL case. A CBL case is not shown, as there were only two CBLs identified in the DH2 profiles, and they are rare in the central Arctic.

**2.4.1 Liu-Liang method**

The application of the Liu-Liang method depends on whether the profile includes a CBL, SBL, or NBL, which is determined using Eq. (2)-(4). To implement the Liu-Liang method for a CBL profile, we first find the lowest altitude at which $\theta_v$ exceeds its the lowest DH2 value by 0.1 K. Then, $Z_{ABL}$ is identified at the next lowest altitude in which $d\theta_v/dz$ exceeds 0.05 K 100 m$^{-1}$ (Liu and Liang, 2010). For an NBL, $Z_{ABL}$ is identified as the altitude at which $d\theta_v/dz$ first exceeds 2.5 K 100 m$^{-1}$, which is adapted from a threshold of 0.05 K 100 m$^{-1}$ used in Liu and Liang (2010), as this
threshold was found to be inappropriate for the current dataset ($Z_{ABL}$ found with the original threshold was always far too low). The basis of this method is to identify the entrainment zone at the top of the ABL through an increased value of $d\theta_v/dz$. The need for a greater threshold for NBL height identification in the current study is likely because the vertical resolution of sounding data used in the development of the Liu-Liang method was ~40-50 m (Liu and Liang, 2010), which would result in a much smoother $d\theta_v/dz$ profile than what is possible with the DH2 data. However, it
would not make sense to interpolate the DH2 profiles to a resolution of 40-50 m before applying the Liu-Liang method, as this would eliminate the ability the identify key features in the often shallow Arctic ABL.

For an SBL, the Liu-Liang method searches for a potential $Z_{ABL}$ associated with either minimal turbulence due to the lack of buoyancy within the ABL, or greater turbulence in the ABL due to the presence of wind shear (Liu and Liang, 2010), both scenarios which may dictate $Z_{ABL}$ for an SBL (Stull, 1988). Thus, SBL height is defined as either the top
of the bulk stable ($\theta_v$ inversion) layer starting from the ground, or the height of the LLJ maximum if present, whichever is lower (Liu and Liang, 2010). The top of the bulk stable layer is identified where the surface-based $\theta_v$ inversion has consistently diminished, and LLJ presence is identified by searching for wind speeds reaching a maximum that is at least 2 m s$^{-1}$ stronger than the local minima above and below (Stull, 1988; Liu and Liang, 2010). For greater detail on

these methods, and the guiding equations, see Liu and Liang (2010). Supplementary Figure S1 shows an example of the Liu-Liang method applied to a case for each stability regime.

### 2.4.2 Heffter method

The Heffter method uses $\theta_v$ difference across a $\theta_v$ inversion ($d\theta_v$) as an indication of $Z_{ABL}$ (Sivaraman et al., 2013), by identifying the lowest $\theta_v$ inversion layer where $d\theta_v/dz$ is greater than 0.5 K 100 m$^{-1}$ throughout the $\theta_v$ inversion, and $d\theta_v$ is at least 2 K (Heffter, 1980; Pesenson, 2003; Sivaraman et al., 2013). Within this $\theta_v$ inversion, the altitude at which $\theta_v$ first becomes more than 2 K greater than $\theta_v$ at the bottom of the $\theta_v$ inversion is labelled as $Z_{ABL}$ (Marsik et al., 1995; Delle Monache et al., 2004; Snyder and Strawbridge, 2004; Sivaraman et al., 2013).

For a CBL or NBL, this method is meant to determine the altitude of the elevated $\theta_v$ inversion marking the entrainment zone between the well-mixed ABL and free atmosphere (Pesenson, 2003). For an SBL, this method determines where the change in strength of the surface $\theta_v$ inversion marks the transition from the ABL to residual layer (if one exists) or free atmosphere above (Stull, 1988). For greater detail on this method, and the guiding equations, see Heffter (1980) or Sivaraman et al. (2013). Supplementary Figure S2 shows an example of the Heffter method applied to a case for each stability regime.

### 2.4.3 Virtual potential temperature gradient maximum (TGRDM) method

The final $d\theta_v/dz$-based method used to find $Z_{ABL}$ is the virtual potential temperature gradient maximum (TGRDM) method (Dai et al., 2014). Since the ABL is typically capped by a well-defined $\theta_v$ inversion layer (Stull, 1988), even in a weakly stable case, we expect to see a local maximum in the $d\theta_v/dz$ profile at this point. By finding the maximum in the $d\theta_v/dz$ profile, the altitude at which the $\theta_v$ inversion is at its strongest and weakens above is identified. To apply this method, local maxima in the $d\theta_v/dz$ profile where $d\theta_v/dz$ is at least 1.75 K 100 m$^{-1}$ greater than the local minimum $d\theta_v/dz$ above are identified. $Z_{ABL}$ is set to the altitude of this lowest peak. Supplementary Figure S3 shows an example of the TGRDM method applied to a case for each stability regime.

### 2.4.4 Bulk Richardson number method

Finally, a bulk Richardson number method for finding the ABL top is applied by determining the altitude at which $Ri_b$ exceeds a threshold value, which indicates where turbulence was likely no longer able to form in a laminar atmosphere. Previous literature suggests a wide range of critical values with 0.25 (Stull, 1988) being the most widely accepted value, though a value of 0.5 is also often used (Sivaraman et al., 2013; Zhang et al., 2014). To determine a viable threshold value for the identifying $Z_{ABL}$ in the DH2 data, a comparison between $Z_{ABL}$ determined from a range of threshold values (we used 0.25, 0.5, 0.75, 1.0, 1.25, and 1.5) and the subjective $Z_{ABL}$ was conducted. In identifying $Z_{ABL}$ from these different threshold values, the level above which $Ri_b$ was consistently greater than the threshold value was found. For this dataset, four consecutive datapoints (20 m) were required to be above the threshold value. We include this requirement due to the method of calculating $Ri_b$ over a rolling 30 m range, rather than always with the ground as the reference layer, as it is possible for $Ri_b$ to locally exceed the threshold, but still be within the ABL. Thus,

only when the $Ri_b$ consistently exceeds the threshold, indicating that the bulk likelihood for turbulence has ceased, can we be confident that the top of the ABL has been reached.

Then, the bottom of the lowest 20 m thick layer in which $Ri_b$ exceeds the threshold value is identified as $Z_{ABL}$. The threshold values deemed to identify $Z_{ABL}$ closest to that identified by the subjective method was 0.5 followed by 0.75. Therefore, further $Z_{ABL}$ presented using the $Ri_b$ method is calculated with threshold values of 0.5 (hereafter called $Ri_b(0.5)$) and 0.75 (hereafter called $Ri_b(0.75)$). Supplementary Figure S4 shows an example of the $Ri_b$ method applied to a case for each stability regime.

**2.5 Applying the objective methods to radiosonde profiles**

As discussed above, some of the objective methods used in this study were modified from their original descriptions to better work with the Arctic UAS data. Primarily, this includes changing the altitude range for determining stability regime, adjusting the threshold for calculating Liu-Liang NBL height, adding the 1.75 K 100 m$^{-1}$ criterion to the TGRDM method, and choosing the best threshold values as well as specifying the necessary vertical distance for the $Ri_b$ method. These adaptations are necessary in part because previous implementations involved analysis of radiosonde profiles, which have a lower vertical resolution than the DH2 profiles, and in mid-latitude locations, where the ABL structure is often quite different than that observed in the Arctic. Thus, profiles of $\theta_v$, humidity, and wind speed from the balloon-borne radiosondes that were launched at least four times per day from the deck of the *Polarstern* (Maturilli et al., 2021) during MOSAiC are leveraged to determine if the objective methods used to identify $Z_{ABL}$ from the UAS data are robust across platforms, despite differences in sampling methods.

To do this, radiosonde profiles with launch times closest to the DH2 flight times (within at most ~3 hours) are used, repeating the same processes for subjective and objective $Z_{ABL}$ identification and comparison. In eight instances, there were two DH2 flights in closest time proximity to the same radiosonde launch, so we use data from a total of 57 different radiosonde profiles. The specs for the Vaisala RS41-SGP sensor, which recorded the radiosonde variables, are the same as those listed in Table 2 for the DH2's RSS421 sensor, with the addition of pressure, temperature, and humidity uncertainty of 1.0 hPa, 0.3 ℃, and 4 % respectively, and a wind uncertainty and resolution of 0.15 m s$^{-1}$ and 0.1 m s$^{-1}$ respectively for velocity, and of 2 ° and 0.1 ° respectively for direction. The radiosonde samples with a frequency of 1 Hz, and an approximate climb rate of 5 m s$^{-1}$, which results in data with a vertical resolution of ~5 m. Altitude measurements are calculated with the hydrostatic equation using the initial pressure at 10 m. Before proceeding with analysis, profiles of temperature, wind, and humidity from the radiosondes were visually compared to those from the corresponding DH2 flight to confirm that the measurements were similar to each other.

Prior to applying the objective methods, data below 23 m altitude were removed, as the lowest part of the radiosonde profiles were found to show inaccurately warm temperatures for several cases (Maturilli et al., 2021), due to the *Polarstern* acting as a "heat island." Additionally, in some cases, the radiosonde data showed anomalously warm measurements some distance above 23 m, which is assumed to be the result of the balloon passing through the *Polarstern*'s exhaust plume. These measurements were adjusted by interpolating the temperature between the closest

good measurements above and below where the radiosonde was presumably in the ship's plume. Applying these adjustment means that radiosonde data near the surface are not available for the determination of the stability regime. Therefore, we adapt the methods applied to the DH2 data in Eq. (2)-(4) and instead calculate $d\theta_v$ between the lowest radiosonde measurement and 30 m above, or the subjective $Z_{ABL}$ if lower. We then compare this $d\theta_v$ to the appropriate

threshold value, $\delta_s$, that is equal to (0.2 K/40 m = 0.005 K m$^{-1}$) times the $\Delta z$ used. For example, if the $\Delta z$ of 30 m is used, the value of $\delta_s$ is 0.15 K. These adaptations in themselves do not result in the identification of a different stability regime than is found in the DH2 profiles; instead, differences in stability regime between the two platforms may result from the lack of near-surface observations from the radiosonde, or a change in atmospheric structure between the two corresponding launches.

Figure 3 shows two examples (one SBL and one NBL) of all of the objective methods applied to both a DH2 flight and its corresponding radiosonde. These examples show that the subjective $Z_{ABL}$ identified using the DH2 and radiosonde data are similar (differ by only 2 m for the SBL and 12 m for the NBL), and that the objective methods reveal a similar outcome when applied to the radiosonde data as they do for the DH2 data for both cases. Similar figures for all DH2 and radiosonde profiles used in this study can be found in Supplementary Figures S5-S69.

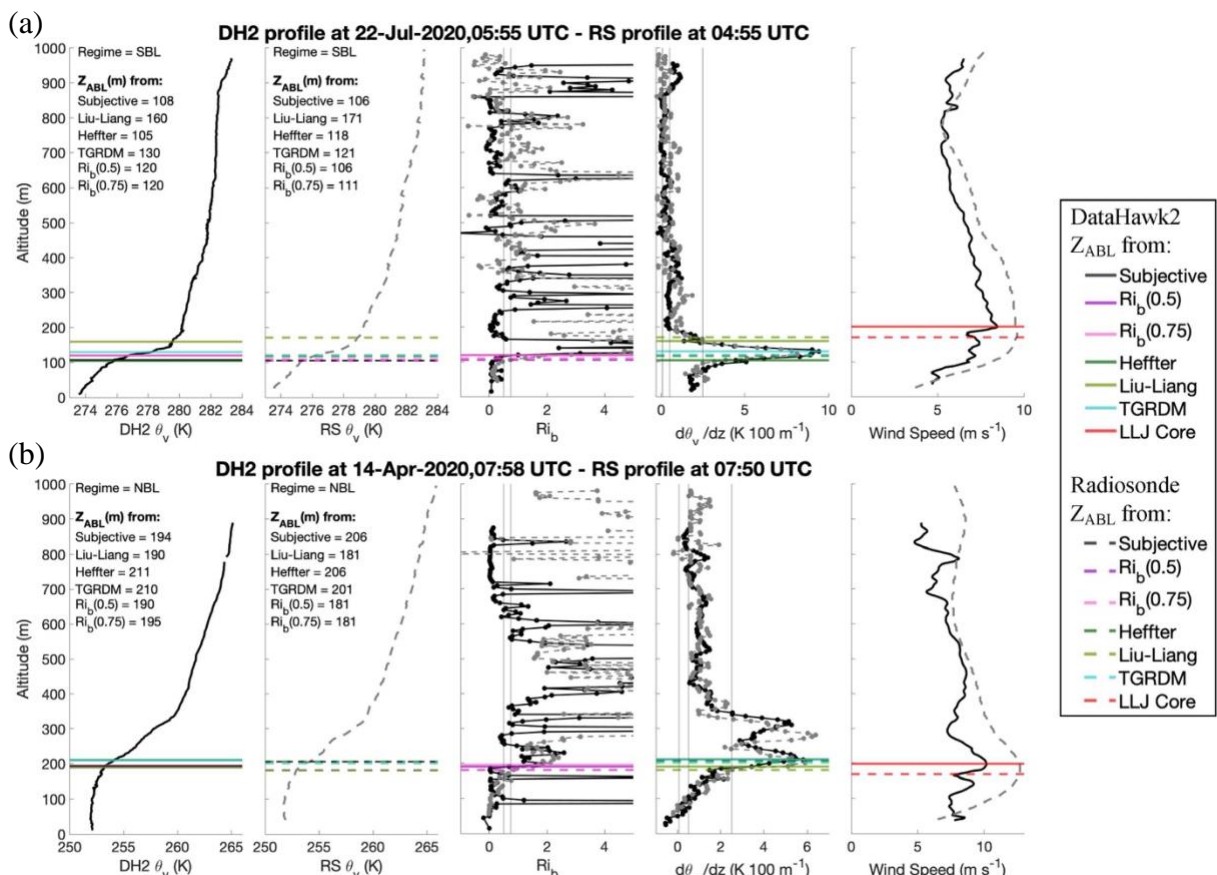

**Figure 3:** Demonstration of $Z_{ABL}$ identification using all objective methods on both the DH2 (represented by solid lines) and corresponding radiosonde (represented by dashed lines) for an **(a)** SBL and **(b)** NBL case. Panel 1: $\theta_v$ profile from the DH2. Panel 2: $\theta_v$ profile from the radiosonde. Panel 3: $Ri_b$ profiles from the DH2 (solid black) and the

radiosonde (dashed grey). Panel 4: $d\theta_v/dz$ profiles from the DH2 (solid black) and the radiosonde (dashed grey). Panel 5: wind speed profiles from the DH2 (solid black) and radiosonde (dashed grey). The legend on the right indicates the $Z_{ABL}$ detection method associated with each horizontal line in the figure. LLJ core is not in itself a $Z_{ABL}$ detection method, but plays into the Liu-Liang method, so it is included. Each $Z_{ABL}$ is written on the corresponding platform's $\theta_v$ profile.

While the radiosonde and DH2 profiles generally exhibit a similar structure due to the close time and space proximity

(the radiosondes were launched <600 m from the DH2 flights), the subjective $Z_{ABL}$ identified in those profiles differ

by 1-101 m. In general, the deviation between $Z_{ABL}$ from the DH2 and the radiosonde increases with increasing time

proximity. Figure 4 shows the absolute difference between DH2 and radiosonde subjective $Z_{ABL}$ (top panel), as well

as the absolute difference between the DH2 and radiosonde objective $Z_{ABL}$ for each method (bottom panel) as a

function of time difference in minutes between the DH2 and radiosonde launch. The best fit linear regression for each

method shows that as time between the DH2 and radiosonde launch increases, the differences in $Z_{ABL}$ increase as well,

though minimally. However, the increase in absolute difference between subjective $Z_{ABL}$ from the DH2 and radiosonde

as time between the launches increases is not significant at the 5% significance level (probability value of 0.74).

Therefore, we are confident that $Z_{ABL}$ does not significantly change for DH2 and radiosonde launches up to 3.16 hours

apart, which justifies the use of the radiosonde closest in time to each DH2 to test if there is similar efficacy of the

different objective methods.

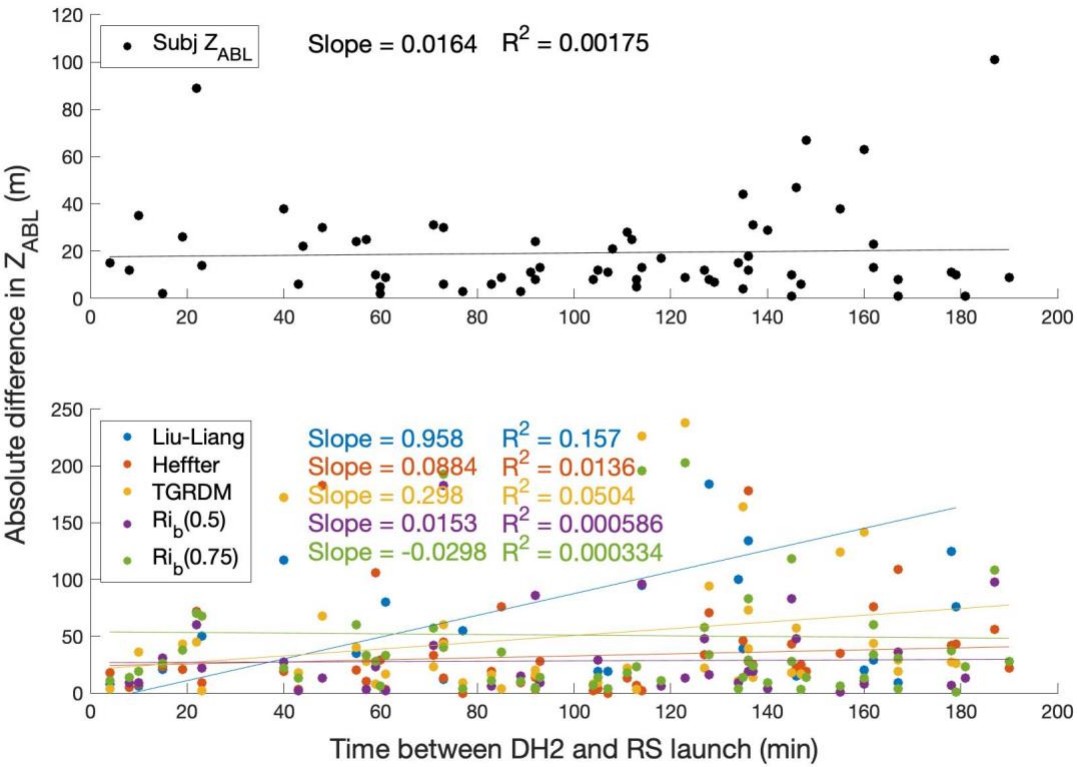

**Figure 4:** Absolute difference between subjective $Z_{ABL}$ from the DH2 and subjective $Z_{ABL}$ from the radiosonde closest in time to the DH2 launch (black dots, top panel) and absolute difference between objective $Z_{ABL}$ from the DH2 and objective $Z_{ABL}$ from the radiosonde closest in time to the DH2 launch (colored dots, bottom panel) versus

absolute time difference in minutes between the DH2 and radiosonde launches. A few outlier points are not shown,

as they lie outside the y-axis range. Lines of best fit are included for the subjective $Z_{ABL}$ and for each objective method, and the slope and $R^2$ value of each line is written next to the legend.

## 3. Results and discussion

### 3.1 Efficacy of objective $Z_{ABL}$ identification methods

Whereas the objective methods all rely on information from one variable (or two, in the case of the Liu-Liang method for an SBL), the subjective method uses a combination of methods which can only be weighted properly by visual analysis. This is why the subjective method arguably results in a more accurate $Z_{ABL}$ identification and provides a good basis for comparison with $Z_{ABL}$ identified by the objective methods.

To determine how well the different objective methods worked, $Z_{ABL}$ identified by each objective method is compared
to the subjective $Z_{ABL}$. Figure 5 shows scatter plots comparing the objective to the subjective $Z_{ABL}$ in each case, along with the associated best fit linear regression, coefficient of determination ($R^2$), slope, and probability value (p-value) resulting from a paired two sample T-test. For instances in which there were two DH2 flights in closest time proximity to the same radiosonde launch, the results from that radiosonde profile are plotted only once.

The $R^2$ value demonstrates how much of the variation in objective $Z_{ABL}$ can be explained by the difference in subjective
$Z_{ABL}$. Slope values (m) are also included to help evaluate the level of correspondence between the subjective and objective $Z_{ABL}$ by comparison to an ideal value of m = 1.00. Additionally, looking at the intercept combined with the slope value tells us whether the objective method tends to over- or underestimate $Z_{ABL}$ compared to the subjective method. Lastly, the p-value tells us whether the relationship between subjective and objective $Z_{ABL}$ can be considered statistically significant at the 5% significance level (a p-value less than 0.05 indicates that there is a 95% chance the
relationship is due to true correlation).

Based on the DH2 data in these scatter plots, the method that gives the greatest $R^2$ is the $Ri_b(0.5)$ method ($R^2 = 0.653$, Fig. 5d), followed by the $Ri_b(0.75)$ method ($R^2 = 0.537$, Fig. 5e). These are followed closely by the Heffter method ($R^2 = 0.485$, Fig. 5b). The TGRDM method has the fourth highest $R^2$ ($R^2 = 0.316$, Fig. 5c). The only objective method with a very low $R^2$ is the Liu-Liang method ($R^2 = 0.0907$, Fig. 5a). The slope values for all methods fall within m =
1.00 ± 0.30, the closest to 1.00 being the $Ri_b(0.75)$ method (m = 1.02), followed by the TGRDM method (m = 1.10) and Heffter method (m = 1.18). These slope values greater than 1.00 and positive intercept indicate that these methods generally overestimate $Z_{ABL}$ when applied to the DH2 data, compared to the subjective $Z_{ABL}$. The results of the $Ri_b(0.5)$ method and the Liu-Liang method, however, are more complex, as the slope values are both less than 1.00 (m = 0.721 and 0.708 respectively), but the intercepts are both positive. This indicates that these methods overestimate $Z_{ABL}$ for
a shallow ABL, but underestimate it for a deep ABL when applied to the DH2 data. Comparing the p-values for all relationships to the 5% significance level, the relationship between subjective and objective $Z_{ABL}$ can be considered significant for every method (p-value is less than 0.05). These p-values follow the same order as the $R^2$ values, with the lowest p-value found for the $Ri_b(0.5)$ (indicating the highest significance) and the highest p-value for the Liu-Liang method (indicating the lowest significance).

The radiosonde data gives a slightly different conclusion. Here, the method that gives the greatest $R^2$ is the Heffter method ($R^2 = 0.558$, Fig. 5b), followed by the $Ri_b(0.5)$ method ($R^2 = 0.420$, Fig. 5d). The $Ri_b(0.75)$ method and the TGRDM method have lower $R^2$ ( R2 = 0.207 and 0.225 in Fig. 5e and 5c, respectively). As was the case for the DH2 data, the only objective method with a very low $R^2$ is the Liu-Liang method ($R^2 = 0.00597$, Fig. 5a), which is also echoed by a slope value far from 1.00 (m = 0.171). The slope values for the rest of the methods are not as close to

1.00 as they are for the DH2 data, but they all fall within m = 1.00 $\pm$ 0.50. The TGRDM has a slope value of m = 1.00, and the method with the next closest value to 1.00 is the Heffter method at m = 1.13. Both of these methods have a positive intercept, which indicates that these method tends to overestimate $Z_{ABL}$ when applied to the radiosonde data used in the current study. The rest of the methods have a slope of less than 1.00 and positive intercept, indicating that they tend to overestimate $Z_{ABL}$ for a shallow ABL, but underestimate it for a deep ABL when applied to the radiosonde

data used in the current study. However, as $R^2$ for the Liu-Liang method is very low, this indicates that there is not much correlation between the objective and subjective $Z_{ABL}$ for this method, so analysis of the slope does not provide reliable information. Lastly, the p-values follow the same order as the $R^2$ values, with the lowest p-value found for the Heffter method (indicating the highest significance) and the highest p-value for the Liu-Liang method (indicating the lowest significance). Unlike the DH2 results, for the radiosonde, the p-values for all relationships compared to the 5%

significance level show that the relationship between subjective and objective $Z_{ABL}$ can be considered significant for every method except the Liu-Liang method, in which the p-value is greater than 0.05.

Lastly, Fig. 5f compares subjective $Z_{ABL}$ from the radiosondes to subjective $Z_{ABL}$ from the DH2. The high $R^2$ (0.752) indicates a rather strong correlation between subjective $Z_{ABL}$ from both platforms, which demonstrates that $Z_{ABL}$ usually did not change much between the DH2 and radiosonde launches in each case. Interestingly, there is enhanced

deviation from the line of best fit for a shallower ABL, and better agreement for a deeper ABL. However, this might simply be due to the greater number of samples with $Z_{ABL}$ below ~200 m. The very low p-value of 2.62e-18 demonstrates the high significance in the relationship between $Z_{ABL}$ from the DH2 and radiosondes.

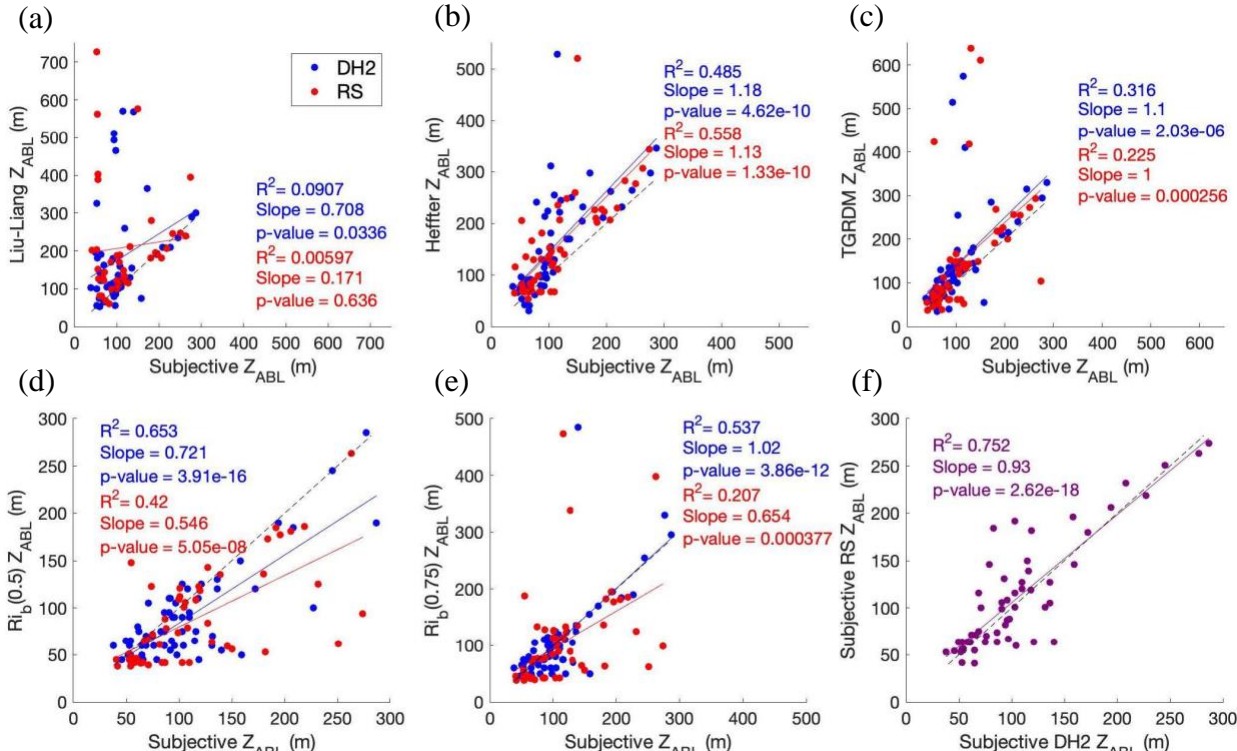

**Figure 5:** Relationships between subjective $Z_{ABL}$ and objective $Z_{ABL}$ from the **(a)** Liu-Liang method (50 DH2 samples and 40 RS samples), **(b)** Heffter method (61 DH2 samples and 53 RS samples), **(c)** TGDRM method (62 DH2 samples and 55 RS samples), and **(d, e)** $Ri_b$ method (65 DH2 samples and 57 RS samples). Blue dots represent DH2 data and red dots represent radiosonde data. The solid blue line (solid red line) on each panel is the line of best fit for the DH2 (radiosonde) data. **(f)** Relationship between subjective $Z_{ABL}$ from the radiosonde and subjective $Z_{ABL}$ from the DH2 with line of best fit in purple (57 samples). Each panel is overlaid by the corresponding $R^2$, slope value, and p-value. The dashed black line on each panel is a line with slope of 1.00 and y-intercept of 0, for reference.

Figure 6 shows the results presented in Fig. 5, but separated by stability regime, where the top panel shows results for only SBLs, and the bottom panel shows results for only NBLs. One primary takeaway from separating the results into stability regime is that, for both platforms, the TGRDM methods perform better for SBLs than it does for NBLs. Similarly, the Heffter method performs better for SBLs than NBLs for the DH2 data, and performs similarly for the radiosonde data. This discrepancy is likely because these two methos search for a $\theta_v$ inversion to identify $Z_{ABL}$, which is often more defined for an SBL than NBL. Next, for the DH2 data, the $Ri_b$ methods show less dependency on stability, with rather high $R^2$ for both regimes, however the higher threshold performs better for NBL cases. Additionally, when splitting into stability regimes, the discrepancy between DH2 and radiosonde results increases for some methods. For example, the $Ri_b$ method has more outliers for radiosonde NBL cases (Fig. 6i and 6j), causing $R^2$ to be rather low. For this category, the $Ri_b(0.5)$ method performs better, suggesting that the lower threshold value is more robust across platforms. Lastly, the Liu-Liang method, aside from a few outliers, performs rather well for NBL cases (Fig. 6f).

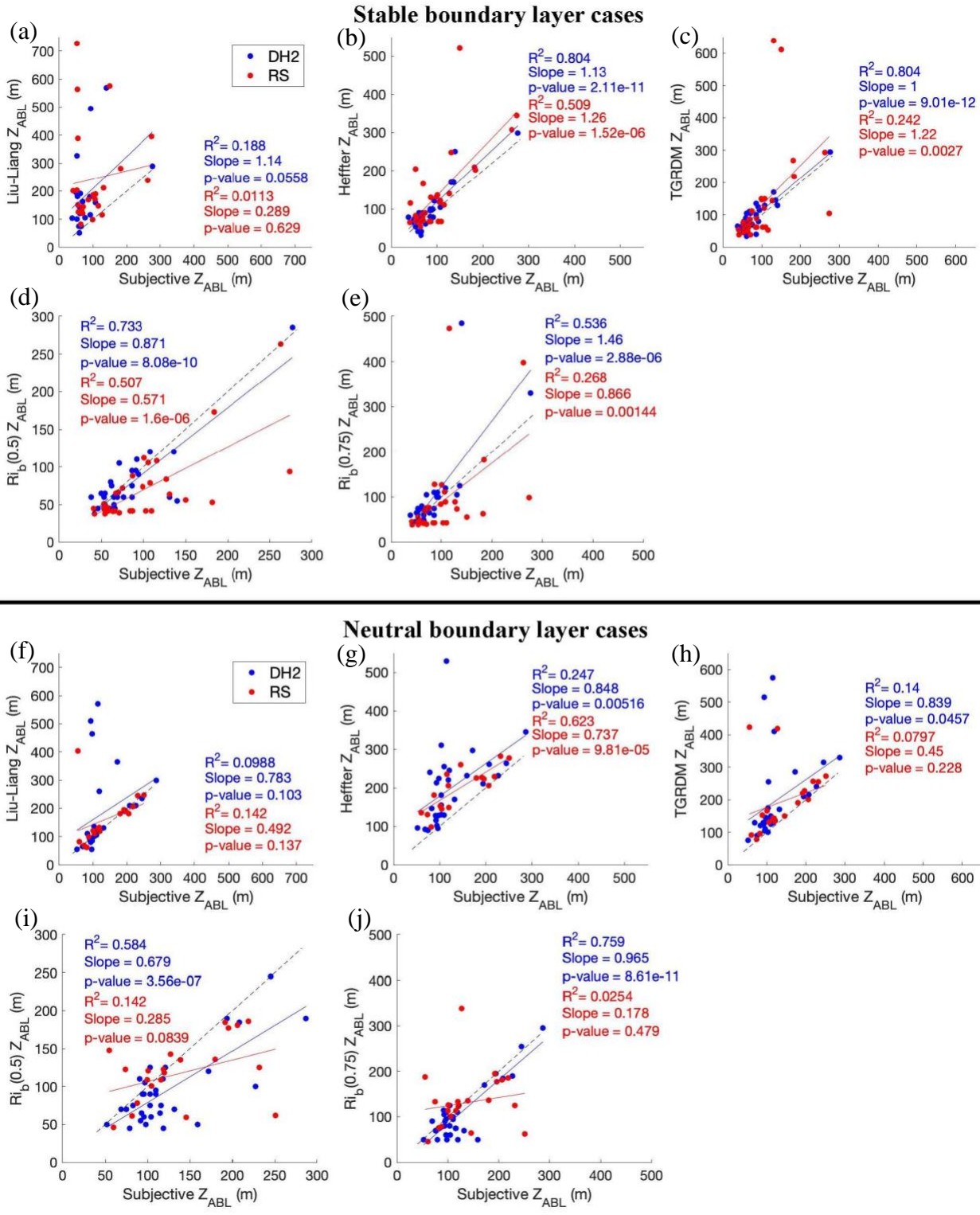

**Figure 6:** Relationships between subjective $Z_{ABL}$ and objective $Z_{ABL}$ for only stable cases (top) and only neutral cases (bottom) from the **(a, f)** Liu-Liang method (20 DH2 and 23 RS samples for SBL cases; 28 DH2 and 17 RS samples for NBL cases), **(b, g)** Heffter method (30 DH2 and 35 RS samples for SBL cases; 30 DH2 and 18 RS samples for NBL cases), **(c, h)** TGDRM method (31 DH2 and 35 RS samples for SBL cases; 29 DH2 and 20 RS samples for NBL cases), and **(d-e, i-j)** $Ri_b$ method (31 DH2 and 35 RS samples for SBL cases; 32 DH2 and 22 RS

samples for NBL cases). Blue dots represent DH2 data and red dots represent radiosonde data. The solid blue line
       (solid red line) on each panel is the line of best fit for the DH2 (radiosonde) data. Each panel is overlaid by the
       corresponding $R^2$, slope value, and p-value. The dashed black line on each panel is a line with slope of 1.00 and y-
       intercept of 0, for reference.

       Additional analysis was completed to assess the cumulative frequency distribution for the difference in objective $Z_{ABL}$
relative to the subjective $Z_{ABL}$. To do this, relative difference between the objective and subjective $Z_{ABL}$ in each case
       and for each method was determined. These results are included in Fig. 7a for the DH2 profiles, and in Fig. 7b for the
       radiosonde profiles. For example, about 26% of the time, the Liu-Liang $Z_{ABL}$ was within 10% of the subjective $Z_{ABL}$
       for the DH2 data.

       Figure 7a shows that, for the DH2 profiles, the $Ri_b(0.75)$ method results in the highest percent of cases to be within
10% of the subjective $Z_{ABL}$, followed by the $Ri_b(0.5)$ method. Interestingly, the Liu-Liang method results in the third
       highest percent of cases to be within 10% of the subjective $Z_{ABL}$. However, the Liu-Liang method falls behind other
       methods as the relative difference range is increased above 20%. Additionally, the Liu-Liang method has the highest
       percent of cases in which no $Z_{ABL}$ is found at all for the DH2 profiles, as well as about 20% of cases that have greater
       than 100% difference from the subjective $Z_{ABL}$. This trend indicates that, while the Liu-Liang method sometimes
works to find a $Z_{ABL}$ close to the subjective $Z_{ABL}$, it also fails to find a $Z_{ABL}$ close to the subjective $Z_{ABL}$, or to find any
       $Z_{ABL}$, in many cases. The primary reason for the failure of the Liu-Liang method, which is listed in Table 4 and
       discussed further in Sect. 3.2 below, is the high prevalence of a weak $\theta_v$ inversion that persists throughout the entire
       lower atmosphere in the Arctic. Another important finding is that the $Ri_b$ method using either threshold value never
       fails to find a $Z_{ABL}$, and the number of cases within each relative difference range is greater for the $Ri_b$ method than
that for all other methods.

       The information presented in the bar graph for the radiosonde profiles (Fig. 7b) leads to a similar conclusion. As for
       the DH2 profiles, the $Ri_b$ method results in the highest percent of cases to be within 10% of the subjective $Z_{ABL}$ (but
       for this platform, the $Ri_b(0.5)$ method does best). Here, the Liu-Liang method results in the fourth highest percent of
       cases to be within 10% of the subjective $Z_{ABL}$, and performs more poorly as the relative difference range is increased.
The Liu-Liang method also has the highest percent of cases in which no $Z_{ABL}$ is found at all, followed by the Heffter
       and TGRDM methods, which was also true for the DH2 data. As for the DH2, there are no radiosonde cases in which
       the $Ri_b$ method with either threshold value finds no $Z_{ABL}$. The main difference between Fig. 7b of the radiosonde data
       and Fig. 7a of the DH2 data is that, while the $Ri_b(0.75)$ method applied to the DH2 data was always more successful
       than the $Ri_b(0.5)$ method for relative difference ranges below 70%, for the radiosonde data, the $Ri_b(0.5)$ method proves
to always be more successful than the $Ri_b(0.75)$ method. We suspect that this results from the radiosonde data being
       more smoothed, which produces less sporadic $Ri_b$ values as the atmosphere transitions from the ABL to the free
       atmosphere, compared to the less smoothed DH2 data. This smoothing of the radiosonde data is applied by the Vaisala
       software to remove any effect of the chaotic pendulum swing directly after launch, while the wire unwinds. Thus, a
       lower threshold $Ri_b$ value may be better applicable when more smoothing or filtering procedures are applied to a
dataset.

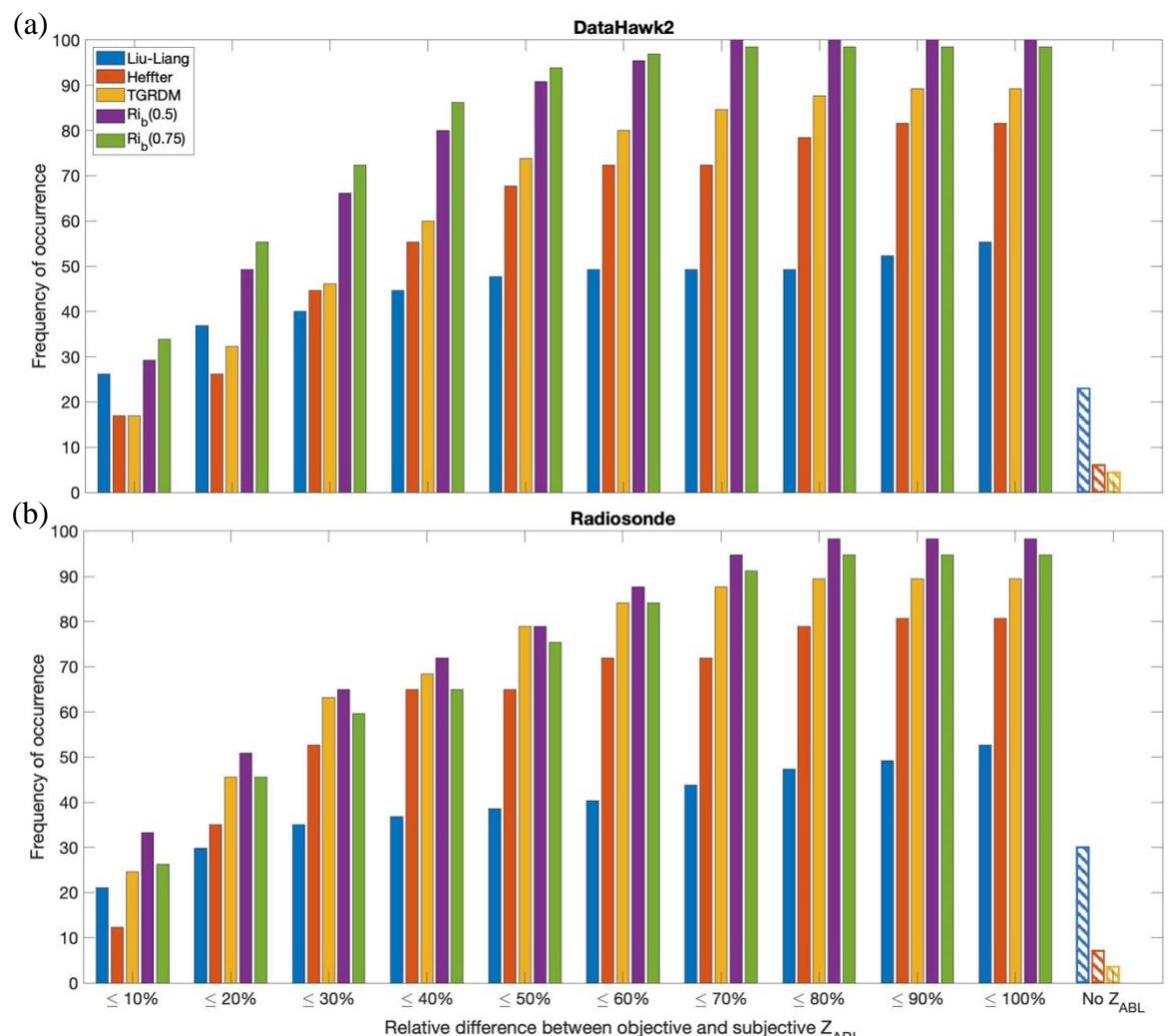

**Figure 7:** Bar plot showing what percent of **(a)** DH2 cases and **(b)** radiosonde cases give an objective $Z_{ABL}$ within different relative difference ranges from the subjective $Z_{ABL}$ using the different objective methods. Plot also shows the percent of cases for each method where no $Z_{ABL}$ is found (labelled as "No $Z_{ABL}$").

Supplementary Figures S70 and S71 show the results presented in Fig. 7, but separated by stability regime, where S70 shows results for only SBLs, and S71 shows results for only NBLs. The primary takeaways from separating the results into stability regime is that, for both the DH2 and radiosonde, the $Ri_b$ method has the most cases and the Liu-Liang method has the least cases with objective $Z_{ABL}$ within 10% of the subjective $Z_{ABL}$ for SBLs, though the Heffter and TGRDM methods also do well. For NBLs, the Liu-Liang method actually has the most cases with objective $Z_{ABL}$

within 10% of the subjective $Z_{ABL}$, followed by the $Ri_b$ method, for both platforms.

After comparing $Z_{ABL}$ from the different objective methods to the subjective $Z_{ABL}$ for both the DH2 and the radiosondes (Fig. 5 and 7), it is found that, with the exception of the Liu-Liang method, all other methods generally provide a reasonable estimate of $Z_{ABL}$ for both datasets, with the $Ri_b$ method being most favorable. This is in agreement with Siebert et al. (2000), Dai et al. (2014) and Zhang et al. (2014) which found an $Ri_b$-based method to be preferred

when mechanically-produced turbulence dominates, as is true in the central Arctic (Brooks et al., 2017). Additionally,

the efficacy of each method is similar for the DH2 and the radiosonde data, as is indicated by similar patterns in the scatter plots (Fig. 5) and bar plots (Fig. 7), despite occasional differences in radiosonde versus DH2-based $Z_{ABL}$ estimates, which likely result from the differences in sampling methods between the two platforms. Most specifically, the DH2 samples very close to the surface (~5 m) in most cases, so it observes important ABL features that support

accurate stability and $Z_{ABL}$ identification, whereas the radiosonde, which only samples down to 23 m at the lowest, may miss these features. Additionally, the DH2 samples with higher vertical resolution (due to higher time resolution of instrumentation and slower climb rate), again contributing to its ability to record complex fine scale features which the radiosonde might miss. However, the similarity in efficacy of the objective methods between both platforms supports the fact that the objective $Z_{ABL}$ identification methods that were adjusted using the high resolution DH2 data

are indeed robust across platforms with different sampling methods.

This is further explored by re-running the analysis with DH2 profiles averaged over 5 m, 10 m, and 20 m bins instead of 1 m bins, to determine how sensitive the efficacy of the methods is to the vertical resolution of the data. When comparing objective $Z_{ABL}$ found using the coarser data to the original subjective $Z_{ABL}$ for each method, the F-test reveals that generally the $R^2$ values do not differ significantly from those found using 1 m binned data at the 5%

significance level. The only exceptions are the Liu-Liang method at all larger bin sizes, and the Heffter method when using a 10 m or 20 m bin size, which all manifest in lower $R^2$ value than those found using 1 m binned data. This reveals that the Liu-Liang method performs even more poorly at lower vertical resolution, and the Heffter method starts to perform more poorly at a vertical resolution of 10 m. On the other hand, the $Ri_b$ and TGRDM methods remain just as successful when vertical resolution is reduced, and the preferred $Ri_b$ threshold value does not appear to depend

on vertical resolution. For vertical resolution of 30 m or coarser, the altitude range over which $Ri_b$ is calculated would have to be increased, and at this point a lower threshold $Ri_b$ value may be more applicable.

While we state an uncertainty in the subjective $Z_{ABL}$ to be less than 30 m, this is only applicable to a handful of DH2 flights (~15%), whereas the majority have an uncertainty on the order of only ~1 m, due to the vertical averaging procedure and sensor response time. Therefore, we do not expect this uncertainty to make any significant effect on

the results.

### 3.2 When the objective methods fail

Table 4 lists the most common features which cause each objective method to fail (meaning the objective $Z_{ABL}$ is much different than the subjective $Z_{ABL}$), along with the corresponding failure (either over- or underestimation, or no $Z_{ABL}$ found) and an example of such a situation shown in the Supplementary Figures. As shows in Sect. 3.1, while the

Liu-Liang method sometimes works well, it is not reliable across a wide range of different profile structures. Option 1a causes failure because the $d\theta_v/dz$ criteria are not met anywhere in the profile, meaning that the method reverts to using the LLJ core height as $Z_{ABL}$. However, the LLJ core was observed to usually be above the subjective $Z_{ABL}$ (supported by Stull, 1988; Jakobson et al., 2013; and Mahrt et al., 2014). This cause for failure agrees with Dai et al. (2014) which found that using LLJ core height to define SBL top produces results inconsistent with those from other

methods. The Liu-Liang method likely performs better for NBL cases (as is evident in Fig. 6 and Supplementary

Figure S71) than SBL cases because the Liu-Liang method for an NBL is not dependent on the sufficient diminishment of the $\theta_v$ inversion, nor the presence or altitude of a LLJ.

Any of the other objective methods would be a good choice for objectively determining $Z_{ABL}$ for a dataset similar to the DH2 and radiosonde datasets (high resolution profiles in the central Arctic environment). However, each method
still struggles in some situations. The primary downfall of the Heffter method is that it identifies $Z_{ABL}$ as the point where $\theta_v$ is 2 K warmer than $\theta_v$ at the bottom of the $\theta_v$ inversion. Failures noted in options 1-3 in Table 4 all occur when this criterion does not accurately identify the ABL top. The primary downfall of the TGRDM method, as noted in options 1-2 in Table 4, is that the strongest point of the $\theta_v$ inversion is not always at the ABL top. The TGRDM method also fails to find any $Z_{ABL}$ if there is no $\theta_v$ inversion strong enough to exceed the threshold necessary for $Z_{ABL}$
identification as laid out in Sect. 2.4.3. Lastly, the failure of the $Ri_b$ method occurs due to the difficulty of defining an accurate threshold value which correctly captures the likelihood of turbulence for all cases.

The last column in Table 4 lists the cases in which the objective $Z_{ABL}$ differs by more than 50% from the subjective $Z_{ABL}$ for the DH2 data, or no $Z_{ABL}$ was found, which can be referenced in the Supplementary Figures for all examples of the profile structures that are not as conducive to the success of the different objective methods.

**Table 4:** Summary of the features which lead to failure by each objective method, along with examples of DH2 cases that exemplify each failure, which can be found in the Supplementary Figures. The last column indicates the Supplementary Figures associated with cases in which the objective $Z_{ABL}$ was greater than 50% different than the subjective $Z_{ABL}$, or no objective $Z_{ABL}$ was found.

| Objective method | Features which lead to failure | Resulting failure | Examples | Cases with >50% difference in $Z_{ABL}$ |
|---|---|---|---|---|
| Liu-Liang | **1.** A weak $\theta_v$ inversion persists throughout the whole profile **a.** LLJ core altitude is well above the ABL top **b.** No LLJ | **1a.** Overestimation of $Z_{ABL}$ **1b.** No $Z_{ABL}$ found | **1a.** S6 on 24 March at 12:09 UTC **1b.** S33 on 30 April at 14:07 UTC | S6, S9, S10, S11, S13, S14, S17, S18, S19, S24, S29, S30, S31, S32, S33, S34, S35, S39, S41, S46, S48, S49, S52, S54, S55, S57, S58, S59, S60, S62, S64, S65, S66, S68 |
| | **2.** NBL capped by weak $\theta_v$ inversion | **2.** Overestimation of $Z_{ABL}$ | **2.** S54 on 17 July at 13:30 UTC | |
| Heffter | **1.** SBL height is not the altitude at which $\theta_v$ is 2 K warmer than $\theta_v$ at the surface **a.** SBL extends higher **b.** SBL does not extend as high | **1a.** Underestimation of $Z_{ABL}$ **1b.** Overestimation of $Z_{ABL}$ | **1a.** S5 on 23 March at 13:52 UTC **1b.** S42 on 21 June at 13:13 UTC | S4, S15, S16, S17, S25, S29, S32, S33, S34, S40, S41, S45, S47, S51, S52, S54, S55, S56, S58, S59, S66 |
| | **2.** NBL capped by weak $\theta_v$ inversion | **2.** Overestimation of $Z_{ABL}$ | **2.** S52 on 18 July at 13:10 UTC | |
| | **3.** Only shallow, weak $\theta_v$ inversion(s) | **3.** No $Z_{ABL}$ found | **3.** S40 on 6 May at 14:50 UTC | |
| TGRDM | **1.** $\theta_v$ inversion is strongest at the surface | **1.** Underestimation of $Z_{ABL}$ | **1.** S10 on 7 April (radiosonde profile) | S12, S13, S14, S24, S25, S29, S32, S45, |

| | 2. $\theta_v$ inversion is strongest within the entrainment zone | 2. Overestimation of $Z_{ABL}$ | 2. S64 on 22 July at 7:37 UTC | S46, S52, S54, S57, S58, S59, S60, S64, S66 |
|---|---|---|---|---|
| | 3. Only shallow, weak $\theta_v$ inversion(s) | 3. No $Z_{ABL}$ found | 3. S57 on 20 July at 11:28 UTC | |
| **$Ri_b$** | **1.** $Ri_b$ is not capturing transition from turbulent to laminar atmosphere **2.** Threshold value is not accurate | **1/2.** Over- or underestimation of $Z_{ABL}$ | **1/2.** S8 on 29 March at 12:24 UTC and S45 on 30 June at 8:39 UTC | $Ri_b(0.5)$: S8, S17, S18, S52, S57, S66 $Ri_b(0.75)$: S17, S52, S57, S66 |

Aside from what is listed in Table 4, the objective methods may produce results different than those found by the subjective method due to the consideration of different variables. Primarily, none of the objective methods directly consider the profiles of RH or mixing ratio (of course, humidity is indirectly considered through the virtual potential temperature profiles). Additionally, the Liu-Liang method for a CBL or NBL, as well as the Heffter and TGRDM methods, do not consider wind shear in the identification of $Z_{ABL}$.

When applying these objective methods to a large dataset to automatically identify $Z_{ABL}$, it is recommended that some level of pre-screening is applied to flag cases that contain the features or structural patterns summarized in Table 4 which can objectively be identified, that would make certain objective methods have difficulty identifying $Z_{ABL}$ (for example, one can screen for whether the $\theta_v$ persists throughout the entire profile or where the $\theta_v$ maximum occurs), and choosing which objective method to use based on that. While not all features in Table 4 may be possible to pre-
screen for, this list should at least help to identify some cases in which certain objective methods are likely to fail.

On the simplest level, one could choose which objective $Z_{ABL}$ detection method to use based on stability regime. Given the results in Fig. 6 and Supplementary Figures S70-S71, the best choice to use for SBLs might be the Heffter method (highest $R^2$ and higher frequency of cases within 10% of the subjective $Z_{ABL}$ when compared to NBL cases, from both the DH2 and radiosonde data) and the best choice to use for NBLs might be the $Ri_b$ method with either threshold value
(highest $R^2$'s from the DH2 data and higher frequency of cases within 10% of the subjective $Z_{ABL}$ when compared to SBL cases, from both the DH2 and radiosonde data). However, when separating out the efficacy of the objective methods depending on stability regime, the $Ri_b$ method has a combination of a high $R^2$ values and a high percentage of cases with objective $Z_{ABL}$ within 10% of the subjective $Z_{ABL}$ for both stability regimes, so this would be the best choice to apply to all profiles if one wanted to choose a single method, preferably with the threshold value of 0.5.

Overall, the objective methods are more likely to agree with each other as well as with the subjective $Z_{ABL}$ for cases with more simplistic structures, such as those with strong $\theta_v$ inversions with a base at or just below the top of the ABL, those with LLJ core altitude at or just above the top of the ABL, and those with consistently and somewhat gradually increasing $\theta_v$ with altitude above the entrainment zone.

**4. Summary and conclusions**

By comparing subjective $Z_{ABL}$ identified visually in $\theta_v$, humidity (both RH and mixing ratio), and $Ri_b$ profiles to objectively determined $Z_{ABL}$, the performance of several published methods (i.e., Liu-Liang, Heffter, TGRDM, and $Ri_b$) are evaluated across 65 DH2 UAS profiles. When comparing objective to subjective $Z_{ABL}$ for each DH2 case, the method that is most successful (combination of high $R^2$ value, low p-value, and slope close to 1.00) is the $Ri_b$ method with either threshold value of 0.5 or 0.75 (Fig. 5). When calculating the percent of DH2 cases in which the objective $Z_{ABL}$ is within certain relative difference ranges from the subjective $Z_{ABL}$, the $Ri_b$ method is also most successful (Fig. 7). The Heffter and TGRDM methods also produce reasonable results according to Fig. 5 and 7. The only objective method that largely fails at accurately identifying $Z_{ABL}$ is the Liu-Liang method.

In the process of applying these different objective methods to the DH2 data, some threshold values were modified to be better applicable to the UAS dataset. While these adjustments were made to best suit the 65 DH2 profiles analyzed in this study which occurred between March and July of 2020, these adjustments should yield better results for identifying $Z_{ABL}$ over sea ice during any season and location in the central Arctic. We hypothesize this because the ABL structures sampled by the DH2 in the current study were diverse and encompass the variety of ABL structures commonly observed in the central Arctic (which are typically shallow and either stable or neutral) throughout the entire year. Additionally, since the locations of the DH2 flights in this study range from deep in the Arctic pack ice to near the marginal ice zone, we are confident that the adjustments made will be applicable for identifying $Z_{ABL}$ in either environment.

Testing these adjustments outside of the 65 DH2 flights, the modified techniques were also applied to the radiosonde profiles closest in time to each DH2 flight, to determine if the methods work similarly on data from another sensing platform with different sampling methods. Radiosonde profiles closest in time proximity to the DH2 flights were used under the assumption that the ABL structure would change minimally between the launch of the two platforms (supported by Fig. 4), and thus applying the methods of subjective and objective $Z_{ABL}$ detection would lead to a similar conclusion. For the radiosonde data, the Heffter and $Ri_b$ methods prove most successful in terms of having a high $R^2$ value, low p-value, and slope closest to 1.00 when compared to the other objective methods (Fig. 5). Additionally, the $Ri_b$ method also proves most successful when looking at the percent of cases in which the objective $Z_{ABL}$ was within different relative difference ranges for the radiosondes, as it did for the DH2 (Fig. 7). Once again, the only method that consistently provided unfavorable results is the Liu-Liang method. These similar conclusions demonstrate that the adapted objective methods are indeed robust across platforms despite differences in sampling method, which suggest that one can take the methods and apply them to UAS, radiosonde, or other profile data alike, without having to tweak them.

These findings show that no single method works well 100% of the time. Given this, the best way to accurately identify $Z_{ABL}$ across a variety of conditions in the Arctic atmosphere is to visually analyze the $\theta_v$, humidity, and $Ri_b$ profiles for each case individually. However, as subjective identification is time consuming and requires expert knowledge of the physical processes that dictate ABL structure, then in the case of large datasets that require automated processing

techniques, the current study reveals that the $Ri_b$, Heffter, or TGRDM methods are most suitable for such a task, with the preferred method being the $Ri_b$ method with threshold value of 0.5. For data with vertical resolution of 10 m or coarser, the Heffter method is no longer recommended. The Liu-Liang method does not provide consistent results in accurately identifying Arctic $Z_{ABL}$ in many cases, especially for SBLs (Fig. S70). The most common occurrence of failure of the objective methods exists for NBLs capped by a weak $\theta_v$ inversion, so that a clear $\theta_v$ slope change between

the ABL and entrainment zone is difficult for automated methods to find. In such cases, the $Ri_b$ method was found to be most reliable for identifying $Z_{ABL}$. A full list of features which cause each objective method to fail is provided in Table 4 above. The objective methods may also fail if the near-surface atmosphere is not well sampled, for example in the case of the radiosonde data; if ABL stability is defined by what is happening near the surface (e.g., a shallow convective layer), then this is missed by radiosonde profiles which only begin 23 m or higher, and stability regime

could be incorrectly diagnosed. This highlights the value of platforms which can sample the near-surface atmosphere, such as the DH2. To accommodate the above problems, a semi-automatic approach may be beneficial in which one would apply all the recommended objective methods, and visually inspect only the profiles for which the resulting $Z_{ABL}$ diverges greatly.

The methods and results of this study for stability regime and $Z_{ABL}$ identification are currently being applied to the

entire year of radiosonde data collected during the MOSAiC expedition (October 2019 – September 2020) to create a data product containing year-long statistics on ABL characteristics in the central Arctic. Additional metrics, such as LLJ height and speed, and temperature inversion layer depth and strength will be included in this product for eventual publication. Value from the DH2 data and methods used in the current study comes from the uniqueness of the location and timing of the profiles collected. Therefore, these data provide a unique opportunity to evaluate any additional $Z_{ABL}$

detection schemes that were not addressed in this study, or that have yet to be developed, as well as can be used to learn about the intricacies of additional structural components of the Arctic atmosphere such as the entrainment zone. Lastly, we are working to derive turbulence parameters from the DH2 fine wire measurements which will enhance the value of the DH2 data in ABL studies.

**Data availability**

All DataHawk2 data used in this study are openly available from the National Science Foundation Arctic Data Center at https://doi.org/10.18739/A2KH0F08V (Jozef et al., 2021) as described in de Boer et al. (submitted). The radiosonde data are available at the PANGAEA Data Publisher at https://doi.org/10.1594/PANGAEA.928656 (Maturilli et al., 2021). These data are subject to the MOSAiC Data Policy (Immerz et al., 2019) and will be openly available after 1 January 2023.

**Author contributions**

GdB and JC planned the DH2 data collection and acquired funding; GJ and JC conducted DH2 flights; SD provided the radiosonde data; GJ, JC, and GdB conceptualized the analysis presented in this paper; GJ analyzed the data; GJ wrote the manuscript; JC, GdB, and SD reviewed and edited the manuscript.

**Competing interests**

The authors declare that they have no conflict of interest.

**Acknowledgments**

Data used in this paper were produced as part of RV *Polarstern* cruise AWI_PS122 and of the international Multidisciplinary drifting Observatory for the Study of the Arctic Climate (MOSAiC) with the tag MOSAiC20192020. We thank all those who contributed to MOSAiC and made this endeavor possible (Nixdorf et al., 2021). Radiosonde

data were obtained through a partnership between the leading Alfred Wegener Institute (AWI), the Atmospheric Radiation Measurement (ARM) User Facility, a US Department of Energy facility managed by the Biological and Environmental Research Program, and the German Weather Service (DWD). Non-author contributors to DH2 design, data collection, and data processing include Dale Lawrence[1], Jonathan Hamilton[1,2,4], Radiance Calmer[2,3], Brian Argrow[1,5], Steven Borenstein[1,5], Abhiram Doddi[1], Julia Schmale[6], and Andreas Preußer[7].

[1]Dept. of Aerospace Engineering Sciences, University of Colorado Boulder

[2]Cooperative Institute for Research in Environmental Sciences, University of Colorado Boulder

[3]National Snow and Ice Data Center, University of Colorado Boulder

[4]NOAA Physical Sciences Laboratory

[5]Integrated Remote and In-Situ Sensing, University of Colorado Boulder

[6]Swiss Federal Institute of Technology Lausanne

[7]University of Trier

**Financial support**

Collection and analysis of atmospheric boundary layer data with the DataHawk2 was funded by the National Science Foundation (award OPP 1805569, de Boer, PI). Additional funding and support were provided by the Cooperative

Institute for Research in Environmental Sciences, the National Oceanic and Atmospheric Administration Physical Sciences Laboratory, and the Alfred Wegener Institute.

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
