# Peer review of "Testing the efficacy of atmospheric boundary layer height detection algorithms using uncrewed aircraft system data from MOSAiC"

_Atmospheric Measurement Techniques, 2021_

## Author Response (AR1)

**Response to anonymous referee comments**

The authors thank the three anonymous referees for taking the time to review our manuscript and for their helpful comments, which have improved the manuscript. Each referee comment is given below in **bold italics** followed by our response to the comment. The line numbers provided in our responses refer to line numbers in the revised manuscript.

**Anonymous referee #1**

***A novel dataset of detailed atmospheric profiles gathered by a UAS in the Arctic region is being explored to determine how to derive the height of the atmospheric boundary layer using automatic methods. "Subjective", visual height detection is used as a reference standard to evaluate a number of "objective" methods available in the literature. While the presented work is promising, highlighting the difficulties of accurate layer height detection in the Arctic region, the manuscript requires major revisions to better demonstrate the research results.***

***Overall, the text is very descriptive and could be made more concise in many sections. The actual research findings need to be pointed out more clearly and should be placed into context to the literature. It is important to highlight how the manuscript provides novel insights and methodological advances. This could be partly improved by removing the severe imbalance between the methods description (Section 2 has 390 lines + 8 figures) compared to the results section (section3 has 150 lines + 2 figures). The focus of the presented figures should be placed on the research findings rather than the introduction of the methods that are in most parts explained in the literature, i.e. the current manuscript does not present a novel method.***

Thank you for your comments. The authors recognize and agree with your comment that there was an imbalance between the length of the methods and results sections. To improve this, we now better highlight how the manuscript provides novel insights and methodological advances, while spending less space on discussing aspects of previously published methods that were unchanged. When describing the subjective methods, we now include the details in a table (Table 3 of the revised manuscript beginning on line 291), which condenses the information and makes it more readily available. When describing the objective methods, we now provide just a brief description of each (lines 315-366) and leave it to the reader to review the original citation for more details about these methods, such as the guiding equations. We also moved Figures 3-6 from the original manuscript, which provided examples of the application of each of the previously published method, to the Supplementary Figures document (they are now Supplementary Figures S1-S4). We have also revised Figure 7 from the original manuscript (which is now Figure 3 of the revised manuscript, on line 403) to show two examples (a SBL and NBL) which demonstrate the ABL heights determined by all of the subjective and objective methods used in this manuscript.

The changes described above have greatly reduced the length of the methods section and thus better balance the methods to results section lengths. We have also added some material to the results section, most notable that we now briefly discuss the results of the scatter and bar plots in the context of each stability regime, and provide similar plots with stable and neutral cases separated out in the Supplementary Figures. We also discuss the sensitivity of the efficacy of the

objective methods to the vertical resolution of the data. We hope that these changes address your concerns.

***Minor comments:***

***At times, a more precise wording could help make the text less descriptive. E.g. use established terms such as "vertical gradient" instead of "change with height".***

The authors have changed the text throughout the manuscript to use the term "vertical gradient" when referring to change with altitude (for example, line 229-230, 238, 260, etc.). Other changes made along these lines include making sure to use active rather than passive voice, changing 'ABL height' to '$Z_{ABL}$' (introduced on line 81), changing 'Rib method with critical value of 0.5' to 'Rib(0.5)' (introduced on line 365), changing phrases like 'ABL height of an SBL' to 'SBL height' (for example, line 327), and specifying local maximum and minimum when these terms are relevant (for example, line 351). If there is additional wording you were intending for us to change, please specify.

***Section2: Maybe the criteria for the subjective height detection could be summarised in a table? What is the expected uncertainty in these methods based on visual assessment of at times very subtle signatures in the profiles?***

We now summarize the subjective ABL height detection criteria in a table (beginning on line 291), rather than writing it out in paragraph form. The primary uncertainty is not due to the exact height of the kinks, as the uncertainty here is subject only to the vertical averaging procedure and response time of the sensors, and thus on the order of ~1 m. Instead, primary uncertainty is due to whether or not the feature we are using to define ABL height is representing the top of the ABL. To address this, we comment in lines 298-304 that 90% of cases had a fairly clear ABL depth, and only 10% had ambiguous depths (for example if the height of the theta_v and humidity kinks that could both be interpreted as ABL height are at a different altitude). In these cases, we state that depending on which kink was chosen, ABL height could vary by 10-30 m.

***Line 54: Please provide a short explanation on the concept of "radiative mixing forced by cloud cover".***

We have changed the wording to "turbulent mixing below cloud base due to cloud top radiative cooling," which can be found beginning on line 55. We hope this is a sufficient explanation to eliminate confusion.

***Line 64: maybe reword. The literature on ABL height detection is obviously very extensive so it would be good to clearly state that Table 1 lists a few examples of relevant publications and atmospheric variable.***

The authors now clarify that we list only some of the atmospheric variables used for ABL height identification (line 86), and only some of the publications which reference them (line 87). We also clarify that we list only thermodynamic and kinematic processes, as these are the focus of the paper, and are what is available from the DH2 data.

*Line 71: which humidity variable is analysed here?*

The authors now clarify in parentheses that we use relative and absolute humidity (line 98).

*Line 73: You state the entrainment zone is located "above" the ABL. Maybe a few words on the relation between ABL height and entrainment zone characteristics would be useful.*

The authors add a few words on lines 69-70 describing that the entrainment zone is a stable layer between the ABL and free atmosphere, but we were unsure what relationship between ABL height and entrainment zone characteristics you are referring to, so we were unable to fully address this comment. However, the DH2 data would be a great resource to conduct such a study in the future, so we mention this when we discuss future work in the conclusion (line 660-661).

*Line 132: what is meant by "assess the ice alongside the Polarstern"*

The wording in the manuscript is "*access* the ice alongside the Polarstern," rather than "*assess*." We assume the confusion comes from misreading this sentence. This text refers to the ability of scientists onboard the Polarstern to be able to exit the ship and go onto the sea ice.

*Line 259: the term "mixed layer" has not yet been mentioned before. Explain why you are using it now for NBL?*

To avoid confusion with using different terms, the authors change "mixed layer" to "well-mixed ABL" (line 341) which is characteristic of an NBL or CBL.

*Line 262-267: These sentences are very descriptive. Please condense the key information and try to generalise.*

The authors have removed this paragraph, as we now present this information in a table (line 291) rather than paragraphs, based on your earlier comment. Thus, the information is now presented in a more generalized manor, with only key information provided.

*Line 270: again, try to be less descriptive. E.g. the term "increase with altitude" could be replaced by "vertical gradient"*

While this specific sentence no longer exists due to the presentation of this information in table form, throughout the text we now say "vertical gradient" instead of "increase with altitude." This is exemplified in Table 3 of the revised manuscript beginning on line 291.

*Line 278: "… extends from the surface to …".*

Due to presentation of the subjective methods in table form, we removed this sentence, and this comment is no longer applicable.

*Line 284: replace by "change in vertical gradient"*

Due to presentation of the subjective methods in table form, we removed this sentence. Throughout the table, we still use the word 'kink' to describe a change in vertical gradient for the sake of conciseness, but in the paragraph before Table 3, we specify that when we say 'kink' we refer to dramatic shift in slope, i.e., drastic change in vertical gradient (line 288).

**Line 340: what about the methods mentioned e.g. by Collaud Coen et al. (2014)?**

Are you referring to the sentence beginning on line 341 of the original manuscript: "There is no recognized equation to determine SBL top height accurately without observations supporting the derivation of turbulent kinetic energy profiles"? We have moved discussion of the TKE method to the paragraph after Table 1 (line 106-108), where we mention that though TKE is perhaps the most valuable profile for ABL height identification, these data are not available to aid in the current study. We also now include methods used in Collaud Coen et al. (2014) in Table 1. However, the methods for SBL height identification discussed in Collaud Coen et al. (2014), including identifying ABL at the top of the temperature inversion or where dtheta/dz = 0, don't perform well for an Arctic atmosphere where a weak temperature or theta_v inversion can often extend deep into the profile, well above the ABL. We have added a sentence to the paragraph below Table 1 (lines 104-106) where we discuss this as an explanation as to why we do not use these methods in the current study.

**Line 452: Explain how the data acquisition platform (radiosonde vs UAS) or the geographic location (mid-latitude vs arctic) are expected to influence the performance of the detection methods and hence warrant the outlined adaptations.**

We have added some text on lines 372-375 to discuss these concerns. For the data acquisition platform, we explain that the difference is that the DH2 samples at a higher resolution. For the geographic location, we explain how the Arctic ABL structure is often quite different than that observed in the mid-latitudes, primarily due to the lack of daytime convection or a diurnal cycle most of the time in the Arctic, which means there are no deep unstable layers, or residual nighttime layers above the SBL in the Arctic. In addition, the Arctic ABL is also often much shallower than in the mid-latitudes and this required some adjustment of fixed height criteria in previously published objective methods. This information has also been added to the introduction in lines 68-73 where we discuss the inherent difference between the Arctic ABL and that in mid-latitudes.

**Line 462: what causes this warm bias in the lowest levels?**

This is due to the *Polarstern* acting as a heat source. A explanation of this has been added to the text on line 389.

**Line 502-511: Shorten introduction for interpretation of linear regression. It can be assumed that the reader of this scientific publication is familiar with this common approach.**

This explanation has been shortened. Now, we simply say "The $R^2$ value demonstrates how much of the variation in objective $Z_{ABL}$ can be explained by the difference in subjective $Z_{ABL}$" (line 442-443).

*Figure 9: list number of samples.*

The number of samples for each scatter plot has been added to the caption of Figure 9 from the original manuscript (Figure 5 of the revised manuscript). This can be found in lines 484-490.

*Line 550: Careful with such statements. Very few samples with DH2 results above 150m*

The authors have added a statement to suggest that this increased variation for shallower ABL height may be due to the greater number of samples of low ABL (lines 480-481).

*Lines 512-552: So what is the interpretation of these results? How do they compare to the expectations in context of the literature?*

The overall interpretation of these results is discussed after Figure 10 of the original manuscript (Figure 6 of the revised manuscript), but we have provided some additional context to this, including how the results compare to the literature. Primarily, we point out that previous literature also favors the Rib method when mechanically-produced turbulence dominates, as is true in the central Arctic (lines 539-541).

*Lines 599: discuss relation of LLJ and ABL in the study area. How is it assessed when the LLJ is located above the ABL? How does this relate to the expectations and literature?*

We now clarify that the altitude of the LLJ core is determined through the application of the SBL Liu-Liang method, which is usually located above the subjective ABL top (line 567-568). We also add a comment that this agrees with what has been found in previous literature and include some citations. Several of these citations specifically state that the LLJ core is typically at or above the ABL top (line 568), and one of these citations states that using LLJ core height to define SBL top produces results inconsistent with those from other methods (line 568-570). Thus, our results are in agreement with previous literature that using the LLJ core height to determine ABL top is often inappropriate.

Anonymous referee #2

*The manuscript "Testing the efficacy of atmospheric boundary layer height detection algorithms using uncrewed aircraft system data from MOSAiC", by Gina Jozef, John Cassano, Sandro Dahlke, and Gijs de Boer, evaluates different published methods for the determination of the ABL height for a unique data set sampled by an uncrewed aircraft system over the sea ice of the Arctic Ocean. These "objective" methods are verified against a "subjective" or visual method to evaluate the height of the ABL, and the robustness of this approach is shown by applying it also to radiosonde profiles sampled in close spatiotemporal proximity. The manuscript is well within the scope of AMT but requires major revisions before it can be accepted for publication.*

*Here my major comments:*

*The information given in the introduction on the concept of an atmospheric boundary layer height is very superficial. Here the authors should expand and clarify why this concept is important, for which applications it is used, and also in which way its definition is debatable. In my eyes, it is a diagnostic parameter used to quantify the altitude up to which the direct surface-atmosphere interaction can be considered relevant. It is some sort of simplification, which is helpful for many applications, but it is not a physical property of the atmosphere, e.g. it may not be continuous for example during regime transitions. A more critical reflection on this should be included. It should also be clear that either method for identifying the ABL height is only providing an estimate (one could even claim that there is no true ABL height only different methods to estimate or diagnose it).*

Thank you for your comments. The authors have added a paragraph to the introduction which explains in more detail why ABL height identification, especially in the Arctic, is important (lines 74-84). It has also been clarified that ABL height is an approximation, which is why the work presented in the paper – to find the best approximation – is important and useful (lines 82-84). We have also added some text discussing why the definition of ABL height is debatable (lines 94-95 and 113-114).

*This also has implications for your line of argumentation, which seems to be based on the assumption that your "subjective method" is giving the true ABL height. I agree that a visual evaluation of the ABL height by an expert may be generally better than any objective method, but also the expert can be wrong, e.g. due to misleading observations (DH2 and RS provide only a "quasi-snapshot").*

The authors now clarify throughout the text that the 'subjective' ABL height is not necessarily the 'true' ABL height, but rather, the best estimate of ABL height given the available data (lines 113-114, lines 220-224).

*More details on the calculation of the Rib number and the determination of underlying parameters (gradients) are required. In particular, I lack details on the determination of the wind speed from the helical flight patterns. There is a reference to unpublished work, but some important details should be mentioned here. I assume you use the 10Hz 3D wind data? How do potential time lags and/or inaccuracies in IMU or GNSS data influence instantaneous wind measurements?*

The authors have added a section to the paper before the section on determining stability regime called 'Preparing the DataHawk2 data for analysis' (beginning on line 187), which gives more detail on the calculation of Rib and gradients. For the winds specifically, a more thorough description of the method used to diagnose winds from the UAS data is currently in review in de Boer et al. (submitted 2022) and we cite this manuscript (line 155), although a brief description of the wind estimation is now provided in the revised manuscript (lines 148-152). A summary of the wind processing can also be found in the metadata for the DH2 data used in this study, which is now cited (line 154). In addition, another paper describing the DataHawk2 platform and use of its data (this paper is more general, not specific to MOSAiC) will imminently be submitted for publication to AMT by Hamilton et al, so the authors now include the Hamilton et al. citation as well (line 155). Though it is unfortunate that neither of these papers are published yet, the

authors are hopeful they may be at preprint stage once the final version of this manuscript is submitted.

*I also noted that the DH2 temperature profiles start at lower altitudes than the corresponding wind speed profiles. Why is this?*

This is because near-surface wind speed values from the DH2 are unreliable, due to the manual, rather than autopilot flight, during take-off and landing, which conflict with the measurements and calculations of wind speed. Therefore, we don't consider wind speeds below 30 m to be reliable. The authors have added text to the manuscript to specify this (lines 156-158).

*In which coordinate system is the wind speed measured, relative to sea ice or in earth coordinates? I think this has only implications for the Richardson number at the first level, but this is still important to mention.*

All variables are measured in the same coordinate system, which is Earth-relative (now written on line 144). We recognize thus that the wind speed in the Earth-relative coordinate system does not represent the actual wind shear between the atmosphere and sea ice that is also moving, and that the correct shear is that between the true wind and the speed and direction of the sea ice movement. However, we don't believe this to be an issue for the calculation of Rib at the lowest level because the ice was moving slowly relative to the wind speed. Krumpen et al. (2021) at https://doi.org/10.5194/tc-15-3897-2021 found that the average drift speed of the ice during MOSAiC was less than 0.1 m/s, which is very small compared to the wind speeds observed. Therefore, an assumption of 0 wind speed at the surface is sufficient for the calculation of Rib at the first level. Above this, we are comparing two wind speed measurements, so the drift of the ice is no longer relevant. The authors have added a sentence to the manuscript to explain that the drift of the ice is not crucial to consider (lines 215-217).

*Furthermore, the response time of the temperature and humidity sensors and their implications for the computation of Ri and theta_v and determination of the ABL height should be discussed. For theta_v two very different response times are combined (T and RH sensor). What effect does bin-averaging, including ascent and descent data have?*

Differences in response times of the RSS421 temperature and RH sensors has a negligible impact on the calculation of theta_v because the moisture content in the Arctic atmosphere is very low. In the coldest temperatures sampled, theta and theta_v values typically differed on the order of ~0.1 K, and in the warmest temperatures sampled, theta and theta_v values typically differed on the order of ~1 K. Regardless, the addition of humidity does not change the structure and location of features in the theta_v or Rib profiles, which is what is important for ABL height identification. We have included a figure below which shows an example of a cold and warm case, to demonstrate how little the potential temperate and Rib profiles differ when theta versus theta_v is used. We have also added text to the manuscript to discuss this (lines 189-193). To address your question about the effect of bin averaging through ascent and descent, we have added some text to the manuscript which describes how we average the theta_v, humidity, and wind speed variables over 1 m altitude bins throughout the entire flight to further eliminate the effects of differences in sensor response times during ascent and descent (lines 193-199).

[Figure]

**Figure 1:** Two example flights, one during the coldest period sampled by the DH2 **(a)** and one during the warmest period sampled by the DH2 **(b)**, showing the $\theta$ and $\theta_v$ profiles (left panel), as well as $Ri_b$ profiles calculated from $\theta$ and $\theta_v$ (right panel) where profiles using $\theta$ are in solid red and profiles using $\theta_v$ are in dashed blue.

*How may non-stationarity, e.g. a substantial temperature change near the surface within the ~30m flight time affect the results, and are there any observations indicating that this may have been an issue (One could simply make use of surface-based observations to detect non-stationary conditions during the flight period)?*

Visually comparing ascent and descent profiles of the UAS flights allowed us to see that there was never substantial change in the near-surface values of variables over the ~30 min flights. However, averaging in 1 m bins throughout the flight would mitigate any impacts this would have on the theta_v or Rib profiles. The authors have added a sentence on lines 195-197 which states this.

*I am missing a general assessment of the different sampling methods (radiosonde vs DH2). There are some important differences like the radiosonde can only sample during ascent, differences in the vertical climb speeds, response time, wind measurements. It' is therefore well possible that one of the data sets is generally more smooth or has higher uncertainties in particular in a specific altitude region or under certain conditions.*

The authors recognize that the sampling methods of the DH2 and radiosonde are different. However, both platforms still observed atmospheric features similarly, which can be seen in the comparisons of DH2 and radiosonde profiles in Figure 7 of the original manuscript (Figure 3 of the revised manuscript) and the Supplementary Figures showing every flight. Some paired DH2 and radiosonde profiles agree better than others (they agree well in the revised Supplementary

Figure S23 but poorly in S40), but the ones that don't agree as well are cases with greater time between DH2 and radiosonde launch in which atmospheric conditions may have changed. Nevertheless, the point of this paper is not to discuss the impacts of different sampling methods on the efficacy of the methods. The purpose of this work was to improve ABL height identification methods using the high resolution DH2 data, discover which one works best in the Arctic regime, and then to test whether these adapted methods work well on another platform (i.e., the radiosonde), despite differences in sampling methods. The results show that they do work with similar efficacy, which argues that one can take these methods and apply them to UAS or radiosonde data alike, without having to tweak them. The authors now try to make it clearer the purpose of including radiosonde profiles (lines 375-378, 623-625), and discuss some of the differences in sampling methods (lines 541-548), with a focus on how these differences don't significantly change the efficacy of the objective ABL height detection methods (lines 548-550, 632-635). The authors have also added a few sentences to explain that the radiosonde data have some smoothing applied to account for the swinging of the pendulum after launch, as well as how this smoothing might affect the choice of critical bulk Richardson number (lines 521-526).

*The results/discussion section is kept on a rather general level and has little content compared to the methods chapter. Interesting research questions are not addressed (systematically), e.g.:*

- *How do different sampling and data processing methods affect the differences for radiosonde vs DH2 based ABL height estimates?*

As described in our response to the previous comment, the authors have added some text to describe the differences in sampling methods between the DH2 and radiosonde, but despite this, both datasets produce similar results (lines 541-550, 632-635), and thus the methods used in this paper are applicable to both platforms. Lastly, we also address how differences in processing methods (smoothing or vertical resolution) affect the efficacy of different objective methods (lines 521-526 and 551-561).

*In Sect 3.2 you address the question of how stability or specific features in the ABL structure cause certain methods to perform better or worse than others (you also scratch on the surface of this in Sect. 2.2 ff) but this is done in a rather episodic manner. Table 3 would be a good starting point for expanding on this. In the corresponding section, you mention that one could list problematic features to be used in a pre-screening procedure. I think you should have the data and knowledge to propose such a list. I consider this as very relevant for the research community using similar systems to determine the ABL height.*

Referee #3 suggested to use Figures 9 and 10 of the original manuscript (Figures 5 and 6 of the revised manuscript) to differentiate between the efficacy of the different methods based on stability regime. We have addressed this by creating identical figures which are shown in the Supplementary Figures document, which contain only SBL datapoints (S70 and S72) and only NBL datapoint (S71 and S73). We discuss the main takeaways of these figures in section 3.1 (lines 491-496 and 531-536) by highlighting which methods have the highest efficacy for each stability regime. In section 3.2, we recommend which objective method(s) to use if one were to choose a different method based on stability regime (lines 592-600). For listing problematic features to be used in a pre-screening procedure, we now list these in Table 3 of the original

manuscript (Table 4 in the revised manuscript, beginning line 584) and have trimmed the text throughout section 3.2 to not be redundant. We hope that with presenting this information in a table in list form will make it easier for a reader to flag these features in a pre-screening procedure.

- *How sensitive are the different methods to the choice data processing methods, or sensors used, e.g. vertical averaging procedure? This would also be of interest to the community, e.g., to make adjustments for different measurement systems.*

To address this, we have re-run our procedures with DH2 data averaged in 5 m, 10 m, and 20 m bins instead of the original 1 m bins, to determine if the methods and results are sensitive to the vertical averaging procedure. We have found that there is no significant difference in the efficacy of objective methods using the coarser data, aside from the Liu-Liang method, and the Heffter method at 10 and 20 m resolution. This suggests that the Liu-Liang and Heffter methods are more sensitive to vertical resolution, but the Rib and TGRDM methods are not. We have added a paragraph at the end of section 3.1 summarizing this (lines 551-561). We also include scatter plots identical to those presented in Figure 9 of the original manuscript (Figure 5 of the revised manuscript) below, which show the results of the 20 m resolution data.

[Figure]

**Figure 2:** Figure identical to Figure 7 in the original manuscript (Figure 5 in the revised manuscript) using DataHawk2 data averaged in 20 m vertical bins to simulate data with 20 m vertical resolution.

*Minor comments/suggestions*

*Since this is a technical paper small details may need some additional attention, e.g. the difference between level and layer (which has a thickness) and the corresponding indexing. Are meteorological parameters, gradients at level k averaged over k to k+1, and how exactly are averages and gradients determined?*

The authors have taken care to make sure the uses of level and layer are correct. Values at altitude k are averaged between k-0.5 m to k+0.5 m. This information, along with more detail on how averaged and gradients are calculated was added to a new section titled "Preparing the DataHawk2 data for analysis" which appear before the determination of stability regime (lines 187-224).

*I also regard the term "subjective" as misleading. In Section 2.3.1-3 you describe criteria that could also be automated (you may have done that). I would therefore claim it is a semi-objective method, where the final decision is made through a visual interpretation by an expert. "Visual" and "automatic" may be the better terms to distinguish the two types of methods. When reading the abstract my first impression was that it is strange to evaluate an "objective" method with a "subjective" one. Generally speaking, one may want to trust an objective method more than a subjective one. This confusion could be avoided by sticking to the terms "visual" and "automatic".*

The authors prefer not to change the wording throughout the paper from 'subjective' and 'objective' to 'visual' and 'automatic', because the inherent dichotomy in the words 'subjective' and 'objective' give these two a clear distinction, which we would like to keep. However, we have adjusted the wording to make it clear throughout the paper, including in the abstract (lines 19-24), that the 'subjective' method refers to a visual/manual method, while the 'objective' methods refer to automated method (lines 111-112 and 220-224). The authors also disagree that the subjective methods can be automated. The strength of the subjective method is that an expert assesses all the features of the profile to identify the subjective ABL height. Here, the knowledge of the expert is critical and can't be automated. We have added some text to the beginning of section 2.4 which highlights this (lines 306-308).

*In general, I also see the potential for condensing the content of in particular the introduction and methods sections.*

We have drastically reduced the amount of content in the methods section by listing the subjective criteria in a table rather than paragraphs (Table 3 of the revised manuscript, beginning on line 291) and by removing a lot of the information provided about each objective method, including specific equations, and instead pointing the reader to the original citation for additional information. We also have moved Figures 3-6 of the original manuscript to the Supplementary Figures document. We have also done our best to condense the parts of the introduction that were present in the original manuscript, however comments from you and other referees have asked for additional information to be added to the introduction (mainly including more background information on the typical Arctic ABL structure, and discussing why knowing the ABL height is important). Therefore, the length of the introduction has not overall been reduced.

*Readability could be increased by avoiding some passive phrases and using "we", as done elsewhere.*

The authors have worked to change passive phrases to using "we" instead.

*Specific comments, technical corrections, and suggestions for improvement:*

*L15: "fixed-wing uncrewed aircraft system"*

This change has been made (lines 15-16 and 143).

*L16: consider introducing an abbreviation for the ABL height, e.g. simply "Z" or "H" or with subscript "ABL"*

The authors now use '$Z_{ABL}$' throughout the paper instead of 'ABL height.' This is introduced on line 81.

*L16: "the ABL structure".*

This change has been made (line 17).

*L18: "the ABL height". In general, I have the feeling that some articles are missing, in particular when abbreviations are used. Please add wherever this could increase the readability. Note that if you would simply use "H" instead of "ABL height" you would not need the article in this case.*

We have changed 'ABL height' to '$Z_{ABL}$' throughout the paper, so most problems like this are no longer an issue. However, we have also double checked that the articles are correct in each case.

*L39: "pack ice" might be better.*

This change has been made (line 40).

*L49-60: This paragraph could be rewritten, giving a general description of the ABL and its typical structures of the ABL and above. This would then naturally lead to the concept of the ABL height.*

An additional paragraph has been added (lines 61-73) which gives a general description of the Arctic ABL and its typical structure.

*L51ff: "...the ABL is mostly impacted by interactions between the atmosphere and sea ice surface features, including the generation of turbulence through surface energy fluxes emitted from open water regions such as leads…". This is hard to understand. I think you want to mention both, mechanical and buoyant production of turbulence, but it reads like buoyancy is part of the mechanical production (by interaction with surface features). It should also be made clear that buoyancy is mostly negative over sea ice.*

We now specify that the ABL is impacted by interactions between the atmosphere and sea ice pack features, which includes the open water and lead components as well. Now, we first introduce that the Arctic ABL is impacted by both buoyant and mechanical turbulence. Then, we provide some examples of each. This can be found in lines 50-58. The authors disagree that buoyant turbulence is only present over open water features. For example, buoyant turbulence can also occur if colder air is advected over warmer sea ice.

*L54: I consider mixing as a turbulent process so "radiative mixing" could be misleading. How about "turbulent mixing forced/triggered by radiation–cloud interaction"...*

This part of the sentence has been changed to accommodate your comment as well as that of another referee to now specify that we refer to "turbulent mixing below cloud base due to cloud top radiative cooling" (line 55).

*L55: Consider mentioning the effect of ice edges e.g., at leads. The roughness is increased due to the freeboard.*

This is now mentioned (line 57).

*L59: include "(LLJ)" here.*

The authors prefer to introduce the abbreviation LLJ farther down on line 77, when we discuss the important features with which the ABL interacts. We fear that if we introduce the abbreviation LLJ where you suggest, it may get lost in all the citations, and it is more easy to notice when introduced on line 77.

*L76: only one ")"*

Two ")" are necessary since this is the end of a citation within a statement in parentheses.

*L76: Here one could expand a bit on where the ABL height could be identified be when there is a capping LLJ? At the core, somewhere below or above? Are there different opinions about this?*

When a capping LLJ is used to identify ABL height (as is done in one of the Liu-Liang SBL methods), the ABL height is identified to be the height of the LLJ core. The authors now specify this more clearly (line 103-104). The authors now also specify in lines 567-568 that according to our observations and to the literature, the ABL is typically at or below the LLJ core. We also cite literature which agrees that identifying the ABL height as the LLJ core height is often not accurate (lines 568-570).

*L114: Is the SHT-85 really measuring at 100Hz. The response time of this sensor is rather slow, but of course, it is allowed to oversample.*

Yes, the SHT is really sampling at 100 Hz. We include this information in section 2.1 for the sake of completeness in describing the instrumentation carried by the DH2. However, as you

point out, the response time of the SHT is quite slow, so this is why we don't use it for the current analysis, but instead use data from the RSS421.

*L117: Here more details on the 3D wind estimates could be given (see comment above).*

Per your above comments, more information has been added about 3D wind estimates (lines 148-152). We have also cited more sources where additional information can be found (lines 154-155).

*L110-L120: Can you also provide similar information (doesn't have to be as detailed) for the radiosonde sensors (here or elsewhere). Does the radiosonde also contain a Vaisala sensor (I think the corresponding radiosonde sensor would be RS41)? Can the radiosonde sensor also be included in Table 2?*

We have added a sentence when we introduce applying the objective methods to the radiosonde profile which indicates that the radiosonde sensor (RS41-SGP) pressure, temperature, and humidity variables have the same resolution, repeatability, and response time as those for the DH2's RSS421 variables, as listed in Table 2. We also add the uncertainty and resolution in the wind speed and direction of the RS41-SGP (lines 382-384).

*L147ff: Were there any cases when a clear determination of the boundary layer height was not possible, even though the max altitude was sufficient. Potential reasons could be non-stationarity or internal boundary layers?*

We were always able to determine the ABL height when the max altitude was sufficient. Sometimes this was difficult, but this is why we consider several variables (theta_v, humidity, Rib) when determining ABL height. Considering all this information, we could always determine the ABL height with reasonable confidence.

*L152-155: Does this mean data from both, the ascent and descent were used? If so you may cause a kink at the level where you start using the first data after takeoff but have data from the descend before landing.*

The authors are aware of this potential influence, but did not notice this to cause in issue in creating false kinks. Nonetheless, we were conscious of this when looking for kinks in the profiles for ABL height identification.

*L174, 175, 177: Should be "an SBL/NBL" but "a CBL". Check the entire manuscript for this type of typo.*

This typo has been fixed throughout the manuscript.

*L176: You mean that the SBL "can range from …" but what is written refers to theta_v.*

The structure of this sentence has been changed and moved to the introduction to accommodate other referee comments, so is no longer a problem (lines 61-63).

*L183: It is not clear what "i" refers to.*

The authors specify now that "i" refers to the initial altitude of the DH2 referenced in the below equations (lines 232-233).

*L208-209: Your data may even suggest that there is a tendency to more stable conditions during the seasons you observed.*

Since we had an almost equal number of SBL and NBL cases during the seasons we observed (actually 1 more NBL than SBL), this is not true (lines 248-249). However, we have removed this statement in the process of condensing the methods section.

*L227: "The Bulk Richardson number"*

We have changed 'bulk Richardson number' to 'Rib' in this case, so 'the' is no longer needed (line 272).

*L227-228: Consider rephrasing the first sentence to allow for buoyant suppression. "buoyancy" may be better than "buoyant production"*

We have changed this sentence to also mention buoyant suppression from static stability (lines 272-274).

*L240ff: Here you should be more precise. See for example https://glossary.ametsoc.org/wiki/Bulk_richardson_number: "In the limit of layer thickness becoming small, the bulk Richardson number approaches the gradient Richardson number, for which a critical Richardson number is roughly Ric = 0.25 … Unfortunately, a critical value is not well defined for the bulk Richardson number, leading to uncertainty in turbulence likelihood for values near the critical value."*

The authors have adjusted the wording in this paragraph to summarize the information provided by the AMS glossary (which we now cite), which highlights that a critical value of 0.25 is not necessarily always the exact number that corresponds to the transition between turbulent and laminar conditions, but low Rib is generally expected in the ABL, and high Rib is expected above the ABL (lines 274-283).

*L247: Here more detailed information would be necessary. The raw data is bin averaged using 1-m bins,. then you use 30-m bins for Rib with 5-m resolution. Note that the choice of bin size may have implications for the choice of thresholds, e.g., for Ri_b. This could be discussed further.*

The authors have added a new section titled "Preparing the DataHawk2 data for analysis" (beginning on line 187) before "Determining stability regime" in which we describe the processing methods (averaging raw data over 1 m bins, and then calculating Rib and dtheta_v/dz over an altitude range of 30 m with 5 m resolution) in greater detail to accommodate this and

other referee comments. We discuss why we choose 1 m as the bin size for averaging the raw data (lines 197-199) as well as how the choice of this bin size affects the results (lines 551-561).

*L250-255: Assuming that in the lowermost ~5m theta_v will have its strongest gradient this method has its weakness when no data from this level can be used, as the CBL would be found at higher theta_v, thus resulting in an underestimation. If only the ascent data below ~5m is ignored but descent data is used this should be clearly indicated. An alternative would be to use IR surface temperature estimates. Overall this appears like a very objective method, which can be fully automated.*

Both ascent and descent data below ~5 m is ignored, since typically below 5 m is sampled with manual flight during takeoff and landing. We have added a sentence to explain this (lines 202-204). The IR surface temperature estimates from the DH2 are not accurate enough to aid in this because they are not thoroughly calibrated, so they are primarily used to qualitatively identify variations in surface temperature (e.g., resulting from leads or melt ponds). Now that we present the subjective criteria in a table, we have condensed the information and in doing so, have also removed the insinuation that CBL height is dependent on the exact 5 m theta_v value when it is subjectively identified. A similar fully automated approach is applied with the objective Liu-Liang method for a CBL (lines 317-319), and since this objective method uses exact thresholds, it is dependent on the lowest DH2 value and thus is more subject to underestimation (this is noted as a drawback of the objective methods in the conclusion on lines 647-650). This is why the subjective ABL height is taken to be the more accurate one.

*L250. "theta_v" without "the"*

Due to the presentation of the subjective methods in the form of a table, this sentence has been rephrased and this comment is no longer applicable.

*L251: Replace "identify" e.g. with "determine" to avoid repetition.*

Due to the presentation of the subjective methods in the form of a table, this sentence has been rephrased and this comment is no longer applicable.

*L266: Consider changing to "slope of theta_v". At which heights are these multiple shifts?*

These multiple shifts occur around 30 m, 100 m, and 130 m. However, to accommodate presentation this information in a table, rather than paragraphs, we had to removed excess information, including details about the location of specific kinks in the example profiles. We hope the difference between the theta_v profiles in Fig. 2b and 2c are evident enough that we don't need to explicitly state this information in the manuscript.

*L271: The determination is not made entirely based on the humidity. In the last step, theta_v is used again.*

The way this information is presented was changed to fit into the table format, so this sentence has been restructured. Thus, this comment is no longer applicable.

*L278-280: If DH2 data is only considered from altitudes above a certain threshold this statement is not well supported unless surface temperature estimates from an onboard IR sensor are taken into consideration. Note that IR surface temperature estimates may be subject to uncertainties related to sensor temperature stability, the emissivity of the surface, radiation flux divergence, and sensor tilt.*

This statement was meant to explain a characteristic of a SBL, by definition, rather than by what was observed with the DH2. However, due to restructuring this information into table format, this sentence has been removed and this comment is no longer applicable, though we do still note that the gradient of theta_v is positive in an SBL on lines 229-230. And again, as explained in response to one of your above comments, the onboard IR sensor measurements cannot be used to derive exact surface temperature measurements.

*L284-285: This is often related to an inflection point in the wind profile or at least the layer where wind shear approaches zero. Showing both profiles of theta_v and wind speed would be illustrative, for the interpretation of Ri_b.*

We specify now that Rib increases when wind shear decreases above the ABL (line 283). We already provide both profiles of theta_v and wind speed in the original manuscript's Figure 7 (Figure 3 of the revised manuscript), as well as for each flight in the Supplementary Figures (S5-69), so this information can be found there. We don't add this to the subjective methods figure, as the wind speed profile is not directly used.

*L290-L291: This statement implies that the correct SBL height is known. One could also choose to define the SBL height as the level where there is such a clear shift in Ri_b. One may then simply end up with a different height.*

The way this information is presented was changed to fit into the table format, so this sentence has been removed. Thus, this comment is no longer applicable.

*L305: See previous comment: Does this imply that the ABL height can be determined with a resolution of 5m or 30m?*

Objective ABL height can be determined with a resolution of 5 m when the method ultimately relies on the dtheta_v/dz or Rib profiles. Otherwise, it can be determined with a 1 m resolution. The authors have added two sentences to specify this on lines 310-312.

*L308-309: Do you use equations 1-3 for the determination of the regime? Please include a statement, making this crossreference since this (is as you indicate) slightly different from the Liu-Liang method.*

The authors now specify on line 317 that the regime is determined using what are labelled as equations 2-4 in the revised manuscript.

*L318: Note that the height of the lowest level is critical. If the levels close to the surface are not sampled this may become an issue. You may want to discuss this later on.*

Since the DH2 samples down to ~5m (and sometimes lower), we can be fairly confident that CBL heights calculated using the DH2 data are accurate to a few meters. Issues do arise in the radiosonde data which does not sample as close to the surface (lowest 23m or higher), but the result of this is that a shallow convective layer is missed, and the stability regime is identified as an NBL or SBL. This uncertainty then is not due to the height of the CBL, but rather due to the fact that other methods were applied, as the CBL was not recognized. We have added a few sentences to the conclusion stating this drawback of the radiosonde data (lines 647-650).

*L320: "the atmosphere"*

In the revised manuscript this sentence was removed so this comment is no longer applicable.

*L323: The notations for vertical gradients are not consistent, compare e.g., line 304. Please stick to one notation.*

Due to shortening of the methods section, we removed this section, and this comment is no longer applicable.

*L326, Eq 6: Gradients would have to be determined between two layers so from k to k+1 (or k-1 to k+1). It is hard to follow which of these two levels is chosen as the ABL height. Depending on your resolution this makes a difference for the ABL height. Since this is a reoccurring issue for all methods relying on vertical gradients a general statement at the beginning of section 2.2 would help.*

The section we added before section 2.2 titled "Preparing the DataHawk2 data for analysis" (beginning on line 187) describes the resolution and how averages and gradients are calculated, which should clear this issue up.

*L334-335: The chosen threshold is quite different from the originally proposed one. What is the reason for considering this as inappropriate?*

We consider it inappropriate because the ABL heights found with the original threshold were far too low compared with the subjective ABL height and had no physical basis when analyzing the profiles; this may be due to differences in the vertical resolution or smoothing methods of our data versus that used by Liu and Liang (2010). The authors add some text on lines 321-323 to specify this.

*L341-342: This statement appears very general. TKE is just one way to define the SBL height. This goes back to my general comment on a more critical reflection on ABL heights. It should also be moved to the introduction.*

The authors have moved this statement to the introduction, and specify now that TKE is just one methods that often works well for SBL height identification, but is not used since we do not have this data (lines 106-108).

*L343: Below the ABL there is only the surface, which is only buoyant in an oceanographic sense.*

The authors meant to say "within the ABL." We have fixed this (line 326).

*L344: simpler: "the SBL height" or "the height of an SBL".*

This change has been made (line 327).

*L356: Consider changing the subscript to account for different ranges of theta^dot_r for different regimes (compare Sections 2.4.1.1-3). BTW, what does the r stand for?*

Due to shortening of the methods section, we removed this sentence, and this comment is no longer applicable.

*L391: better "starts" - "extends" may be associated with an upward direction.*

Due to shortening of the methods section, we removed this sentence, and this comment is no longer applicable.

*L396: layer or level?*

Layer is correct (line 336).

*L402: "the stability regime"*

Due to shortening of the methods section, we removed this sentence, and this comment is no longer applicable.

*L413: "local maximum"*

This change has been made (line 349).

*L416: "local minimum"*

This change has been made (line 351).

*L420: Can you provide a brief interpretation of this figure as done for the previous methods/figures?*

To shorten the methods section, we have removed interpretations of the figures for the other methods and moved Figs 3-6 to the Supplementary Figures. To remain consistent, we don't add interpretation of this figure.

*L429: I suggest using the term "threshold value" to avoid a bit of the discussion on the critical value of Ri (see also comment above). One could interpret the following paragraphs as you*

*were trying to find Ri_bc for the transition between turbulent and laminar flow based on your observations, but in fact, this is not the scope of this paper and you don't make use of any turbulence observations.*

We no longer use the term 'Ri_bc', but rather use the phrase 'threshold values' (line 360) to indicate how we identify the ABL height. We don't believe that the text can be interpreted that we are trying to find the critical value for the transition between turbulent and laminar flow based on our observations (line 358-360 and 363-364). If you feel that this is still implied, please let us know what statements convey this idea and we will revise them.

*L445-447: I get an idea of what is meant here, but I would suggest reformulating this sentence to make it more clear. Can this be broken up into two sentences?*

Based on the suggestion of another referee, we have removed this sentence altogether.

*L451-452: Can you summarize the main differences that may play a role?*

We have added a brief summary of the main modifications to the original objective methods (lines 369-372) and a brief discussion of why they are necessary (lines 372-375): "These adaptations are necessary in part because previous implementations involved analysis of radiosonde profiles, which have a lower resolution than the DH2 profiles, and in mid-latitude locations, where the ABL structure is often quite different than that observed in the Arctic (due to the lack of daytime convection or a diurnal cycle in the Arctic most of the time)."

*L453: "...applied to radiosonde data ..." is enough, "to identify ABL height" is implicit.*

This paragraph has been restructured, so this comment is no longer applicable. However, we made sure in revising this paragraph to remove redundant or implicit information.

*L454: It would be natural to state the number of the radiosonde profiles you used somewhere here.*

This has been added (line 380-382).

*L458: no "the" before "theta_v"*

This paragraph has been restructured, so this comment is no longer applicable.

*L459: Consider changing to e.g., "create profiles of the same parameters as for the DH2 data"*

This paragraph has been restructured, so this comment is no longer applicable.

*L462: can you give the reason for this inaccuracy?*

This is due to the Polarstern acting as a heat source. Specification of this has been added to the text (lines 388-390).

*L469: Stick to one common unit for vertical temperature gradients. I propose K/km.*

The authors disagree in this case, and think that changing the units in this sentence would be confusing to a reader. This sentence is discussing how we calculate stability regime for the radiosondes over a 30 m range using an adaptation of the values given in equations 2-4 of the revised manuscript. Changing the units here would make it confusing where we got the numbers from, whereas now we believe it is clear that we are adapting the original threshold of $\delta_s = 0.2$ K over 40 m to 0.15 K over 30 m (see lines 394-397).

*L474-476: Do you mean: "Similar figures for all available DH2 and radiosonde profiles can be found in …". Is it possible to use a hyperlink to get directed to this online supplementary material?*

We have changed the wording of this sentence to match what you recommend (lines 401-402). It does not seem to be possible to hyperlink to the Supplementary Figures document, however this is provided on the AMT site at the same DOI as the paper.

*L481: Consider reformulating, e.g.: "In general, the deviation between ABL heights from DH2 and the radiosonde increases with decreasing time proximity".*

This change has been made (lines 414-415).

*L507: This is one example where the subjective method appears as the "truth", although it is most likely not perfect, either. Consider adding "... compared to the subjective method".*

This change has been made (line 445).

*L510-511: Please note that there have been some high-level debate on the use of p-values and the 5% statistical significance, see e.g., https://www.nature.com/articles/d41586-019-00857-9. I am not an expert in statistics but I recommend at least using a somewhat "softer" formulation, like "can be considered statistically significant when the p-value is less than 5% (or 0.05)." If your p-value is 0.05, this means there is still a 5% chance that your result is completely random.*

We have adjusted the text to be "softer" based on your recommendations (lines 445-447).

*L513 and elsewhere: Consider using superscripts "Ri_b^0.5" and Ri_b^0.75.*

Instead of using superscripts, the authors use Rib(0.5) and Rib(0.75). These are introduced on lines 364-366. We fear that the use of superscripts might make it appear to a reader like we are taking an exponent of the bulk Richardson number.

*L519-520: See comment above*

We have clarified that this is compared to the subjective ABL height (line 454-455).

*L520: What is more complex, the method or the result from the method?'*

We specify now that we refer to the results of the method, not the method itself (line 455-456).

*L532: "R^2" without "number"*

This change has been made (line 465).

*L534: "0.5 to 1"*

The authors prefer the original text, as the wording you suggest may be confusing to a reader (line 467).

*L556-559: Very wordy sentence for saying that you assess the (cumulative) frequency distribution for the difference of the objective methods relative to the subjective one.*

We have condensed this sentence using the wording you suggest (line 497-498).

*L571: Consider changing to "number of cases within each (relative difference) category"*

We have made this change (line 510).

*L575: Consider replacing "predicts" since it's a diagnostic method.*

We have replaced "predicts" with "results in" (line 514). We have also made this change wherever else the word "predicts" was used.

*L581-584: Here, I would like to see some discussion on such aspects. Are there differences in the sampling or data processing methods that may lead to the fact that different threshold values work best? Such discussions may be very useful for the research community as they may have to adapt threshold values depending on their observational approach.*

We predict that the better efficacy of the lower Rib critical value for the radiosonde data results from the fact that more smoothing procedures are applied to the radiosonde data when they are processed by Vaisala. Minimal smoothing is applied to the DH2 data – the only smoothing comes from the vertical averaging over 1 m bins. We believe this to be the cause, rather than the difference in vertical resolution, because we have run the routines on the DH2 with different vertical resolutions (we tried vertically averaging the data over 5 m, 10 m, and 20 m bins) and found no significant difference in the efficacy of the Rib method with either critical value. We have added some text on lines 521-526 (before Fig 6 of the revised manuscript) which states the hypothesis that smoothing is what makes the difference in Rib critical value, and some more text on lines 551-561 which discusses that Rib critical value is not sensitive to vertical resolution.

*L588: "the ABL heights"*

We now say "After comparing $Z_{ABL}$ from the different objective methods…" as we have changed the syntax for ABL height (line 537).

*L595: Consider replacing "it is not consistent enough to be reliable" with "it is not reliable".*

This change has been made (line 563).

*L598-599: Do you mean the LLJ core? This statement could need a reference.*

We now clarify that we refer to the LLJ core (line 567). This was intended to mean that the LLJ core was observed to usually be above the subjective ABL height in the data used in the study, which also agrees with the literature. We have thus added citations, but also note that this is seen in our observations (lines 567-568).

*L600-601: "throughout the whole profile"*

This section has been restructured to list the reasons for objective method failure primarily in a table, rather than paragraphs. Thus, this sentence no longer exists as it originally did, but we use your suggestion when we phrase option 1 of Liu-Liang failures in the table (Table 4 of the revised manuscript, beginning on line 584).

*L625: radiosondes were launched from the deck so "right at the surface" should be changed to "close to the surface"*

Due to restructuring of this information in table form, some details were removed, including this one. Therefore, this comment is no longer applicable.

*L628: I suggest changing to "during polar night".*

Due to restructuring of this information in table form, some details were removed, including this one. Therefore, this comment is no longer applicable.

*L630: There is at least a debate on whether the free atmosphere above the ABL is really laminar or rather weakly turbulent. The choice of a threshold value for Rib largely depends on the vertical resolution you use to compute Ri_b.*

The authors have restructured this sentence, and now say: "Lastly, the failure of the $Ri_b$ method occurs due to the difficulty of defining an accurate critical value which correctly captures the likelihood of turbulence for all cases" (lines 579-580). In attempting to determine if the threshold value for Rib depends on the vertical resolution of the data used to compute Rib, we found no sensitivity of the efficacy of the critical values of 0.5 and 0.75 when vertical resolution of 1 m, 5 m, 10 m, and 20 m, are tested (discussed in lines 551-561). Above this, the range over which Rib itself is calculated would have to be adjusted, and at this point the applicable threshold value for Rib would probably increase. We have added a sentence to the end of section 3.1 which recognizes this (lines 560-561).

*L649: only "several", "different" is implicit*

This change has been made (line 607).

*L649: "methods (i.e., Liu-Liang …)"*

This change has been made (line 607).

*L652: You could state the threshold values 0.5 and 0.75.*

This change has been made (line 610)

*L656: It is occasionally quite good, better to use"largely".*

This change has been made (line 613).

*L663: Is this also true for largely ice-free areas in the Arctic, which are likely underrepresented by a sea-ice-based campaign?*

We now specify that this refers to sea ice regions, as the ABL structure over ice free regions in the Arctic can be quite different and we don't address this in the current study (line 617).

*L671: This repetition should not be necessary.*

This sentence has been removed.

*L672: It should be safe to use active voice: "would change minimally"*

This change has been made (626).

*L679: "These similar conclusions" (plural)*

This typo has been fixed (632).

*L681: "no method" or "no single method"*

We now say "no single method" (line 636).

*L681-683: Again: The objective methods may be better than a visual inspection by a non-expert. A combination of both visual + objective may be better. For the semi-automatic approaches, the list of features that may cause certain methods to fail would be very useful. One different approach could be to use an ensemble of automatic methods and visually inspect only the profiles for which the resulting ABL heights diverge.*

The authors now specify that the subjective methods are most accurate, but require much time and knowledge of ABL dynamics, and thus the objective methods may be a better choice for a

non-expert (lines 636-641). We also now suggest the semi-automatic approach of using an ensemble of automatic methods and visually inspect only the profiles for which the resulting ABL heights diverge, as you suggest (lines 651-653). Additionally, the features that cause certain methods to fail is now listed in a table (Table 4 of the revised manuscript), so that they can be more easily found and applied by a reader. We now reference this table in the conclusion (lines 646-647) so that in case a reader is skimming the paper by reading the conclusion first, they will find that this information is included in the paper.

***Figure 2: Caption: "each flight" is misleading when showing only selected flights***

We now specify that we refer to each flight shown in the figure (line 294).

***Figure 3: The caption is extremely long. The legends could be merged and plotted only once (applies to more figures)***

We have moved Figures 3-6 of the original manuscript to the Supplementary Figures document and now provide one legend for each figure, instead of for each panel (see Supplementary Figures S1-S4). With this, we also have shortened the captions without removing key information.

***Figure 7: This figure should be redone. Here are some suggestions: Use different color schemes for the figures, e.g., not two different shades of green in the same panel (Panel 5 also has several different shades of green for the horizontal lines to indicate ABL heights from the different methods). What are the dashed lines in Panel 5? Since the ABL heights shown as text in Panel 1 and 2 are also related to Panel 3-5 it would make more sense to put them in a small table (2 lines 6-7 rows), placed under the 5 panels. Use one common legend for all 5 panels. Consider using a logarithmic scale for Ri (only if all values are positive) or a narrower range. Condense the caption and structure it better. These suggestions could partially also improve the other figures.***

This figure has been redone according to your suggestions (see Fig. 3 of the revised manuscript). Now, we use black and grey lines for the vertical profiles of DH2 and radiosonde data respectively. Only horizontal lines are colored, and we have changed these colors to be more distinct. Additionally, all DH2 related lines are solid and all radiosonde related lines are dashed. We have moved the wind speed profile to be the 5th panel, instead of the 3rd. We now provide only one common legend which refers to the whole figure. We have decreased the range for Rib from 10 to 5. We have added a SBL example to this figure since we have moved Figures 3-6 of the original manuscript to the Supplementary Figures document. We leave the text of ABL heights on Panels 1 and 2 so that the information is all in one place and is easier to keep track of, now that we have added an additional set of panels for the SBL. With all of these changes, we were also able to shorten the caption without removing key information.

***Figure 8: Consider, using a smaller range for the y-scale in the bottom panel and rather mention that a few outliers are not visible with this scaling. It may also be simpler to use "Relative difference" instead of "Absolute value of the percent difference".***

These changes have been made (see Figure 4 of the revised manuscript).

*Figure 10: This may be my personal preference, but it might be better to display this as a CDF plot (four lines) or a histogram (four bars for each bin) using bins with a constant width (e.g., ranging from 10% to 20%). It would also be possible to combine the CDF and histograms in one panel. Consider using a different y-label, e.g., "frequency of occurrence" and only one common x-label. For the "No ABL Height found" class you could simply add NaN or display them differently, e.g., plot them as shaded bars or horizontal dashed lines (sort of downward from 100%). I also note that the bars in the last two columns don't add up to 100%. Do the missing cases indicate a relative difference exceeding 100%?*

Some of these changes have been applied, but we still use the same bar plot as before, as we believe it accomplishes what you want to see (see Fig. 6 of the revised manuscript). The changes made are 1) we use the y-label of "frequency of occurrence" that you recommend, and now list the platform type in a title for each panel, 2) we use one common x-label and just indicate the % below each set of plots, 3) for the "No ABL height found" class, we label this as "No Z subscript ABL", as we found NaN to not be fitting, and 4) we use diagonal lines in the bars for the "No Z subscript ABL" cases to differentiate them. Yes, the missing values that cause it not to add up to 100% indicate a relative difference exceeding 100%. We do not add comments on this to the text as we believe this is implied.

**Anonymous referee #3**

*Review of « Testing the efficacy of atmospheric boundary layer height detection algorithms using uncrewed aircraft system data from MOSAiC" by Jozef et al.*

*This study compares different methods to determine the ABL height from uncrewed aircraft system and radio-sounding from an icebreaker in the central Arctic Ocean during the MOSAiC expedition. "Subjective" visual detection methods are compared to four "objective" methods for both the UAS and RS profiles. The difficulties of ABLH estimation largely depends on the peculiarities of ABLH in Arctic, which are shallow ABLH and a high percentage of NBL cases (and a low percentage of CBL cases).*

*Major comments:*

- *The methodology sections (§2.3-2-5) is very long and contains many redundancies. I encourage the authors to structure the paper with greater caution. For example,*

  o *2.1 describes the DH2 UAS data. § L152-162 relates however to the ABL identification methods.*

The authors have moved this information to a new section titled "Preparing the DataHawk2 data for analysis" (beginning on line 187) where it better belongs so that it is not conflicting or distracting from the purposes of the section, which is to describe the DH2 UAS data.

  o *the four objective methods are cited four times (L23, L89, L156 and L300) before to be described under §2.4.*

The authors have changed the text so that the objective methods are only cited twice before described under section 2.4 – once in the abstract (lines 24-25) and once in the introduction (lines 120-122). We hope this change helps the text to not be as redundant.

> o *Part of the subjective method refers to some objective method, so that I would first describe the objective methods and then refer to similarity of the subjective method. For example the bulk Richardson method is to some extent describes at several places*

The authors think it is important to first explain the subjective methods, since subjective ABL height is included in the figures which demonstrate the application of objective methods (Fig. 3 and the Supplementary Figures S3-S6 of the revised manuscript), to show how the two compare. If we had not yet explained subjective ABL height identification, the figures could then be confusing. However, we have worked to not repeat the underlying physics guiding ABL height identification in the objective section that are already described in the subjective section.

> o *The description of the ABL structure and the corresponding profiles -with Stull as reference (30 mentions)- is dispatched in the whole paper. A clear description of the ABL structure in the Arctic ocean would benefit the comprehension of the reader*

The authors have added a paragraph to the introduction which summarizes the primary features of the Arctic ABL structure: stability regime, turbulence, and capping by a theta_v inversion (lines 61-73). We have thus removed this information from where it is stated later in the text. We do leave some of the description of ABL structure in the subjective methods section, as it is immediately relevant to methods described.

> • *The term "subjective" and "objective" methods seem to be inappropriate. The "subjective method" relates to "manually" or "visual inspection" performed by a person. The objective method relates to automatic algorithms performed by computers. Moreover, the subjective method contains to some extent elements of the objective method (e.g. the use of the Rib profile, but with a threshold described as 0.25, CBLH is the parcel method, the use of RH gradient,…).*

The terms 'subjective' and 'objective' were chosen to simplify what they really mean (visual/manual identification vs. automated algorithms performed by computers.) The authors choose not to change the use of these words throughout the paper, but have better specified their true meaning near the beginning (lines 111-114 and 220-224). Additionally, though the subjective methods do look at the Rib profile, we subjectively decide where Rib increases significantly rather than using a fixed threshold as in the objective method. Thus, just because we use Rib doesn't mean the method isn't subjective. The authors have adjusted the text describing the subjective methods to clarify this (lines 272-285 and Table 3 of the revised manuscript).

> • *The subjective method is considered as the best ABLH estimate. It is however also prone to error and several experiences have shown a clear ABLH uncertainty if several persons were estimating the ABLH. Were all the ABLH estimated by a*

*unique person? A comparison between several subjective estimation could be performed? Second the criteria/profiles mostly used for the subjective SBL and NBL heights estimation are not mentioned. For example, the best correlation between the subjective method and the bulk Richardson number method found (Figs 9-10) is perhaps due to large weight of the Rib profiles in the subjective method. The subjective method is then a mixing of several criteria, a mixing that could also be applied to the objective methods. Since this exercise can be quite difficult, why not comparing visual and automatic detection with the same criteria/profiles (e.g. only with the potential temperature or the Rib profiles?)*

The authors recognize that even a subjective ABL height identification is prone to error, and have added some text discussing this at the beginning of section 2.3 (lines 251-255). Since one best subjective ABL height identification criteria is not found amongst the literature (since there is much disagreement) the best way to get around this is for us to take into account some of the most commonly used methods in the literature and remain consistent in our ABL height criteria across all profiles, which was agreed upon amongst the authors. Discussion of this has also been added to the beginning of section 2.3 (lines 251-255). To ultimately settle on the subjective ABL height for each case, the lead author and the second other author reviewed all the cases together, and based on the patterns in all cases, discussed and refined the criteria and ABL depth, based on both of our analysis of the profiles. The other authors, as well as some non-co-author colleagues, also provided input for especially difficult cases. Next, you say that the best correlation between the subjective method and the bulk Richardson number method found (Figs 9-10 of the original manuscript, or Figs 5-6 of the revised manuscript) is perhaps due to large weight of the Rib profiles in the subjective method, but really the most weight in the subjective methods is given to theta_v profiles, and secondarily to the humidity profiles - Rib is only used heavily in few difficult cases. The authors have clarified this in the subjective methods section (lines 288-290). Lastly, you are correct that the subjective method is a mixing of several criteria, which would be difficult to apply objectively, which is precisely the point of the paper – to find which single objective method estimates an ABL height most similar to that found when mixing several criteria manually. Only comparing a single subjective criteria to a single objective criteria would not provide as useful of results as are already given. Therefore, the authors do not perform this analysis, but do add a sentence clarifying this point in the beginning of section 3.1 (lines 433-436).

- *Similarly, the subjective method uses RH profiles, why not applying an objective gradient detection to the RH profiles?*

This was not done as there is very little literature on applying an objective method to the RH profile. Most weight in literature is given to the methods currently discussed in the paper, which is why we choose these 4 to focus on.

- *This comparison of the objective methods between themselves would also be of high interest and is really lacking.*

The efficacy of the different objective methods can be seen in the bar plot (Fig. 10 of the original manuscript, or Fig. 6 of the revised manuscript). If you are suggesting similar comparisons using

each objective method as the basis for figures like Fig. 6, we feel that this would add significant length to the manuscript without much benefit. Since all of the objective ABL heights for each profile are listed on the figures in the Supplementary Figures document (S4-S69), it is possible for someone reading this manuscript to compare one particular objective method to all of the others. Since the purpose of this paper is to compare the objective with the subjective ABL heights, the authors think this amount of comparison of the objective methods between themselves is sufficient. Please let us know if this does not address what you were looking for, and please provide more input on what you were intending.

- ***The stability regime could also be further used to explain the differences between the applied ABLH detection. For example, the symbols of Fig. 9 (apart from the blue and red colors for UAS and RS) could be differentiated as a function of the stability regime, allowing to identify potential systematic causes for the observed differences. Some similar uses could also be thought for Fig. 10.***

The authors like this idea, and have addressed this by creating identical figures which are shown in the Supplementary Figures document, which contain only SBL datapoints (S70 and S72) and only NBL datapoint (S71 and S73). We chose this instead of using different symbols for SBLs and NBLs as these plots are easier to interpret than if we use different symbols in one plot. We discuss the main takeaways of these figures in section 3.1 (lines 491-496 and 531-536) by highlighting which methods have the highest efficacy for each stability regime. In section 3.2, we recommend which objective method(s) to use if one were to choose a different method based on stability regime (lines 592-600).

- ***To avoid confusion, I would use the term "inversion layer" instead of only "inversion" to describe the atmospheric layer with temperature or humidity increase with altitude.***

This change has been made.

*Minor comments:*

- ***L63-64 and Table 1: I think that Table 1 mixes measured profiles, computed variables and, to some extent, detection methods. E.g. the virtual potential temperature is a variable that is used alone in the Parcel Method (PM) but it is also the main component of the bulk Richardson number (Rib) method. In that sense, I think that the name of the method and the used profile data are much more useful than the "variables" given in Table 1. The potential temperature is also used for the PM, Rib and gradient method. The difference between "component-wise wind speed perturbations" and "wind shear" are not directly understandable. Moreover, this list of "variables previously used to identify ABLH" is not complete (e.g. all the methods based on aerosol concentration are not mentioned).***

The authors have added some text introducing Table 1 to clarify that we list only some of the atmospheric variables used for ABL height identification (lines 85-90). We also clarify that we list only thermodynamic and kinematic processes, as these are the focus of the paper. Since not all of the variables (or used profiles) listed have a specific 'method name', we don't add this

column to the table, but we do change the heading 'Variable' to say 'Quantity Used.' We have also merged "component-wise wind speed perturbations" with "wind shear" into one common quantity labelled as "wind shear," as these two quantities are essentially the same thing.

- *L69-71: rephrase, grammatical problem.*

The authors see no grammatical problem with this sentence, but have added a comma to make it flow more smoothly (line 96-99).

- *L 74: "to either decrease or increase more above the ABL": does this correspond to an increase/decrease of the gradient of the humidity?*

This does refer to an increase/decrease of the vertical gradient of humidity. We now write it this way (line 101).

- *L86-95: (see also main comments) I wonder if the use of subjective/objective methods is the right one or if it corresponds to an opposition between "manual" (=performed by a person) and automatic/operational detection method. I suppose that a person estimate the ABL heights "visually identified through combined evaluation of θv, humidity (both relative humidity (RH) and mixing ratio), and Rib profiles" via the same criteria as the objective methods.'*

We now clarify that the 'subjective' method refers to a manual process, while the 'objective' methods refer to automated processes, when we introduce these two terms (lines 111-112). One could manually apply the objective methods as well, however combining aspects of the different objective methods into the subjective method we have developed provide the best estimate of ABL height based on taking into account all of the relevant physical processes at play, and considering them in conjunction which is not possible with an automated method in the same way as is possible with the human brain. We now point this out as well (lines 251-255 and 306).

- *L154-159: This information has already be given.*

This sentence has been removed in the process of restructuring the Data and Methods section to add a subsection titled "Preparing the DataHawk2 data for analysis." Thus, we no longer repeat this information.

- *L164-167: Before to compare ABL height from radiosondes and UAS measurements, a comparison of the measured T, wind and humidity profiles should be performed. However, this is perhaps already included in other joined papers on the MOSAiC expedition.*

We did visually compare the T, wind, and humidity profiles between the DH2 and corresponding radiosonde to make sure the measurements were similar, before proceeding with the analysis. We add a sentence which states this when we introduce the radiosonde (lines 384-386). The theta_v and wind speed profiles from the corresponding DH2 and radiosondes are shown in Figure 7 of the original manuscript (Figure 3 of the revised manuscript) and in the

Supplementary Figures to demonstrate the similarity between the DH2 and radiosonde profile. Usually, the closer in time of the DH2 and radiosonde launch, the more similar the profiles are (they agree well in the revised supplementary figure S23 but poorly in S40). Additionally, the UAS and radiosonde profiles for a few case studies are shown together in the MOSAiC atmosphere overview paper (Shupe et al., 2022) which can be found at https://doi.org/10.1525/elementa.2021.00060.

- *L 191: "in either direction" means in increasing/decreasing direction? I think it should be rephrased.*

We have rephased this sentence to be more clear. It now reads: "If this minimum is not either negatively (in the case of a CBL) or positively (in the case of an SBL) reached, the ABL is identified as an NBL" (lines 239-240).

- *L 193: does this number also depends on the uncertainties/noise of the measurements?*

Yes, it does. We have added some text to state that this number also depends on inherent uncertainties or noise in the measurements (lines 241-243).

- *L197: "between a SBL, NBL or CBL" in arctic (I suppose).*

This change has been made (line 245-246).

- *L194-203: this could be more efficiently explained and, consequently, shortened.*

This paragraph has been significantly condensed and now only contains the vital information (lines 244-247).

- *L215-216: what is meant by " the θv inversion is at its strongest"? does it mean that the positive θv gradient is at its strongest ?*

Yes, it means the gradient of theta_v is greatest at the surface. We now clarify this (line 260).

- *L236: delta represent the difference between the elevation z (at which Rib is computed) and the ground level.*

The way we calculate Rib, this is not true. To calculate Rib throughout the profile, the delta is always 30 m, but refers to a different altitude range. As we explain (and in the updated draft we try to explain more clearly), Rib is calculated over an altitude range (delta) of 30 m, with a 5 m resolution, starting at 30 m. Therefore, the lowest Rib value is calculated between measurements at 30 m and 60 m. The next Rib value is calculated between measurements at 35 m and 65 m, and so on. This information can now be found in lines 212-214.

- *L247: I don't understand what you mean by "over 30 m bins"?*

The authors realize that the word "bin" is inappropriate for what we are trying to describe. Instead, what we mean is that Rib is calculated over an altitude range of 30 m. We have changed the wording in the text to reflect this, and also provide an example so that this is more understandable. This reads: "$Ri_b$ profiles are created by calculating $Ri_b$ over a 30 m altitude range (Δ), at 5 m resolution (i.e., between 30 and 60 m, then between 35 and 65 m, and so on)" on lines 212-214.

- **L251-252: this is the description of the parcel method and has nothing subjective. The simultaneous increase of Rib is obvious since the bulk Richardson method with a threshold=0 corresponds to the parcel method.**

We have changed the presentation of this information to be in the form of a table rather than paragraphs (Table 3 of the revised manuscript). In doing so, we highlight only the subjective aspects of this method, and have removed reference to the specific threshold of theta_v returning to its surface value, since this was not exactly implemented, but rather just visually estimated.

- **L252 "this will not be the first altitude at which the virtual potential temperature increases with altitude": unnecessary, this is obvious.**

This has been removed with the presentation of this information in a table.

- **L 334: why is Liu and Liang's threshold is inappropriate for the current case? Due to arctic condition? How did you identify that it is inappropriate?**

We found this threshold to be inappropriate because the ABLs heights found with the original threshold were far too low. This may be due to differences in the vertical resolution of our data versus that used by Liu and Liang (2010). We added some text on lines 321-323 to state this.

- **L446-447: I do not see the use of this sentence**

This sentence has been removed.

- **L467-469 should be rephrased.**

We have split this into a few sentences to be more understandable (lines 392-397).

- **8: It's very nice to see the differences in ABL heights as a function of the time between launches. Other criteria such as the stability classes could also be used. Fig. 9 f) with ABLH from RS versus from DH2 also brings a nice overview of the comparison and should be discussed with Fig. 8. By the way, isn't Figure 8 already a result so that it should appears under §3?**

The purpose of this figure is simply as justification for using the radiosonde closest in time to each DH2 to test whether the methods also work well for the radiosonde data. The justification is that the radiosonde measured approximately the same ABL structure which is identified similarly by the different objective methods. Therefore, in this case we don't separate this analysis into

stability regimes, however we do add stability-regime-separated analysis for Figures 9 and 10 of the original manuscript (Figures 5 and 6 of the revised manuscript). Additionally, we leave this in the methods section since the purpose of the results section is to compare the objective to subjective ABL height for each platform rather than compare the objective ABL heights from each platform to each other. As just mentioned, this figure only justifies the validity of the methods which is why it is included in the methods section. We have added a sentence to state this (lines 421-423).

- *L549-551: is it due to the fact that RS cannot be used below 30 m? Or due to the interpolation if the RS go through icebreaker's plume?*

We do not think it is due to the fact that RS below 23 m cannot be used, or the interpolation through the plume, but rather because there are simply more samples of lower ABL heights, so we are likely to see more deviation. We have added some text on lines 480-481 to mention this.

- *L565-570: The main point with the comparison with Liu-Liang method is 1) (as described in the manuscript) this method works well for ~40% (within 20%) of the cases and 2) in about 40% of the cases, Liu-Liang has more than 100% difference with the subjective method.*

We have added your observation that the Liu-Liang has a high % of cases that are more than 100% different than the subjective ABL height (lines 506-507). However, this number is about 20%, not 40%. This is determined by adding the 52% of cases with Liu-Liang ABL height within 100% of the subjective ABL height, and 23% of cases with no ABL height found by Liu-Liang.

- *Liu-Liang method results in the largest differences with the subjective method: is it due to a bad classification of the stability leading to a false applied detection method? Which are the stability cases with the greatest differences?*

As mentioned in the text, the Liu-Liang method struggles with SBLs typically because the dtheta_v/dz criteria are not met anywhere in the profile, usually because a weak $\theta_v$ inversion persists throughout the whole profile, meaning that the method reverts to using the LLJ core height as the ABL height. However, the LLJ core was observed to usually be above the subjective ABL, so this predicts the ABL height to be too high. We have added discussion of this (lines 566-570). Likely the buoyancy thresholds set by Liu-Liang do not work well for a similar reason as why the Liu-Liang NBL threshold also doesn't work – due to differences in the resolution of the data they use to develop their methods.

---

## Author Response (AR2)

**Response to anonymous referee comments**

The authors again thank the three anonymous referees for taking the time to review the resubmitted draft of our manuscript and for their helpful comments, which have improved the manuscript further. Each referee comment is given below in **bold italics** followed by our response to the comment. The line numbers provided in our responses refer to line numbers in the revised manuscript.

**Anonymous referee #1**

***Second review of « Testing the efficacy of atmospheric boundary layer height detection algorithms using uncrewed aircraft system data from MOSAiC" by Jozef et al.***

***The manuscript was largely improved, the description of the methods is better structured and shortened and the results are more clearly presented. There is however still some points that have to be further clarified:***

***Major comment:***
***1. The calculation of the Richardson number (L213-215) has to be clarified: I don't understand why Rib is computed over 30 m range at 5 m resolution. If I understand the authors well, the Δθv and Δz of equation 1 are built over 30 m, i.e. the "ground reference" is always 30 m below the altitude at which Rib is measured ? I checked in Seibert et al 2000 and in Zhang et al., 2014, the Rib is always computed with the reference level at the ground. Stull mentioned (p. 177) that the critical value of 0.25 applies only for local gradients, not for finite differences across thick layers. No size is however given to estimate what is meant by "thick". Up to now, I have never seen such a calculation of Rib (if I understand well what the authors have done).***

Additional information has been added to the manuscript to specify why we calculate bulk Richardson number over a 30 m altitude range, rather than using the ground as the reference level. First, on lines 222-225 we specify that we use this 30 m range rather than ground level in order to isolate local likelihood of turbulence rather than calculating that over the full depth from the surface. Next, on lines 289-299 we discuss why using this running 30 m range, rather than the ground as the reference level, is actually beneficial for having a critical value that is consistent throughout the profile, and for that critical value being somewhat close to 0.25. This is because, according to Stull 1988 (page 177) as you mentioned, the critical value of 0.25 applies only for local gradients, and with a thicker layer, we are more likely to average out large local gradients. Well, when we use a consistently 30 m thick layer, we are essentially finding local gradients and are a lot less likely to average out large local gradients which may be averaged out when calculating Rib over an ever-increasing depth using the ground as the reference level. Thus, a critical value of 0.25 is more likely to apply when using the methods of Rib calculation that we use. Lastly, such a calculation of Rib has been used in previous literature, so we are not the first to introduce it. While many papers do use the method in which the reference level is always the ground, Georgoulias et al., 2009 (now cited in the paper) and Dai et al., 2014 (already cited in the paper) both specify that they calculate the Rib profile using gradients over distinct layers in the atmosphere, not using the ground as a reference. And they then use these profiles of

Rib to identify ABL height. Therefore, I am confident that we are justified in our method for calculating and our application of Rib profiles, and in fact our method makes more sense for detecting the likelihood of turbulence than the method of always using the ground as a reference level.

*Minor comments:*
*• Table 1: my comments on Table 1 remain valuable. E.g. how can we understand the differences between "virtual potential temperature" and "vertical gradient of virtual potential temperature" ? What is used in the first case ? Similarly, the gradient (or kink that is a difference in the gradient) are used for liquid water content and absolute humidity. The description could be much more precise.*

We have added a column to Table 1 called "Application of Quantity" in which we describe how the variables in the "Quantity Used" column are applied to find ABL height (Table 1 begins on line 97). Within each "Quantity Used" there is one or more subsections in "Application of Quantity" since some variables are used in multiple ways depending on the method. We hope this addresses your comment as we now describe the difference in the application of simply "virtual potential temperature" versus "vertical gradient of virtual potential temperature." We also now specify how the moisture profiles are used to identify ABL height.

*• L110: "manual visual analysis" or human visual analysis ?*

I don't understand the difference. How can manual visual analysis refer to anything but that done by a human? If there is no difference, the authors prefer the word "manual" to "human" and we leave it as is (line 116).

*• L114: can you explain the mention of "the definition of this quantity is not constant over time"?*

The authors apologize for this being poorly worded. We did not mean to say that the definition of the quantity is not constant over time, we meant to say that ABL height is not constant over time. We have changed the wording to reflect this line 119-120.

*• You should clearly mention in page 4 that "manual visual analysis" is refered as "subjective" later on. (this is done at line 221, but it is somewhat late)*

We had already included on line 117-118 that manually determined ABL height is referred to as 'subjectively' determined ABL height. Therefore, we do already mention this on page 4 and no additional change has been made.

*• Table 3 and Fig. 2:*
*o As stated in table 3 for the CBL cases " ZABL is the altitude at which the vertical gradient of θv is positive and is the bottom of a layer of enhanced stability ". Such a description remains quite evasive and the definition of the Parcel method suits probably much better ZABL: the altitude at which the virtual potential temperature becomes equal to the one at the ground (i.e. at 2 m). We can quite easily identify visually (and objectively) this altitude. If the increase in*

**Rib is found at the identified ZABL, the kink in humidity profile is not unique and could have been put at ~250 m**

The reason we do not include the criteria that $\theta_v$ reaches that at the surface (as is the case in the Parcel method) is because the surface value of $\theta_v$ may not be known in instances when these subjective criteria are applied. For example, usually the lowest DH2 measurement in a given profile is at 5 m, and the lowest good radiosonde measurement in a given profile is around 23 m (at least for the radiosondes from MOSAiC). Thus, such subjective criteria that rely on a surface measurement cannot be applied in many cases. Instead, indicating that the gradient of $\theta_v$ is positive and is the bottom of a layer of enhanced stability at the top of the ABL does not require the use of a surface measurement, and essentially results in the same conclusion of what the ABL height is in the central Arctic where the unstable layer of the ABL, if present, will be very shallow. With regard to your comment that a kink in the humidity profile can be also found at 250 m, this is true of the mixing ratio profile, but a clear kink at 136 m can be seen in the RH profile, and a faint kink here can also be seen in the mixing ratio profile. Since this altitude has kinks in several of the shown profiles ($\theta_v$, RH, r, and Rib), this is the altitude chosen for ABL height. We have thus changed the wording in this section of the table to say "…corresponding to a kink in the relative and/or absolute humidity profiles…" (CBL section of Table 3, beginning on line 311).

**o Fig. 2c: once again the increase in Rib is well marked but the kink in humidity seems much more obvious at ~150 m.**

The authors disagree with you in this case. While there is a change in the humidity profile at around 150 m, it is certainly not as pronounced of a kink as that which occurs at 100 m. In this case, we believe the increase in humidity above 100 m to be a humidity inversion, and when there is a humidity inversion, this usually occurs above the ABL.

**o Fig. 2d, 2e: similarly to Fig 2c, I would estimate the kink in humidity some 30-50 m higher. (I deduce that the unique description of Table would not allow several independent persons to deduce the same ZABL and I do understand much better the significance of "subjective". Moreover Rib seems to be a more determinant parameter than what is explained in the paper.**

We have laid out in our subjective criteria that we primarily look for a kink in the $\theta_v$ profile and secondarily in the humidity profiles (line 307-309); thus, if there are multiple clear humidity kinks, we look for the humidity kink which corresponds to a $\theta_v$ kink. Given that, in 2d and 2e, the relevant humidity kinks were found to be those that also corresponded to the most clear kink in the $\theta_v$ profile and an increase in Rib. You have brought up a good point that Rib seems to be a more determinant parameter than what is explained in the paper, so we have changed the wording to indicate that Rib is used more heavily for CBL and NBL cases, but for many SBL cases, the humidity profiles are used more heavily (lines 307-310) as they often provide more useful information especially in cases of very weak turbulence  Lastly, to clear up some confusion as to which humidity kink or $\theta_v$ kink to consider as ZABL, we have added a sentence to specify that, if there are multiple clear kinks in different profiles at different altitudes, then preferential treatment is given to the kink that also corresponds to an increase in $Ri_b$. We also clarify that if kinks in the relative humidity and mixing ratio profiles occur at different altitudes,

preferential treatment is given to the kink which occurs at the same altitude as that in the $\theta_v$ and/or Rib profiles (lines 323-325). We hope this added specification will better result in independent persons finding the same ZABL from these criteria. But in the end, these are "subjective" not "objective" criteria for a reason – they cannot be automated and are subject to uncertainty, but we have addressed this uncertainty (lines 318-327), and have concluded that the uncertainty is relatively small.

*• Fig. S4a: I do not understand why the ZABL identified with the Rib method is not set at the first altitude where Rib is greater than the threshold at about 100 m? This is also the case for e.g. Fig. S10, S13, S18, S19, S20, S21, S23*

The reason we do not simply take the first altitude at which Rib exceeds the threshold as ZABL is because, since we are calculating Rib over a 30 m range with ascending altitude, it is possible for there to be small local non-turbulent layer within the turbulent ABL. By requiring Rib to exceed the threshold for at least 20 m, we are looking for where the likelihood for turbulence has truly ceased above the ABL. We have added some discussion to section 2.4.4 to cover this (lines 389-393). Looking at Figure S4a, based on the $\theta_v$ profile, and the other subjective criteria, the subjective top of the ABL is just below 200 m, which is also the level above which Rib is consistently greater than the threshold. So this is actually a good example as why we do not simply take the first altitude where Rib exceeds the threshold as ZABL, because if we took it at the level of first exceedance of the threshold, the ABL height found would be too low. For the other supplementary figures you reference, if we take the first altitude where Rib exceeds the threshold as ZABL, this will also give a ZABL much lower than the subjective ZABL. We settled on a exceedance of the threshold by a consecutive 20 m (4 datapoints) as the criteria for identifying ZABL after trying many options and seeing which one performed best. Some examples of other options we tried are 2 of 3 consecutive datapoint, 3 of 3, 3 of 4, 3 of 5, 4 of 5, 5 of 5, etc.

*• Fig 4: since the ABL is rather shallow in Arctic and the uncertainties remains of about some 10 m (e.g. an uncertainty of 30 m is given for the subjective method (p. 11)). An absolute difference would probably give a better view. From Fig. 5f we can see that the relative difference between the subjective method applied to DH2 and RS can be as high as 200%.*

We have changed this plot to show absolute difference, rather than relative difference.

*• Fig. 5a: the red slope (correlation between the subjective method and RS) is mostly determined by the isolated points> 500m and does not seems me a relevant value to characterize the difference.*

This is a good point, and additionally, the very low R2 value for the Liu-Liang method indicates that there is not much correlation between the objective and subjective ZABL for this method, so analysis of the slope does not provide reliable information. We have added a sentence to the manuscript to address this (line 510-512).

*• L507-508: could you give a tentative explanation for the high number of cases with no Liu-Liang ZABL identification and with a relative difference > 100% ?*

The reasons for failure of the Liu-Liang method are already listed in Table 4 and discussed in Section 3.4, but we add a preview sentence at the point in the paper to which your comment refers, that the primary reason for the failure of the Liu-Liang method is the high prevalence of a weak θv inversion that persists throughout the entire lower atmosphere in the Arctic (lines 566-568).

**Anonymous referee #2**

*Minor comments:*

*L68: The entire free atmosphere and in cases of an SBL even the entire troposphere can be characterized by a potential temperature inversion. Do you mean "capping temperature inversion"? In an SBL the inversion is often surface-based and transitions directly into the free atmosphere without a capping inversion.*

Thank you for this comment – we agree that the way it was written before does not properly get the point across. We have changed the sentence to now say: "While the various forms that the Arctic ABL may take are complex, most of the time, the Arctic ABL is capped by a temperature inversion (which may extend to the surface for a stable ABL) and local maximum in potential temperature gradient, marking the entrainment zone…" (lines 71-73). We hope this clarifies what we mean.

*Table 1: It is not clear to me in which way the "Quantities used" are evaluated, e.g. 1) "virtual potential temperature" vs. 2) "vertical gradient of virtual potential temperature". Is this supposed to mean that in 1) you look at a bulk temperature gradient or difference between two layers? If taking the difference of potT between two layers this is essentially also a gradient. Or are you referring to a 2nd order derivative (curvature)? I suggest trying to make this clear by always using terms like: "gradient", "(bulk) difference", etc. Note that "bulk Richardson number" can stay as it is since the gradients/differences are implicit and since you use a simple threshold (same for TKE).*

We have added a column to Table 1 called "Application of Quantity" in which we describe how the variables in the "Quantity Used" column are applied to find ABL height (see updated Table 1, beginning on line 97). Within each "Quantity Used" there is one or more subsections in "Application of Quantity" since some variables are used in multiple ways depending on the method. We hope this addresses your comment as we now describe the difference in the application of simply "virtual potential temperature" versus "vertical gradient of virtual potential temperature."

*L104-106: The statement here contradicts what is stated in L68-69. I agree with the statement made here.*

We have changed the wording on the original L68-69 (now lines 73-74) so that it does not seem we are implying that the temperature inversion ceases at the top of the ABL: "…marking the entrainment zone, which is a stable layer that makes the transition from the ABL to the free

atmosphere (Stull, 1988)." Now, it should read that the temperature inversion just makes the transition between the ABL and the free atmosphere, but that temperature inversion can still extend (though weaker) throughout the free atmosphere. We hope you no longer find these two statements contradictory.

*Section 2.1/2.1.1: How is the altitude used for all of the following analyses determined? I suppose the DH2 provides different altitude estimates from different sources, e.g. GNSS, barometer, extended Kalman-Filter using both, etc. Each method has its uncertainty. Unless RTK is used GNSS uncertainty is in the order of 5m and often behaves "jumpy" when satellites are disappearing or popping up; barometers are subject to drift due to non-sufficient temperature stabilization of the autopilot (on older models) or background pressure changes (not always negligible over a 30min period). The bigger problem with pressure-based altitude estimation is that most autopilots do this internally based on the assumption of a standard atmosphere, which is, however, way too warm for Arctic conditions and thus causing errors in the order of ~10% or worse in very in cold conditions. Recomputing altitude based on pressure detrending and the hypsometric equation using ambient (measured) temperature is a simple way to reduce this uncertainty (see Barbieri et al., 2018 (https://www.mdpi.com/1424-8220/19/9/2179); Greene et al. 2022 (https://link.springer.com/article/10.1007/s10546-022-00693-x)). Radiosonde data may be subject to similar uncertainties.*

The altitude estimates are obtained using a GNSS receiver and barometer onboard. The barometer derived altitude (pressure altitude) is typically accurate to within a couple of meters but due to long observation periods (on the order of 30-40 minutes) tends to drift because the barometer is calibrated only once before each flight. Your concern with GNSS derived altitude uncertainty is justified. Although GNSS altitudes are 'jumpy' locally, they provide drift-free altitude estimates over the duration of the flight. The altitude used in DH2 analysis, termed 'GPS calibrated pressure altitude', corrects for the drift in high-resolution barometric pressure altitude with the GPS/GNSS obtained over the duration of the flight. We have added two sentences to summarize how the altitudes used for this study were obtained (line 154-157). As far as the radiosonde data goes, altitude is calculated using pressure measurements which are compared to the initial pressure at 10 m to determine altitude via the hydrostatic equation (line 418). Thus, one source of uncertainty is due to that of the pressure measurement, which is 1 hPa. Additionally, the pendulum swing and other motion under the balloon after launch could influence the altitude calculation, but these effects are generally smoothed during the data processing by Vaisala.

*L148 ff: Although this has been improved I am still not sure how the wind speed is computed: Can you cite one published paper describing at least a similar method. Mayer et. al (2012, https://doi.org/10.1260/1756-8293.4.1.15) is to my knowledge the first publication introducing the no-flow sensor method for determining horizontal wind speed based on GPS speed along circular flight paths. I can only guess that your method is similar, possibly including a correction for variations in true airspeed. Another useful reference might be Rautenberg et al. (2018, https://doi.org/10.3390/atmos9110422).*

We have restructured this paragraph so that the information for how the winds are calculated can primarily be found in the references cited. Winds from the DH2 have been calculated by two

methods, which are both included in the B1 netcdf files, via a "standard" approach, and a "hybrid" approach. We now briefly introduce each of these approaches and point the reader to two citations for each approach where they can read about the methods in depth. We also now specify that the winds used in this study are those from the "hybrid" approach (line 158-166). We hope that this now satisfies your comment, and a reader should easily be able to find the information on how the DH2 winds are calculated by primarily referring to the references, now that the paper describing the technical information regarding the DH2, which includes detailed information about how the winds are calculated, is in preprint (Hamilton et al., 2022: https://doi.org/10.5194/amt-2022-96).

*L154: There is some additional information given here, however, I don't regard this as very clear. Can you add some references here? It is not really subject to this review, but I expected the information to be also found in the metadata of the \*.nc files. For the sake of creating self-explaining files, can you add this?*

With the restructuring of this paragraph, we have added more references which describe how the DH2 winds are calculated. Now, at the end of the paragraph where we describe the wind estimation, we provide a sentence to inform the reader that more information about the processing of all of the variables that the DH2 measures can be found in the metadata for the netcdf files (line 168-170). This information is, and has been, provided in the metadata for the netcdf files already, so we are not sure what you are missing.

*L168-169: What is meant by "filtered to remove the impact of angle and ground speed". Do you mean a coordinate transformation from true airspeed to wind speed or filtering to remove high-frequency variations in the angle (which angle? Yaw or all Eularian attitude angles) and ground speed?*

This information is actually no longer true for the current way the DH2 winds are calculated. In previous iterations of the wind processing, we had applied more filtering routines, but in the final iteration, we do not. Thank you for bringing our attention to this sentence, as it was not our intention to include it. Thus, we have removed this sentence from the manuscript.

*L452: "0.3 to 1.2" not "1" also in line 467 an upper limit of 1 (which may be understood as 1.0) is not appropriate. I also suggest giving the same amount of digits for all values.*

The authors apologize for not being clear with what we meant here. In this section, when we discuss the slope values, we are comparing the slope to an ideal value of 1.00, which is what the slope would be if objective ZABL = subjective ZABL in every case (if the intercept is also 0). We have added some text on line 480-481 to introduce this concept before we get into the discussion. Then, when we said all the slopes are "within 0.3 of 1" for the DH2 and "within 0.5 of 1" for the radiosonde, what we really meant was that all slopes fall within 1.00 +/- 0.30 and 1.00 +/- 0.50 respectively. We have changed the text to reflect this (line 490 and line 505). We have also made sure to give the same number of digits for all slope values and R2 values in our discussion.

*L492-496 and S70-71: It is great that you included these additional analyses. I consider these*

*results very interesting and would encourage you to give higher attention to this. At first sight, I noted some additional important results that should, in my eyes, be highlighted, e.g.: the high discrepancy in the performance of the Heffter and TGRDM methods for the two different regimes. The Ri methods show less dependency on stability (for DH2) however they work better with a higher threshold value for NBL and lower SBL cases. In addition, the discrepancy between radiosonde and DH profiles becomes larger for some methods when dividing between regimes (Rib for NBL and RS is performing very poorly – any idea why?). This may suggest that some of the methods are not as robust across platforms as originally thought. These points should also be reflected in the discussion. Liu-Liang does apart from the 4-5 extreme outliers do very well in the NBL regime (see also S73). Consider moving these two figures to the main manuscript or maybe better, mark the different regimes in Fig5 with different symbols and add a table with the results now in the different figure panels to not overload the plots.*

We have moved supplementary figures S70 and S71 from the previous draft to the main text, and have combined them as a two-part figure that is now Figure 6 in the revised manuscript (line 544). We already tried portraying this information as one figure with different symbols for the different stability regimes, but found it to be much too busy to try to decipher the information from – it is much easier to visualize the main takeaways when you can see the datapoints from the different stability regimes separately. We have also added more discussion along with these figures to address the points you mention in your comment. Specifically, we address the difference in the efficacy of the Heffter and TGRDM methods between different stability regimes, the stability dependence of the Rib method, how the Rib method especially with threshold value of 0.75 does not perform as well for NBL cases, and how the Liu-Liang is actually pretty good for NBL cases. This discussion can be found in lines 532-543.

*Section 3.2 and Table 4. I wonder whether the question should rather be "Why does the subjective method result in different results" rather than "why does it fail?". E.g., 1a. " LLJ core altitude is well*
*above the ABL top" is not very helpful since we don't know the true ABL height. It would be more helpful to focus on the feature that is causing the subjective method to be different, e.g., wind shear or humidity profile which is not taken into consideration in the objective method. I am not 100% sure whether an approach like this is feasible for pre-screening, but I am not convinced that some of the currently listed features are either. For example, how should one use feature Hefter 1 ("SBL height is not the altitude at which θv is 2 K warmer than θv at the surface") if the SBL height is unknown? I recommend at least checking all the listed features for their applicability in a pre-screening routine.*

We have specified now that when we refer to "failure" of a method, we really mean that the objective ZABL is much different than the subjective ZABL (line 627-628). Since we have already concluded that the subjective ZABL is the best estimate of ZABL that we have, given the available data, if the objective ZABL is much different, we believe this constitutes a failure, i.e. the objective ZABL is most likely incorrect. The authors would like to keep the information provided in the table, but we have also added some discussion to point out that some objective methods might produce different results than the subjective method due to the lack of inclusion of humidity and/or wind shear features into the calculation (line 655-659). We have also added

some text which specifies that not all features in Table 4 may be conducive for pre-screening, but at least it is a starting point to identify some cases in which it is likely that certain objective methods may fail (lines 660-665).

***Technical corrections and suggestions:***

***Abstract: Include a statement on the stability dependency?***

We have added a statement which mentions we also discuss in the paper how the success of the methods differs based on stability regime (line 24).

***L52 "sea ice pack features": Remove "pack" and mention also open water.***
***Maybe it's best to use "underlying surface" and then specify the different surfaces and their features that can be relevant. Here a distinction between thick and thin ice (allowing for a substantial heat flux) may also be relevant and can be combined with the cold air advection example given in the following sentence.***

The two sentences have been adjusted to read: "In the central Arctic, the ABL is impacted by interactions between the atmosphere and underlying surface, including both sea ice and open water portions, which can cause either buoyantly or mechanically produced turbulence. The generation of buoyant turbulence can occur through surface energy fluxes emitted from open water regions such as leads (Lüpkes et al., 2008), cold air advection, especially over thin ice (Vihma et al., 2005), or turbulent mixing below cloud base due to cloud top radiative cooling (Tjernström et al., 2004)." (line 52-57).

***L55-58: It would be more natural to start with mechanical production due to surface roughness and larger features such as ridges (ocean waves are also roughness features only moving) and ice edges and then continue with LLJs.***

The sentence has been adjusted to read: "Mechanical generation, which is the dominant driver of turbulence in the central Arctic (Brooks et al., 2017), can occur due to the interaction between the atmosphere and surface roughness features such as ridges and ice edges (Andreas et al., 2010) or oceanic waves (Jenkins et al., 2012), or due to the presence of a low-level jet (Brooks et al., 2017; Banta, 2003)." (line 57-60).

***L59: change to "plays a minor role". The entire sentence is a bit misleading since solar heating still can play a role even though it may not cause buoyant thermals but can still decrease stability substantially.***

We have changed the wording to "plays only a minor role" (line 62), as you are correct that, even if solar radiation isn't creating an unstable situation with buoyant thermals, the addition of solar radiation to the surface energy budget will lessen the radiative cooling and thus lead to less stable conditions then if no solar radiation were present.

***L61 ff: Add the information on the frequency of occurrence of CBL to the beginning of the***

*paragraph. Although rather rare CBL is still important in the central Arctic and it is often linked to openings in the sea ice (leads and polynyas).*

We now mention that a CBL is rare in the first sentence of the paragraph (line 63). We also mention later on, when we describe how a convective ABL occurs, that when this does occur in the Arctic, it is likely due the presence of leads or polynyas (line 70-71).

*L 74: Consider mentioning the relevance for NWP or atmospheric modeling in terms of enabling boundary layer parameterization schemes.*

We now include mention of ABL parameterization in NWP models as a reason why it is important to know the ABL height from observations (line 86-87).

*L 84: add a statement like "based on thermodynamic and kinematic profile data from UAS"*

The sentence now reads: "The goal of the current work is to determine which methods, based on thermodynamic and kinematic UAS profile data, can best accomplish this" (line 88-90)

*L85-87: some redundancies*

We have combined the information on these lines into one sentence, which reads: "The depth of the ABL has been previously defined using a variety of approaches that involve visualizing the profiles of different thermodynamic and kinematic variables, which are listed in Table 1, along with some examples of associated literature that references use of that variable" (line 91-93). This should reduce the redundancy.

*L88: "vertical structure"*

This change has been made (line 94).

*L122: Which method is adapted to best suit DH2 data?*

All objective methods, aside from the Heffter method, needed some level of adaptation to best suit the DH2 data. This is now stated (line 128-129).

*L173: "near-surface wind speeds"?*

We now clarify that it is indeed the near-surface wind speeds (line 183).

*L186: "unfavorable environmental conditions"?*

This change has been made (line 196).

*L193-197: I suggest reformulating these two sentences. You give three reasons for bin averaging the entire flight, the first at the beginning and the second at the end of the first sentence. Reason 2 links well to the sentence before so why not start with this.*

We have restructured these sentences to read: "To further eliminate the effects of differences in sensor response times during ascent and descent, and for ease of visualization, we average the θv, humidity, and wind speed variables over 1 m altitude bins throughout the entire flight (e.g., values at 10.5 m are averaged from 10 to 11 m). This also mitigates the effect of changes in atmospheric conditions near the surface throughout the span of a flight, though the near-surface observations largely remained constant during a given flight" (line 203-207).

**L213: should be "(delta z)"**

This change has been made (line 223).

**L215: consider adding that this results in Ri at z=15m, 45m, 50m, 55m, etc.**

This has been added (line 227).

**L216: the average is only an argument that it doesn't matter in most cases, but there might still be extremes that cause an error. Can you mention the maximum drift speed during your flights and include a statement that the error is in any case limited to the first level where Rib is determined?**

We include now the maximum drift speed during the DH2 flights, which was 0.3 m/s (we also determined the average drift speed during DH2 flights was 0.09 m/s, which is approx. the same as the stated average drift throughout mosaic), and have added a sentence stating that any error that due to the drift speed is limited to the first level where Rib is determined (line 229-231).

**L219: replace "dthetav/dz" with "it"**

This change has been made (line 232).

**L224: "physical processes" doesn't fit in this context.**

What we meant with this statement is that the underlying physical processes that dictate the ABL structure and height are considered similarly in both the subjective and objective methods. We have changed the wording to clarify this (lines 237-238).

**L225: "the stabiltiy regime"**

This change has been made (line 239).

**L238: replace "gradient" with "difference"**

This change has been made (line 252).

**L243: replace "number" with "threshold"**

This change has been made (line 257).

*L272-273: More precisely, Rib is an approximation of this ratio.*

We specify now that Rib is an approximation of this ratio (line 286).

*L286: "stability regimes"*

The authors disagree that "regimes" should be plural here. We therefore leave it as is (line 305).

*L287 ff: Can these kinks be found more reliably from the second derivative (or its bulk approximation)?*

While these kinks can also be found using a second derivative, this removes the benefit of analyzing the direct profiles of the variables, which allows an expert to better visualize how different physical processes which impact the ABL may be at play. With the visualization of the direct profiles, one can understand better the ABL structure, so we believe that searching for the kinks this way, rather than using a second derivative profile, yields a better estimate for the ZABL.

*L290: Better something like: "Only in few especially difficult cases the Rib profiles were used heavily" or even "Only in few especially difficult cases we largely relied on the Rib profiles"?*

Based on another reviewer's comments, we have restructured/changed the end of this paragraph, but to also accommodate your comment, this now reads: "The primary methods applied to determine ZABL are those in which there are either one or two θv kinks, where we rely most heavily on the θv profile, and secondarily on the humidity and Rib profiles. For SBL cases, the humidity profiles often provide more insight than the Rib profile in identifying ZABL. In only a few especially difficult cases, we relied primarily on the Rib profiles" (lines 307-310).

*Fig 2 and similar figures: Adding y-tick marks (I don't mean labels) to all subplots would make it much easier to read the heights from e.g. the Rib profiles. In addition, you could increase the width of each subplot by decreasing the horizontal spacing between them (keep a larger space between the right and left groups (a)-(b), etc.*

I have added y-tick marks to all subplots in Figure 2, Figure 3, and Supplementary Figures S1-S69. I have also increased the width of subplots and increased the horizontal spacing between the groups (a), (b), (c), etc. for these same figures.

*L299-301: Can you refer to the corresponding examples in the supplementary figures?*

We have added a list of the corresponding examples in the supplementary figures, which were cases in which the ABL height was more ambiguous (line 319-320).

*L303: Can you make use of this 30m max. uncertainty, e.g. in the analyses related to Fig. 5 and 6?*

While we state an uncertainty in the subjective ZABL to be less than 30 m, this is only applicable to a handful of DH2 flights (~15%), whereas the majority have an uncertainty on the order of only ~1 m, due to the vertical averaging procedure and sensor response time. Therefore, we do not expect this uncertainty to make any significant effect on the results. We have added some text to the manuscript later on to mention this (line 622-625), but since we expect no significant effect on the results, we do not discuss it further.

*L306: I would argue it can be automated e.g. through machine learning, but it may be complex and time-consuming.*

We have added a note about the possibility of automation with machine learning, with the caveat that this may still not be fully reliable (lines 330).

*L314: Considering that there were only two CBL cases it might make sense to exclude CBLs entirely from the analyses carried out in this manuscript. This would make it possible to slightly shorten the methodology section, e.g. L317-319, first row in Table 3, Fig 2a, etc.*

The authors wish to include the discussion of the CBLs in the manuscript. While rare, as you yourself mentioned in a previous comments, they can be important still. More importantly, we want this paper to be a one-stop-shop for how to determine ABL height in the Arctic, so the exclusion of any discussion about a CBL would not accomplish this.

*L321-323: Could this also be related to the fact that Liu and Liang use ~40m and 160m to determine the stability regime? Can you give more details on their vertical resolution and smoothing? I also wonder whether the method would work better with more smoothing and original thresholds. This can be discussed a bit later on.*

We have added some text which discusses in more detail the difference in vertical resolution between ours and that used in Liu and Liang (2010). The vertical resolution of their data was ~40-50 m, which is much coarser than ours. This is likely the reason they were able to use a lower threshold. However, it does not make sense to interpolate the DH2 data to their resolution and use original thresholds because this would eliminate the ability the identify key features in the often shallow Arctic ABL (line 347-351). We don't believe that the need for a higher threshold has anything to do with the fact that Liu and Liang use a different altitude range to determine stability, as the altitude range they use is simply not appliable to the central Arctic environment, but it does relate back to the core issue of their having much lower resolution data.

*L356: It would be good to avoid the use of the term "critical value" for Rib here and in other occurrences. Further down you use "threshold value", but I get that you first want to indicate that this threshold value is sort of related to the critical value.*

We now use the word "threshold" rather than "critical" in this section (line 382-398), and throughout the paper when we refer to the threshold values which are applied to the DH2 and RS data. We only use the word "critical" when we explicitly refer to the standard critical Richardson number of 0.25, to which the threshold values we use are related.

*L373: "vertical resolution"*

This change has been made (line 405).

*L374-375: "(due to the lack of daytime convection or a diurnal cycle in the Arctic most of the time)" is mentioned before.*

We have removed this statement in parentheses here, as it is already discussed in the introduction.

*L382-384: I consider the sampling rate, climb rate, response time (mentioned in Table 2), and resulting vertical resolution as equally important as the accuracy specs.*

We have added information on the sampling rate, climb rate, and resulting vertical resolution for the radiosonde (line 416-417). We have also added information on the uncertainty in the pressure, temperature, and humidity measurements (line 414-415). We do not provide uncertainty from the RSS421 on the DH2 since this information is not provided in the data sheet.

*L393: "the determination of the stability regime"*

This change has been made (line 427).

*L394-395: Do these adaptations in some cases result in different stability regimes compared to the corresponding DH2 profiles?*

We have added the following sentence to address this: "These adaptations in themselves do not result in the identification of a different stability regime than is found in the DH2 profiles; instead, differences in stability regime between the two platforms may result from the lack of near-surface observations from the radiosonde, or a change in atmospheric structure between the two corresponding launches" (line 431-434).

*L396: avoid this hybrid between text and mathematical expression or at least replace "x" with "times". Better to use "\delta z".*

These changes have been made (line 430).

*L415 and 416: "relative difference"*

Another reviewer has asked we change this plot to absolute difference rather than relative difference, so this comment is no longer relevant.

*L434 and 435: You use only one "subjective method".*

We have changed the wording to reflect one "subjective method" (line 471).

*L435: Consider including "arguably" or similar. You don't have any proof supporting this statement.*

We have added "arguably" to the sentence (line 472).

*L448 ff: I suggest writing "(R2 = 0.653)" and "(slope = …", consider also replacing "slope" with e.g. "m". You can also simply use "R2" instead of "R2 value".*

These changes have all been made (line 479-522).

*L459 ff: I suggest consequently sticking to terminology like "can be considered as statistically significant". Although highly unlikely it can all be a coincidence.*

This wording has been changed as you suggest (line 496-497, 515-516).

*L478: I suggest changing to "rather strong correlation". For a strong correlation, I would expect R2>0.90 or even 0.95*

This change has been made (line 518).

*L545: "very close to the surface (~5 m)"*

This change has been made (line 604).

*L547: Consider specifying. Higher time resolution and slower climb rate result in higher vertical resolution.*

We now say: "Additionally, the DH2 samples with higher vertical resolution (due to higher time resolution of instrumentation and slower climb rate)…" (line 606-607).

*L549: What is meant by "improved"?*

What we really meant is "adjusted." This refers to the fact that some threshold values were changed from the original published methods to best work for the DH2 data, so we mean to say that the objective methods with adjusted threshold values also work well for the radiosonde data. We have changed the wording accordingly (line 609).

*L556: "10 m or 20 m bin size" or rather "10-m or 20-m bin size"*

We say "10 m or 20 m bin size" (line 616). The AMT journal specifically says they do not want authors to use a dash between numbers and units.

*L561: Why a "higher critical Rib value" and not a lower one. For the smoother RS data 0.5 works better. Consider using "threshold".*

You make a good point. We have changed this sentence to say: "For vertical resolution of 30 m or coarser, the altitude range over which Rib is calculated would have to be increased, and at this point a lower threshold Rib value may be more applicable" (line 620-621).

***L563 ff: Consider starting more general with all methods. It reads like Table 4 lists only the reasons for Liu-Liang.***

We have restructured this section to first introduce what is shown in Table 4 for all objective methods. We then go on to discuss the specific failures for each method. This should now read like Table 4 lists reasons for all methods, and no one method is singled out (lines 627-629).

***L570: A sentence on why it works relatively well for NBL cases would fit in here. May it have something to do that there is often no LLJ when the ABL is neutral?***

We have added a sentence here which reads: "The Liu-Liang method likely performs better for NBL cases (as is evident in Fig. 6 and Supplementary Figure S71) than SBL cases because the Liu-Liang method for an NBL is not dependent on the sufficient diminishment of the $\theta v$ inversion, nor the presence or altitude of a LLJ" (line 635-637).

***L614: What is meant by "qualifying values"?***

We have changed this to simply say "threshold values" (689).

***L620-621: Consider using "Arctic pack ice to near the Arctic ice edge" or "marginal ice zone" if this fits better.***

We now say: "…deep in the Arctic pack ice to near the marginal ice zone…" (line 695-696).

***L631: Consider changing "percent difference" to "relative difference" throughout the manuscript.***

We have made this change throughout the manuscript.

**Anonymous referee #3**

None